# Multi-Distribution Robust Conformal Prediction

**Yuqi Yang** [1]  **Ying Jin** [2]

## Abstract

In many fairness and distribution robustness problems, one has access to labeled data from multiple source distributions yet the test data may come from an arbitrary member or a mixture of them. We study the problem of constructing a conformal prediction set that is uniformly valid across multiple, heterogeneous distributions, in the sense that no matter which distribution the test point is from, the coverage of the prediction set is guaranteed to exceed a pre-specified level. We first propose a max-p aggregation scheme that delivers finite-sample, multi-distribution coverage given any conformity scores associated with each distribution. Upon studying several efficiency optimization programs subject to uniform coverage, we prove the optimality and tightness of our aggregation scheme, and propose a general algorithm to learn conformity scores that lead to efficient prediction sets after the aggregation under standard conditions. We discuss how our framework relates to group-wise distributionally robust optimization, sub-population shift, fairness, and multi-source learning. In synthetic and real-data experiments, our method delivers valid worst-case coverage across multiple distributions while greatly reducing the set size compared with naively applying max-p aggregation to single-source conformity scores, and can be comparable in size to single-source prediction sets with popular, standard conformity scores.

## 1. Introduction

Reliable uncertainty quantification is critical for deploying machine learning systems in high-stakes domains (Platt

[1]Mathematics Department, The Hong Kong University of Science and Technology, Hong Kong, China [2]Department of Statistics and Data Science, University of Pennsylvania, Philadelphia, PA, USA. Correspondence to: Ying Jin <yjin-stat@wharton.upenn.edu>.

*Proceedings of the 43rd International Conference on Machine Learning*, Seoul, South Korea. PMLR 306, 2026. Copyright 2026 by the author(s).

et al., 1999; Gal, 2016; Guo et al., 2017; Kompa et al., 2021). Conformal prediction is a powerful distribution-free framework for this purpose. Given any prediction model, it offers prediction sets whose coverage guarantees hold without modeling assumptions on the data generating process (Vovk et al., 2005; Lei et al., 2018).

This paper studies how to maintain such reliability when models are deployed across multiple heterogeneous environments (Crammer et al., 2008; Mansour et al., 2008; Hashimoto et al., 2018; Romano et al., 2019a). For example, a clinical risk prediction model trained on data from several hospitals must remain reliable when a new patient's record comes from one of these sites. Our goal in this work is to construct prediction sets with valid coverage even when it is impossible to reveal where that patient came from.

Formally, we assume access to labeled data $\mathcal{D} = \cup_{k=1}^{K} \mathcal{D}^{(k)}$ from $K \in \mathbb{N}^+$ heterogeneous sources, where each $\mathcal{D}^{(k)} = \{(X_i^{(k)}, Y_i^{(k)})\}_{i \in I_k}$ consists of i.i.d. samples from an unknown distribution $P^{(k)}$, with features $X_i^{(k)} \in \mathcal{X}$ and response $Y_i^{(k)} \in \mathcal{Y}$. Let $n = |\mathcal{D}|$. For a new test point $X_{n+1}$ drawn from any of these sources, we aim to build a prediction set $\hat{C}(X_{n+1}) \subseteq \mathcal{Y}$ with *uniform coverage*:

$$\min_{k \in [K]} \mathbb{P}_{\mathcal{D} \times P^{(k)}} \big( Y_{n+1} \in \hat{C}(X_{n+1}) \big) \geq 1 - \alpha, \quad (1)$$

where $\mathbb{P}_{\mathcal{D} \times P^{(k)}}$ denotes the joint distribution of the labeled data and a new test point $(X_{n+1}, Y_{n+1}) \sim P^{(k)}$. In words, $\hat{C}(\cdot)$ should achieve the nominal coverage level simultaneously for all possible sources. Several practically important scenarios motivate such a guarantee:

**Fairness without protected attributes.** Fair prediction across protected attributes such as race and gender is a central goal of equitable machine learning (Madras et al., 2018; Hashimoto et al., 2018). In this context, group-conditional coverage demands the coverage of the prediction sets to hold for all groups (Romano et al., 2019a; Jung et al., 2022; Gibbs et al., 2025). However, existing methods rely on the test group information (i.e., which distribution the test point is from) which may be unavailable or protected in sensitive scenarios (Gupta et al., 2018; Martinez et al., 2021; Lahoti et al., 2020), necessitating a single prediction set with coverage over all groups. When $P^{(k)}$ denotes a sensitive group, uniform coverage (1) provides such a guarantee.

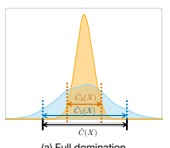 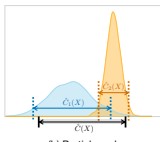 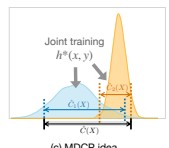

(a) Full domination    (b) Partial overlap    (c) MDCP idea

*Figure 1.* Prediction sets with uniform coverage need to balance multiple distributions. (a) When one source has heavier tails, a valid prediction set $\hat{C}(X)$ may coincide with the larger single-source set. (b) When two sources *partially overlap*, a uniformly valid prediction set $\hat{C}(X)$ may need to be longer than each single-source set. (c) MDCP achieves efficient prediction set with uniform coverage by training a conformity score and aggregating multiple prediction sets from the trained score.

**Subpopulation shift.** When each distribution represents a subpopulation, a prediction set with uniform coverage (1) protects against arbitrary subpopulation shift (Sagawa et al., 2019; Santurkar et al., 2020; Subbaswamy et al., 2021; Yang et al., 2023). Formally, subpopulation shift assumes the labeled data come from $P_{\text{train}} = \sum_{k=1}^{K} \pi_k P^{(k)}$, with non-negative mixture weights $\{\pi_k\}_{k=1}^{K}$ that sum to 1, and each $P^{(k)}$ is a subpopulation (e.g., hospitals, demographic groups). The test distribution is $P_{\text{test}} = \sum_{k=1}^{K} \pi'_k P^{(k)}$, with distinct weights $\{\pi'_k\}_{k=1}^{K}$. Any prediction set obeying (1) guarantees valid coverage under any such shift, since $P_{\text{test}}(Y_{n+1} \in \hat{C}(X_{n+1})) = \sum_{k=1}^{K} \pi'_k \mathbb{P}_{P^{(k)}}(Y_{n+1} \in \hat{C}(X_{n+1})) \geq 1 - \alpha$ once $\sum_{k=1}^{K} \pi'_k = 1$.

**Multi-source data.** Many scientific and engineering applications naturally aggregate heterogeneous datasets collected under different protocols or environments (Crammer et al., 2008; Mansour et al., 2008), which has attracted interest in conformal prediction (Lu et al., 2023; Spjuth et al., 2019; Liu et al., 2024). Standard conformal prediction (calibrated on pooled data from all sites) is valid only for the mixture distribution induced by the training sample. In contrast, (1) offers guarantees to all individual sources, ensuring reliability even when the test data align with only one of them.

To achieve uniform coverage (1) with a single prediction set $\hat{C}(X_{n+1})$, the key challenge is to balance the heterogeneous sources for reasonable efficiency (prediction set size). We demonstrate the efficiency-validity tension via two examples in Figure 1(a-b). In panel (a), the label in one source has a more dispersed distribution and therefore requires larger source-wise prediction sets; any single set that attains $1 - \alpha$ coverage for that source will typically be conservative for the more concentrated source. In panel (b), different sources place probability mass in different regions of the response, so a uniformly valid set may need to cover multiple regions and can be substantially larger than a single-source set.

### 1.1. Preview of Results

In this paper, we propose Multi-Distribution Conformal Prediction (MDCP), a general framework for constructing

efficient prediction sets that achieve the uniform coverage guarantee (1) given per-source datasets. Our starting point is a simple, distribution-free *max-p aggregation* mechanism for achieving uniform validity. For each source, we compute a conformal p-value using any conformity score, and then aggregate these p-values by taking their maximum. Inverting the aggregated p-value yields a prediction set that is exactly the union of the single-source conformal sets, and therefore delivers *finite-sample* uniform coverage.

We then proceed to study the efficiency of this aggregation when sources are heterogeneous. We analyze population-level optimization programs to theoretically characterize the optimal prediction sets with the minimal size/length subject to uniform coverage, which yield concrete guidance for score design. In specific, there exists one single conformity score function $h^*: \mathcal{X} \times \mathcal{Y} \to \mathbb{R}$, which depends on a dual problem involving all the source distributions, such that the max-p aggregation based on this score converges to the optimally efficient prediction set. Moreover, max-p aggregation is *tight*: the optimal set achieves (nearly) exact $1 - \alpha$ coverage for at least one source distribution when coupled with proper per-source conformity scores.

Building on these insights, we develop an end-to-end MDCP procedure that learns the optimal score and combines it with max-p aggregation to produce a final prediction set (Fig. 1(c)). We prove that MDCP achieves finite-sample uniform coverage (1), and it asymptotically matches the oracle optimal set under mild consistency conditions. We instantiate MDCP for both classification and regression using general learning algorithms and practical training strategies. Finally, in extensive simulations and real-data applications to satellite imagery and medical service datasets, MDCP achieves tight worst-case coverage while substantially improving efficiency over naive aggregation baselines, often approaching the size of single-source prediction sets.

We summarize our contributions as follows:

- We introduce a general max-p aggregation scheme that achieves uniform coverage based on single-source conformal p-values.
- We characterize the population-optimal prediction sets subject to uniform coverage, and show that the max-p aggregation is optimal and tight when paired with properly chosen conformity scores.
- We propose an end-to-end pipeline to learn the conformity scores and construct efficient MDCP sets for both classification and regression.
- We demonstrate the effectiveness of MDCP in extensive simulations and real data applications.

**Related work.** MDCP sits at the intersection of (i) conformal prediction with multiple data sources, (ii) robust conformal inference under distribution shift, and (iii) group-

conditional or fairness-motivated coverage guarantees. Prior work in multi-source/federated conformal prediction studies distinct settings from us such as communication-limited calibration, borrowing strength across sources, or aggregating source-specific prediction sets (Lu et al., 2023; Liu et al., 2024). Robust conformal methods typically protect against shifts modeled as perturbations around a reference distribution (Cauchois et al., 2024; Jin et al., 2023; Yin et al., 2024; Ai & Ren, 2024), while we address settings where the test distribution is an unknown member or mixture of the sources, thereby necessitating different techniques. Group-conditional and multi-valid conformal methods target coverage across subgroups, but require subgroup identity at inference (Romano et al., 2019a; Jung et al., 2022; Gibbs et al., 2025). In contrast, MDCP constructs *one* prediction set with finite-sample coverage uniformly over all source distributions without observing the test membership. Finally, this work parallels distributionally-robust optimization (DRO) in standard machine learning, and more specifically, group-DRO which optimizes worst-case performance over several groups/sources (Hashimoto et al., 2018; Sagawa et al., 2019; Mohri et al., 2019). Finally, the efficiency optimization is connected to a line of work in optimizing the volume of conformal prediction sets (Lei & Wasserman, 2014; Bai et al., 2022; Huang et al., 2023; Bars & Humbert, 2025). See Appendix A.1 for an extended discussion and references.

## 2. Max-$p$ Conformal Prediction

In this section, we introduce the general max-p aggregation scheme and establish its finite-sample uniform validity. Following split conformal prediction (Vovk et al., 2005; Lei et al., 2018), we assume each data source $\mathcal{D}^{(k)}$ is randomly split to a training fold $\mathcal{D}^{(k)}_{\text{train}}$ and a calibration fold $\mathcal{D}^{(k)}_{\text{calib}}$. We assume $\{\mathcal{D}^{(k)}_{\text{train}}\}_{k=1}^{K}$ are used to obtain any conformity score function $s_k \colon \mathcal{X} \times \mathcal{Y} \to \mathbb{R}$ associated with each distribution $k \in [K]$, which can be viewed as independent of the calibration folds and the test data.

Our method begins with single-source conformal p-values

$$p^{(k)}(y) := \frac{1 + \sum_{i \in \mathcal{D}^{(k)}_{\text{calib}}} \mathbb{1}\{s_k(X_i, Y_i) \le s_k(X_{n+1}, y)\}}{1 + |\mathcal{D}^{(k)}_{\text{calib}}|}.$$

By conformal prediction theory, inverting $p^{(k)}$ leads to a valid prediction set for the $k$-th distribution:

$$\mathbb{P}_{\mathcal{D} \times P^{(k)}}\big(Y_{n+1} \in \hat{C}^{(k)}(X_{n+1})\big) \ge 1 - \alpha, \quad (2)$$

where $\hat{C}^{(k)}(X_{n+1}) = \big\{y \in \mathcal{Y} \colon p^{(k)}(y) \ge \alpha\big\}$.

Our max-p aggregation scheme computes the maximum p-value, which is then inverted to yield the prediction set:

$$\hat{C}(X_{n+1}) = \big\{y \in \mathcal{Y} \colon p(y) \ge \alpha\big\}, \quad p(y) = \max_{k \in [K]} p^{(k)}(y).$$
$$(3)$$

It is straightforward to show that this prediction set is the union of single-source prediction sets in (2), and it enjoys finite-sample uniform validity. The proof of Theorem 1 is included in Appendix B.1.

**Theorem 1** (Finite-sample uniform validity)**.** *Let* $\{p^{(k)}(y)\}_{k=1}^{K}$ *and* $p(y)$ *be defined above. Then, the aggregated set equals the union of the per-source conformal sets:* $\hat{C}(X_{n+1}) = \bigcup_{k=1}^{K} \hat{C}^{(k)}(X_{n+1})$. *For an independent test point* $(X_{n+1}, Y_{n+1}) \sim P$ *from a mixture distribution* $P = \sum_k \pi_k P^{(k)}$ *with arbitrary weights* $\sum_{k=1}^{K} \pi_k = 1$, $\pi_k \ge 0$, *the prediction set obeys*

$$\mathbb{P}_{\mathcal{D} \times P}\big(Y_{n+1} \in \hat{C}(X_{n+1})\big) \ge 1 - \alpha,$$

*which implies the uniform coverage* (1) *as a special case.*

Despite the generality and validity of this approach, several questions remain. The first is *tightness*. Since the aggregated set is larger than any single-source prediction set that is valid in its respective distribution, it is unclear whether the worst-case per-source coverage of (3) may be way above $1 - \alpha$. The second is *efficiency*, that is, how to design the individual scores $\{s_k\}_{k=1}^{K}$ so that the aggregated set is of a reasonable size/length. If the individual sets $\hat{C}^{(k)}(X_{n+1})$ do not overlap enough, their union may be large. These are the main questions addressed in the rest of the paper.

## 3. Optimality of Max-$p$ Aggregation

In this part, we address the questions above by studying the optimality of max-p aggregation. First, we solve several population-level optimization programs to derive the optimal prediction sets with the smallest size/length subject to uniform coverage. Then, we show that our max-p aggregation is asymptotically equivalent to such optimal sets when the conformity scores converge to an optimal score.

### 3.1. Size Optimization under Uniform Validity

We begin with minimizing prediction set size/length subject to uniform validity when the distributions are known. Since our final prediction sets are calibrated to satisfy the marginal uniform coverage (1), one could study a marginal size-minimization program (we include this in Appendix A.2 for completeness). However, in the multi-source deployment setting, the test covariate distribution can be any mixture of $\{P_X^{(k)}\}_{k=1}^{K}$, so there is no canonical choice of how to average $|C(X)|$ over $\mathcal{X}$ when defining the "optimal size." Instead, we analyze a *pointwise* program: for each fixed $x \in \mathcal{X}$, minimize $|C(x)|$ subject to uniform conditional coverage across sources: $\min_{k \in [K]} \mathbb{P}_{P^{(k)}}(Y \in C(X) \mid X = x) \ge 1 - \alpha$. Importantly, we do not claim conditional validity, which is in principle impossible to achieve in finite sample (Foygel Barber et al., 2021). Rather, we use

the optimal form of prediction sets to guide the learning of conformity scores.

Let $(\mathcal{X}, \mathcal{A}, \nu)$ and $(\mathcal{Y}, \mathcal{B}, \mu)$ be finite measure spaces with $\nu(\mathcal{X}) < \infty$ and $\mu(\mathcal{Y}) < \infty$, and write $\rho := \nu \otimes \mu$, where $\mu$ is the count measure for classification and the Lebesgue measure for regression. Throughout the paper, we assume that for each $k = 1, \ldots, K$, the covariate distribution $P_X^{(k)}$ admits a density $r_k(x)$ with respect to $\nu$, and that $Y \mid X = x$ has density $f_k(\cdot \mid x)$ with respect to $\mu$. For a measurable subset $C(X) \subseteq \mathcal{Y}$, we define $|C(X)|$ as the cardinality in classification problems when $|\mathcal{Y}| < \infty$, and the Lebesgue measure in regression problems when $\mathcal{Y} = \mathbb{R}$.[1]

**Optimal prediction set under conditional validity.** Consider the following problem for any $x \in \mathcal{X}$:

$$\underset{C(\cdot)}{\text{minimize}} \ |C(x)| \ = \ \int_{\mathcal{Y}} \mathbb{1}_{\{y \in C(x)\}} \, d\mu(y) \qquad (4)$$

$$\text{s.t.} \int_{\mathcal{Y}} \mathbb{1}_{\{y \in C(x)\}} f_k(y \mid x) \, d\mu(y) \ \geq \ 1 - \alpha, \ \forall k \in [K].$$

Any sets obeying the constraints in (4) for every $x \in \mathcal{X}$ also obeys uniform marginal coverage, so the conditional feasible set is smaller than the marginal one. Therefore, integrating $|C^*(x)|$ over any distribution over $\mathcal{X}$ is no smaller than the corresponding marginal optimum (Appendix A.2).

Solving (4) amounts to a change-of-variable via the indicator function $I_x(y) := \mathbb{1}\{y \in C(x)\}$. For a clear presentation, we relax the range of $I_x(y)$ to $[0, 1]$, so that $I_x(y)$ can be viewed as the probability of $y \in C(x)$ for a randomized prediction set. This relaxation is without loss for our characterization: the objective and constraints are linear in $I_x(y)$, so an optimum is attained by an indicator except possibly on the boundary where randomization can be used to achieve exact coverage. Theorem 2 offers the form of optimal prediction sets, whose proof relies on solving the dual problem of (4), and is included in Appendix B.3.

**Theorem 2** (*X-conditional optimality*). *For a fixed $x \in \mathcal{X}$, the dual problem of* (4) *is (writing $u_+ := \max\{u, 0\}$)*

$$\max_{\{\lambda_k\}_{k=1}^K} \ (1 - \alpha) \sum_{k=1}^K \lambda_k(x) - \int_{\mathcal{Y}} (h_\lambda(x, y) - 1)_+ \, d\mu(y),$$

$$\text{where } h_\lambda(x, y) := \sum_{k=1}^K \lambda_k(x) \, f_k(y \mid x). \qquad (5)$$

*Let $\lambda^*(x) = (\lambda_1^*(x), \ldots, \lambda_K^*(x))$ be an optimal solution to* (5)*. Then an optimal solution to* (4) *is*

$$C^*(x) = \{y \in \mathcal{Y} : h_{\lambda^*}(x, y) > 1\} \cup S(x) \qquad (6)$$

*for some $S(x) \subseteq \{y \in \mathcal{Y} : h_{\lambda^*}(x, y) = 1\}$. Moreover, $\lambda^*(x)$ determines which source-wise coverage constraints are met with equality: (i) if $\lambda_k^*(x) > 0$, then $P^{(k)}(Y_{n+1} \in$*

$C^*(x) \mid X_{n+1} = x) = 1 - \alpha$; (ii) if $\lambda_k^*(x) = 0$, then $P^{(k)}(Y_{n+1} \in C^*(x) \mid X_{n+1} = x) \geq 1 - \alpha$; and (iii) there exists some $k^* \in [K]$ such that $\lambda_{k^*}^*(x) > 0$. Finally, if additionally $\mu(\{y \in \mathcal{Y} : h_{\lambda^*}(x, y) = 1\}) = 0$, then $C^*(x)$ is unique up to $\mu$-null sets.

Here, in the complementary slackness results, the coverage $P^{(k)}(Y_{n+1} \in C^*(x) \mid X_{n+1} = x)$ should be interpreted as randomizing the inclusion of $y \in \mathcal{Y}$ with some probability $I_x^*(y) \in [0, 1]$, where $I_x^*(y)$ is the optimal solution to (4).

Theorem 2 reveals that for each $x$, the optimal set keeps the smallest $\mu$-measure subset of $\mathcal{Y}$ that contains at least $1 - \alpha$ conditional probability mass under every source. The dual weights $\lambda_k^*(x)$ quantify which sources are locally hardest to cover: $\lambda_k^*(x) > 0$ iff the $k$-th constraint is active at $x$. The resulting score $h_{\lambda^*}(x, y) = \sum_{k=1}^K \lambda_k^*(x) f_k(y \mid x)$ acts as a "shared score," and the optimal set is its superlevel set at threshold 1 (plus optional boundary randomization).

### 3.2. Asymptotic Optimality of Max-p Aggregation

Having studied the population-level optimal solutions, we proceed to show that our max-p aggregation is indeed *optimal* and *tight*. Connecting Section 3.1 with max-p aggregation, our result states that, when using individual conformity scores that converge to an optimal score, the resulting prediction set converges to the optimal (while retaining finite-sample coverage due to Theorem 1).

As discussed, we focus on the (conceptually natural) conditional problem (4), and analogous results for the marginal problem follow similar ideas. Let $\lambda^*(x) \in \mathbb{R}_+^K$ be a dual maximizer for the program (4), and let $h^*(x, y) := \sum_{k=1}^K \lambda_k^*(x) f_k(y|x)$. Recall that the optimal set is of the form $C^*(x) = \{y : h^*(x, y) > 1\} \cup S^*(x)$, with $S^*(x) \subseteq T(x) := \{y : h^*(x, y) = 1\}$. Here, one may either randomize the inclusion of the boundary set $T(x)$ to achieve exact $1 - \alpha$ coverage, or include $T(x)$ with slightly inflated coverage. In classification problems, such choices would affect the prediction set size since $|T(x)|$ is non-negligible under the count measure. Therefore, to avoid over-complication, we stick to the generic form of $C^*(x)$ above and isolate $S^*(x)$ in our results.

To describe the prediction set under max-p aggregation, we follow the procedure in Section 2. Splitting each source into training and calibration folds, we write $n_k = |\mathcal{D}_{\text{calib}}^{(k)}|$. Assuming access to estimators $\hat{f}_k^{(n)}(\cdot \mid \cdot)$ and $\hat{\lambda}^{(n)}(\cdot)$ obtained from $\cup_{k=1}^K \mathcal{D}_{\text{train}}^{(k)}$, we define $\hat{h}(x, y) := \sum_{k=1}^K \hat{\lambda}_k(x) \hat{f}_k(y \mid x)$, and use the conformity score $s_k(x, y) := -\hat{h}(x, y)$ to construct the prediction set (3). To simplify boundary conditions, we adopt the randomized version of our general max-p aggregation which remains finite-sample valid. Specifically, for source $k \in [K]$ and a candidate label value $y \in \mathcal{Y}$ at a feature value

---

[1] For clarity, we assume sufficient regularity of the underlying distributions so the prediction sets considered are measurable.

$x \in \mathcal{X}$, we define the randomized p-value $p^{(k)}(x, y) := \frac{\sum_{i \in \mathcal{D}_{\text{calib}}^{(k)}} \mathbb{1}\{S_{k,i} > s_k(x,y)\} + (1 + \sum_{i \in \mathcal{D}_{\text{calib}}^{(k)}} \mathbb{1}\{S_{k,i} = s_k(x,y)\}) \cdot U_k}{n_k + 1}$, where $U_k \sim \text{Unif}([0,1])$ are i.i.d. and independent of everything else, and we write $S_{k,i} = -\hat{h}(X_i^{(k)}, Y_i^{(k)})$, and $s_k(x,y) = -\hat{h}(x,y)$. Finally, we construct our prediction set at level $\alpha \in (0,1)$ via $\hat{C}^{(n)}(x) := \{y : p(x,y) \geq \alpha\}$, where $p(x,y) := \max_k p^{(k)}(x,y)$. We use the superscript to emphasize the dependence on the sample size.

Theorem 3 provides a size guarantee for our aggregated set as $n_k \to \infty$, controlled by the tie-region size. With slight abuse of notation, we identify $C(x)$ from its graph $\{(x,y) : y \in C(x)\} \subseteq \mathcal{X} \times \mathcal{Y}$, and denote the volume as $|C| := \int_X \mu(C(x)) d\nu(x)$. The proof is in Appendix B.4.

**Theorem 3.** *Assume for each $k$, there exists constants $B_k \in \mathbb{R}^+$ such that $\sup_{x,y} f_k(y|x) \leq B_k$, and $\sup_x \|\hat{\lambda}(x) - \lambda^*(x)\|_\infty \xrightarrow{p} 0$, and $\sup_{x,y} |\hat{f}_k(y|x) - f_k(y|x)| \xrightarrow{p} 0$ as $n_k, n \to \infty$, where $\sup_x \|\hat{\lambda}(x)\|_\infty \leq M$ almost surely for a constant $M > 0$. Then we have $\sup_{x,y} |\hat{h}(x,y) - h^*(x,y)| \xrightarrow{p} 0$. Furthermore, let $T := \{(x,y) : h^*(x,y) = 1\}$ and write its measure as $\rho(T) = \int_T d\mu(y) d\nu(x)$. Then for any optimal solution $C^*$ to (4), we have*

$$\limsup_{n \to \infty} \big| |\hat{C}^{(n)}| - |C^*| \big| \leq \rho(T).$$

*Thus, if $\rho(T) = 0$, then we have $\rho(\hat{C}^{(n)} \triangle C^*) \xrightarrow{p} 0$.*

Among the conditions above, the consistency of the conditional density estimator $\hat{f}_k$ holds as per-source sample sizes tend to $+\infty$ under classical conditions for parametric and nonparametric estimation theory with appropriate estimation strategies (Van der Vaart, 2000; Györfi et al., 2002). The consistent estimation of $\lambda_k^*(\cdot)$ will be studied in Section 4.1 under standard conditions. Accordingly, the estimation error of $\hat{f}_k$ and $\hat{\lambda}_k$ translates to the gap between the size of MDCP and that of the oracle sets; however, this only affects the efficiency, and the finite-sample uniform validity is always guaranteed by max-p aggregation.

In words, Theorem 3 shows that, as long as our procedure in Section 2 is instantiated with individual scores $\{s_k(\cdot, \cdot)\}$ that are consistent for $-h^*(\cdot, \cdot)$, the MDCP set is asymptotically equivalent to $C^*(x)$ up to the boundary set $T(x)$ (whose inclusion may depend on practitioners' choice). This result has two key takeaways. First, the max-p aggregation is *optimal* up to the set $T$, since it attains the oracle-optimal prediction set size. Second, the max-p aggregation is (pointwise) *tight*, since $C^*$ is shown in Theorem 2 to achieve exact $1 - \alpha$ (conditional) coverage for at least one distribution.

**Remark 4.** *The proof of Theorem 3 also yields a characterization of $\hat{C}^{(n)}$ when $\rho(T) \neq 0$. In this case, for $\nu$-almost all $x \in \mathcal{X}$, there exists a $\mathcal{Y}$-measurable set $S_\infty(x) \subseteq T(x) \subseteq \mathcal{Y}$ such that there exists a subsequence*

$\{n^{(j)}\}$, *along which $\rho\big(\hat{C}^{(n^{(j)})} \triangle (\{h^* > 1\} \cup S_\infty)\big) \xrightarrow{p} 0$. Consequently, if $C^*$ in (6) is chosen with $S^*(x) = S_\infty(x)$, then $|\hat{C}^{(n^{(j)})}| \xrightarrow{p} |C^*|$ (even when $\rho(T) > 0$).*

While we focus on the conditional problem throughout, we shall see *empirically* that aiming for the conditionally-optimal prediction set typically leads to tight *marginal* worst-case coverage when evaluated over specific test samples.

# 4. Practical Algorithms

Having established the conceptual foundations of MDCP, in this section, we develop concrete algorithms in classification and regression problems. We focus on approximating the conditionally optimal score in Theorem 2, which relies on the conditional models $f_k(y \mid x)$ and the unknown dual functions $\{\lambda_k^*(x)\}_{k=1}^K$. A natural idea is to estimate the conditional models and approximate $s_k(x,y) := -h_{\lambda^*}(x,y)$, and couple them with the max-p aggregation.

In Section 4.1, we introduce the general dual objective that allows the estimation of $\{\lambda_k^*(x)\}_{k=1}^K$, and demonstrate the consistency of this approach under suitable conditions. We then present the concrete implementations for classification in Section 4.2 and for regression in Section 4.3, respectively.

## 4.1. Optimizing Scores via an Empirical Dual Objective

Section 3.2 motivates us to approximate the optimal solution $\lambda(\cdot)$ to the (integrated) dual problem

$$\Phi(\lambda) := \int_{\mathcal{X}} \phi_x(\lambda) d\tilde{\nu}(x) \tag{7}$$

for a properly chosen distribution $\tilde{\nu}(\cdot)$, where $\phi_x(\lambda)$ is the maximization objective in (5), and recall that $\mu(\cdot)$ is the counting (resp. Lebesgue) measure in classification (resp. regression). Theorem 5 is a simplified statement of a formal result in Appendix B.5.

**Theorem 5** (Informal). *For any $\tilde{\nu}(\cdot)$ that covers the support of $P^{(k)}(X)$, the optimal solution $\lambda^*(\cdot)$ that maximizes $\Phi(\lambda)$ in (7) coincides with the dual solution $\lambda^*(x)$ in Theorem 2.*

A convenient option is to take $\tilde{\nu}(\cdot)$ as the covariate distribution for the pooled dataset. This leads to the empirical dual objective (replacing expectation by empirical average, and unknown quantities by estimates)

$$\hat{\Phi}_{\text{marg}}(\lambda(\cdot)) := \frac{1}{n} \sum_{i \in \mathcal{D}} \left[ (1 - \hat{h}_\lambda(X_i, Y_i))_- / \hat{p}_{\text{pool}}(Y_i \mid X_i) \right]$$
$$+ (1 - \alpha) \frac{1}{n} \sum_{i \in \mathcal{D}} \sum_{j=1}^K \lambda_j(X_i), \tag{8}$$

where $\mathcal{D} = \{(X_i, Y_i)\}_{i=1}^n = \cup_{k=1}^K \{(X_i^{(k)}, Y_i^{(k)})\}_{i \in I_k}$ is the pooled dataset. Here, $\hat{h}_\lambda(x,y) = \sum_{k=1}^K \lambda_k(x) \hat{f}_k(y \mid x)$ is a plug-in estimate of $h_\lambda(x,y)$ in (6), $\hat{p}_{\text{pool}}(y \mid x)$ estimates $p_{\text{pool}}(y \mid x) = \sum_{k=1}^K w_k f_k(x,y) / \sum_{k=1}^K w_k f_k(x)$, the conditional density for the pooled data, and $w_k$ is the fraction

of the $k$-th source data among the pooled dataset. A natural idea is then to parameterize the function $\lambda(\cdot)$ and solve the empirical risk minimization (ERM) problem (8). In Appendix A.3, we justify this approach with sieve approximation; below is a simplified statement of our formal results.

**Theorem 6** (Informal). *Under smoothness conditions, with a sieve approximation for $\lambda^*(\cdot)$ and consistency of $\{\hat{f}_k(y \,|\, x)\}_{k=1}^K$ and $\hat{p}_{pool}(y \,|\, x)$, the ERM minimizer $\hat{\lambda}(\cdot)$ to (8) obeys $\|\hat{\lambda} - \lambda^*\|_\infty = o_P(1)$.*

In words, if $\{\hat{f}_k(y \,|\, x)\}_{k=1}^K$ and $\hat{p}_{\text{pool}}(y \,|\, x)$ are consistent (c.f. the discussion below Theorem 3), and the underlying distribution obey standard smoothness conditions, the ERM minimizer $\hat{\lambda}(\cdot)$ obtained from sieve estimation (Chen, 2007) is consistent. These standard conditions then lead to the asymptotic optimality of MDCP by Theorem 3.

### 4.2. Algorithm for Classification

We now state the concrete MDCP algorithm for the classification setting. Recall that we split the labeled data into the training fold $\mathcal{D}_{\text{train}} = \cup_{k=1}^K \mathcal{D}_{\text{train}}^{(k)}$ and the calibration fold $\mathcal{D}_{\text{calib}} = \cup_{k=1}^K \mathcal{D}_{\text{calib}}^{(k)}$. For each source $k$, we fit a classifier $\hat{p}_k(y \,|\, x)$ on the training fold $\mathcal{D}_{\text{train}}^{(k)}$ by any off-the-shelf algorithm. Next, we learn $\lambda^*(x)$ via solving an empirical optimization objective. We approximate the covariate-dependent, nonnegative weights $\lambda_k(x)$ via basis functions such as splines or hidden representations from neural networks. Let $\Lambda(x) \in \mathbb{R}^m$ denote the vector of basis functions evaluated at a covariate value $x$. For $K$ sources, we collect the basis coefficients into a matrix $\Theta \in \mathbb{R}^{K \times m}$, with row $\theta_j^\top$ parameterizing the $j$-th weight function. We then define

$$\lambda_k(x; \Theta) = \text{softplus}\big(\Lambda(x)^\top \theta_k\big), \quad k = 1, \ldots, K, \quad (9)$$

where $\text{softplus}(t) = \log(1 + e^t)$ is applied elementwise. Accordingly, the score function with parameter $\Theta$ is $h(x, y; \Theta) := \sum_{k=1}^K \lambda_k(x; \Theta) \cdot \hat{p}_k(y \,|\, x)$.

We fit the parameters $\hat{\Theta}$ by maximizing the Lagrangian-inspired empirical objective in Section 4.1. For a miscoverage level $\alpha$, the objective as a function of $\Theta$ is given by

$$\hat{\mathbb{E}}_{\mathcal{D}_{\text{train}}} \left[ \frac{(1 - h(X, Y; \Theta))_-}{\hat{p}_{\text{pool}}(Y \,|\, X)} + (1 - \alpha) \sum_{k=1}^K \lambda_k(X; \Theta) \right], \quad (10)$$

where $\hat{\mathbb{E}}_{\mathcal{D}_{\text{train}}}[\cdot]$ denotes the empirical mean over the pooled training fold, $(t)_- = \min\{t, 0\}$, and $\hat{p}_{\text{pool}}(y \,|\, x)$ is an estimator for $p_{\text{pool}}(y \,|\, x)$ which can be obtained by fitting a classifier over the pooled data. Finally, given the fitted parameters $\hat{\Theta}$, we define the (source-invariant) score function

$$s_k(X_i, Y_i) := -\sum_{\ell=1}^K \lambda_\ell(X_i; \hat{\Theta}) \hat{p}_\ell(Y_i | X_i), \ k = 1, \ldots, K,$$

and use them to calibrate the MDCP set following (3). The entire procedure is summarized in Algorithm 1 in Appendix A.4, which also covers regression problems below.

### 4.3. Algorithm for Regression

In regression problems, the data splitting, parameterization and estimation are similar, and we follow Section 4.2. The key difference is in fitting the conditional density function $f_k(y \,|\, x)$. While one can use any estimator, here we model $Y = \mu_k(X) + \sigma_k(X) \cdot \epsilon$ for some $\epsilon \sim N(0, 1)$. Then, we use flexible methods such as gradient boosting to estimate $\mu_k(x)$ and $\sigma_k(x)$ using each $\mathcal{D}_{\text{train}}^{(k)}$; see Appendix C.3 for a detailed procedure.

Given estimators $\hat{\mu}_k(x)$ and $\hat{\sigma}_k(x)$, our working model is $\hat{f}_k(y \,|\, x) \propto e^{-\frac{(y - \hat{\mu}_k(x))^2}{2\hat{\sigma}_k(x)^2}}$. In the single-source case, this reduces to (Lei et al., 2018). We follow the same parameterization and ERM objective as in classification to obtain parameters $\hat{\Theta}$ and scores $s_k(x, y) = -\sum_{\ell=1}^K \lambda_\ell(x; \hat{\Theta}) \hat{f}_\ell(y \,|\, x)$, which are then used to calibrate the single-source p-values and the MDCP set identical to Section 4.2.

Finally, we note that the MDCP set is nontrivial to compute since thresholding the score function $s_k(x, y) = -\sum_{\ell=1}^K \lambda_\ell(x; \hat{\Theta}) \hat{f}_\ell(y \,|\, x)$ does not necessarily lead to an interval. However, as we model $\hat{f}_k(y \,|\, x)$ as a normal distribution, the MDCP set must be the union of a finite number of intervals. This allows us to compute a super-set of our MDCP set via an efficient grid search. For brevity, we defer the details and justifications to Appendix C.4.

## 5. Simulation Studies

In this section, we assess the validity and efficiency of our algorithms in diverse classification and regression settings, and investigate how the heterogeneity and separation among sources impact the performance.

### 5.1. Simulation Settings

We outline the common setup in both classification and regression settings. We consider $K = 3$ sources, a feature dimension of $d = 10$, and a nominal level at $\alpha = 0.1$. Across all settings, features follow $X_i^{(k)} \sim \mathcal{N}(0, \Sigma)$ with $\Sigma_{ij} = 0.2 + 0.8 \, \mathbb{1}\{i = j\}$, and the heterogeneity across sources is in the conditional label distribution. In each run, we draw a set $\mathcal{I} \subset \{1, \ldots, d\}$ of size $|\mathcal{I}| = 4$ uniformly at random, so labels depend on $X$ only through $X_\mathcal{I}$. We study three suites: LINEAR, where labels depend linearly on $X_\mathcal{I}$; NONLINEAR, which replaces this dependence with a nonlinear function of $X_\mathcal{I}$ (all else unchanged); and TEMPERATURE, a linear model in which a "temperature" parameter $\tau$ controls inter-source heterogeneity or separation.

The specific data generating processes (DGPs) are given in Section 5.2 and Section 5.3 for classification and regression settings, respectively. Across all experiments, we generate $n_k = 2000$ labeled samples from each source, and randomly

split pooled data into training (37.5%), calibration (12.5%), and test (50%). We compare our MDCP Algorithm 1 with two competing methods: (1) BASELINE-SRC-$k$: Standard conformal prediction set $\hat{C}_{\text{src-}k}$ with calibration data from source $k$. (2) BASELINE-AGG: A simple max-$p$ aggregation of per-source prediction sets $\hat{C}_{\text{max-p}} := \cup_{k=1}^{K} \hat{C}_{\text{src-}k}$. This is the baseline without efficiency-oriented score learning. Finally, to demonstrate the efficiency gap from density estimation, we include an ORACLE method with access to the ground-truth density for estimating $\lambda^*$ and the scores.

The results are averaged over 100 independent runs. For fair comparison, all methods build on the same prediction models specified later, under which the single-source baseline is standard conformal prediction sets with the widely-used TPS score (Sadinle et al., 2019) in classification and the variance-adaptive score of (Lei et al., 2018) in regression.

## 5.2. Simulations in Classification Settings

We consider 6-class classification with source-dependent label distributions $f_k(y = c \mid x) \propto \exp\{\eta_{kc}(x)\}$ with $\eta_{kc}(x) = \xi_k(b_{kc} + \beta_{kc}^\top x) + \mathbb{1}\{c > 1\}g(x)$ where $c \in [6]$; see Appendix C.1.1 for the full DGP and hyperparameter sampling, Appendix C.1.2 for model-fitting details.

Figure 2 reports average coverage and set size on the pooled test set, along with worst-case coverage (LINEAR). The single-source sets lead to severe under-coverage. Due to max-$p$ aggregation, both BASELINE-AGG and MDCP achieve valid worst-case coverage, yet MDCP delivers (i) significant efficiency improvement, and (ii) tight worst-case coverage. MDCP sets are on average 34.39% smaller than BASELINE-AGG. The tight worst-case coverage suggests that complementary slackness of the conditional optimal problem remains strong marginally (Theorem 2). Finally, the efficiency of the MDCP is close to ORACLE, showing the robustness to density estimation.

In NONLINEAR settings (Figure 3), while single-source sets severely under-cover, MDCP maintains tight worst-case coverage. MDCP produces much smaller prediction sets than BASELINE-AGG. In the softplus setting, MDCP sets can be smaller than single-source sets, showing the benefits of both max-$p$ aggregation and efficiency optimization. Across settings, MDCP performs similar to ORACLE.

We further investigate the effect of the separation of sources in Figure 4. As the temperature parameter $\tau$ increases and the per-source distributions move farther apart, the coverage of single-source baseline declines. Nevertheless, MDCP maintains tight worst-case coverage and substantial efficiency gain over BASELINE-AGG, and its performance remains close to ORACLE throughout.

**Additional experiments**. As additional checks, we study (i) optimization stability and (ii) covariate shift settings.

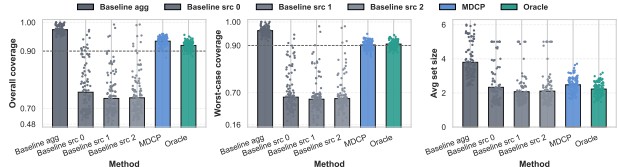

*Figure 2.* Results in the classification LINEAR experiments, where the bars are averaged over 100 runs, and the dots are per-run results. Left: coverage over all test data. Middle: worst-case coverage over single-source test data. Right: average set size over all test data.

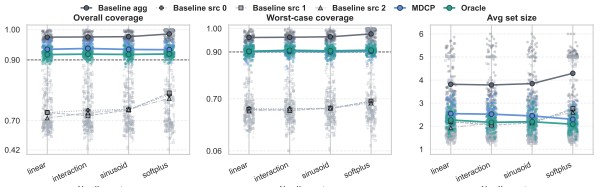

*Figure 3.* Results in the classification NONLINEAR experiments. The $x$-axis is the setting of $g(x)$. Lines show means over 100 runs; dimmed dots are per-run results. Panels as in Fig. 2.

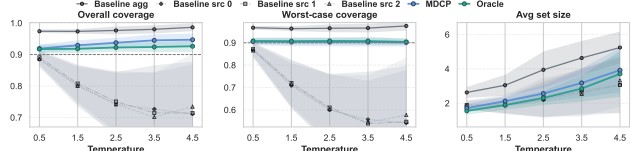

*Figure 4.* Results in the classification TEMPERATURE experiments. The $x$-axis is the parameter $\tau$. Lines show means over 100 runs with shaded $\pm 1$ standard deviation. Panels as in Fig. 2.

Appendix D.1 studies a norm-regularized version of (10) with regularization-tuning learning procedure; the results suggest the stability in learning $\hat{\Theta}$. Appendix D.3.1 explores *covariate-shift* and *combined covariate and concept shift* settings, with qualitatively similar conclusions. Appendix

## 5.3. Simulations in Regression Problems

In regression problems, we set the DGPs via $Y = \mu_k(X) + \varepsilon_k$, where $\varepsilon_k \sim \mathcal{N}(0, \sigma_k^2)$ is independent noise and $\mu_k(x) = \beta_k^\top x + b_k + g(x)$ is source-specific, across the same LINEAR, NONLINEAR, and TEMPERATURE suites; see Appendix C.1.3 for the full DGP details. We implement MDCP following Section 4.3 and compare to BASELINE-SRC-$k$ and BASELINE-AGG using the variance-adaptive conformity score of Lei et al. (2018), together with ORACLE which has access to the true density functions throughout; see Appendix C.1.4 for estimation details.

Across the regression suites (Figures 5, 6, and 7), the single-source baseline can substantially under-cover under cross-source shift, whereas BASELINE-AGG maintains worst-case validity but is often conservative, producing unnecessarily wide intervals. MDCP consistently strikes a better balance: it achieves near-nominal worst-case coverage with shorter intervals (about 22% narrower than BASELINE-AGG in LIN-

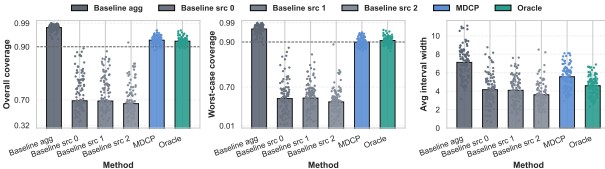

*Figure 5.* Regression LINEAR suites; details are as in Fig. 2.

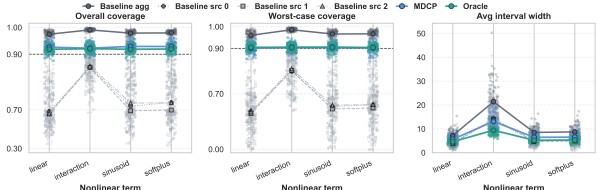

*Figure 6.* Regression NONLINEAR suites; details are as in Fig. 3.

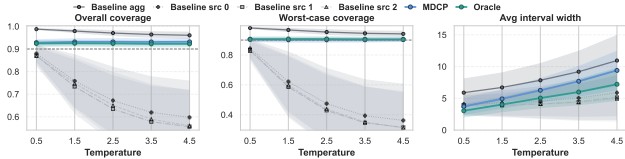

*Figure 7.* Regression TEMPERATURE suites; details are as in Fig. 4.

EAR). This pattern persists in NONLINEAR suite and, in TEMPERATURE, as inter-source separation increases with $\tau$, BASELINE-SRC-$k$ degrades, while MDCP remains tight. Finally, the moderate performance gap between MDCP and ORACLE shows the fundamental challenge of density estimation for continuous outcomes, yet such gap is much smaller than the benefit relative to the naive approach.

**Additional experiments**. Repeating optimization stability and covariate-shift ablations deliver similar messages as in Section 5.2, see Appendix D.1 and Appendix D.3.2.

### 5.4. Other Ablation Studies

In addition to the main results, we design a suite of experiments across classification and regression settings to study randomization, runtime, and efficiency of MDCP in more breadth. In Appendix D.2, we study the effect of including the tie-breaking randomness $U_k$, which shows negligible difference. In Appendix D.4 and D.5, we study the computation efficiency of MDCP, varying the sample size, number of sources, and source balance. We find that the major computation-time bottleneck is in the estimation of $\hat{\lambda}_k$. In addition, the total runtime of MDCP scales approximately linearly with the total sample size $N = \sum_{k=1}^{K} n_k$, increases moderately (and linearly) with the number of sources $K$, but remains stable with respect to group balance.

## 6. Real-Data Applications

**Data and setup.** We demonstrate the efficacy of MDCP on three real datasets exhibiting heterogeneous sources. For

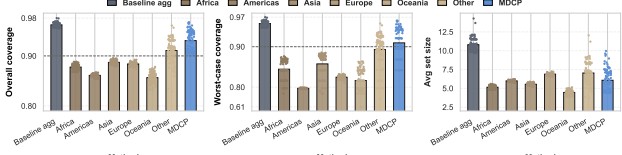

*Figure 8.* Performance of MDCP, BASELINE-AGG, and BASELINE-SRC-$k$ by region on FMoW across six sources: Africa, Americas, Asia, Europe, Oceania, and *Other*. Details are as in Fig. 2.

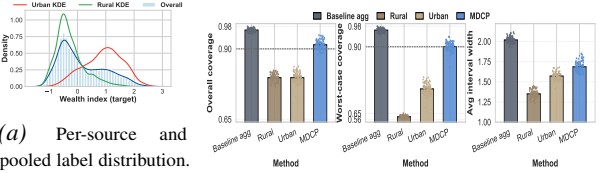

*(a)* Per-source and pooled label distribution. The curves come from a kernel density estimation.

*(b)* Performance of MDCP and baselines averaged over 100 splits. Details are as in Fig. 2.

*Figure 9.* Results of MDCP and baselines on PovertyMap.

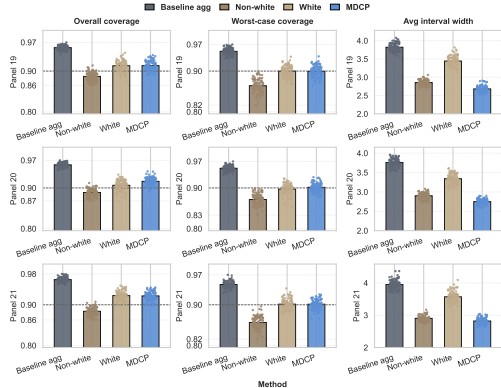

*Figure 10.* Results of MDCP and baselines on MEPS across three panels (rows). Each column corresponds to one metric, as in Fig. 2.

each dataset, we evaluate the same baselines and performance metrics as in Section 5.1. The results are averaged over 100 random splits at a nominal level of $\alpha = 0.1$. We briefly outline the datasets and method implementations below. In **FMoW** (Christie et al., 2018), a classification dataset exhibiting geographic shift, we use the 2016 time slice with 62 classes and treating six geographic regions (Africa, Americas, Asia, Europe, Oceania, *Other*) as sources; we fit pooled and region-specific probabilistic classifiers on a shared image representation, then learn $\lambda^*(x)$ and calibrate MDCP following Section 4.2. In **PovertyMap** (Yeh et al., 2020), a regression dataset used to study urban-rural shift, we leverage a 2014-2016 subset with a continuous wealth-index label and treat Urban/Rural as two sources; we fit pooled and source-specific conditional Gaussian densities using neural nets and learn $\lambda^*(x)$ as in Section 4.3. In **MEPS**, a regression dataset with heterogeneity across sensitive groups, we revisit MEPS Panels 19-21 (Agency for Healthcare Research and Quality, 2017; 2018; 2019) following Romano et al. (2019a), treating race (white vs non-white) as the source label; we log-transform the utilization

outcome, then learn conditional densities and apply MDCP as in Section 4.3. Data preprocessing, density modeling, and other experimental details are deferred to Appendix C.5.

**Experimental results.** Across all three datasets, we consistently observe that calibrating on a single source (BASELINE-SRC-$k$) can yield undercoverage when test points come from a different source, whereas naive aggregation (BASELINE-AGG) is typically conservative and produces larger prediction sets/intervals. In contrast, MDCP attains near-nominal worst-case coverage across sources while substantially improving efficiency.

Concretely, on **FMoW** (Figure 8), the geographic shift induces noticeable coverage disparities for single-source baselines; single-source sets calibrated with data from—and models trained with—data-rich regions like America turn out to severely undercover in other regions, because the calibration underestimates the uncertainty in the target region. With max-p aggregation, BASELINE-AGG remains conservative, while MDCP achieves near-tight worst-case coverage across regions with small sizes (even smaller than some single-source sets), providing an efficient solution for multi-region robust prediction sets.

On **PovertyMap** (Figure 9), Urban and Rural environments exhibit substantially different outcome distributions (Figure 9a). As such, single-source calibration does not reliably transfer across domains and leads to very coverage as low as about $0.6$. In contrast, MDCP maintains tight worst-case coverage with clear reductions in interval width relative to BASELINE-AGG, representing a robust solution to address the changes across rural and urban areas.

On **MEPS** (Figure 10), single-source baselines can undercover on the other sensitive group and/or yield wide intervals, whereas MDCP achieves near-nominal worst-case coverage and shorter intervals on average. In this dataset, MDCP provides equalized coverage across sensitive groups without requiring the group labels at test time. Indeed, the resulting uniformly valid prediction sets can even be more efficient than the single-source ones.

Finally, similar to the simulations, we study a penalty-tuning variant of MDCP, which leads to similar performance and further testifies the stability of our estimation procedure in these applications; see Appendix D.1.2.

## 7. Conclusion

MDCP builds one prediction set with finite-sample worst-case coverage across heterogeneous sources without test-source labels, using max-p aggregation and an optimized score. It is tight, asymptotically optimal, and empirically efficient, offering robust solutions to various problems in multi-distribution robustness and fairness. Open questions

include extending to regression scores that yield interval sets, and establishing guarantees with inferred sources or subpopulation shifts with unknown calibration sources.

## Reproducibility Statement

Reproduction code is available at https://github.com/AragornBFRer/MDCP.

## Impact Statement

This work advances the reliability of machine learning systems deployed in heterogeneous environments by offering multi-distribution robustness in uncertainty quantification. By ensuring worst-case coverage without requiring source labels at test time, our method advances algorithmic fairness, protecting marginalized subgroups against subpopulation shifts without necessitating access to sensitive attributes during inference. This framework is useful for quantifying the uncertainty of ML models in high-stakes applications in healthcare and policy, where it mitigates the risk of localized model failure in multiple environments.

**Limitations and potential negative societal impacts.** While our framework aims to mitigate the distribution shift issue across distributions, some limitations and risks are worth discussing. First, the worst-case marginal coverage guarantee operates as a "high-probability" statement at the population level and does not preclude risk for specific individuals. Second, in fairness-motivated applications, we still require source/group labels during calibration, which might not be available in highly sensitive scenarios. It would be interesting to study whether similar ideas may apply if one has access to a pooled dataset without group labels. Finally, if calibration data for a source is scarce, MDCP may produce unnecessarily wide prediction sets for that group, potentially lowering the utility in the underrepresented groups even though the uncertainty quantification is valid.

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

# A. Deferred Discussion

## A.1. Related Work

**Multi-source/distribution conformal prediction.** Our setting is connected to several recent work on conformal prediction from multiple data sources, where the goals vary, including learning with limited communications across sources (Lu et al., 2023), using other sources to improve efficiency in one source (Liu et al., 2024), aggregating individual prediction sets without data sharing (Spjuth et al., 2019) for i.i.d. data (Humbert et al., 2023), and leveraging density ratio for coverage over one single test distribution (Plassier et al., 2024). Most recently, Duchi et al. (2024) consider the hierarchical setting, using jackknife and data-splitting methods to ensure coverage for a new, previously unseen test-time distribution drawn exchangeable from a super-population. In contrast, our goal is to leverage all data sources during training to construct a uniformly valid prediction set for a new test point drawn from any of the source distributions, thereby motivating different techniques and leading to different guarantees.

**Distributionally-robust conformal prediction.** This work falls broadly into the strand in conformal prediction regarding distribution robustness, which studies the construction of prediction sets with valid coverage when the test distribution differs from that of the labeled data. Earlier works assume the test distribution is unidentified but is under various types of perturbations around the labeled data distribution, and seeks to protect against the worst-case among these perturbations, such as a divergence ball (Cauchois et al., 2024) or conditional shift within a divergence ball (Jin et al., 2023; Yin et al., 2024; Ai & Ren, 2024), and adversarial attacks (Gendler et al., 2021; Ghosh et al., 2023). This work can be viewed as distributionally-robust conformal prediction when the test distribution is an unknown member of the source distributions or an arbitrary mixture of them. The structure in our setting makes valid calibration possible. Compared with the covariate shift and label shift settings (Tibshirani et al., 2019), in our setting the test distribution is non-identifiable, and both $P^{(k)}(X)$ and $P^{(k)}(Y \mid X)$ are allowed to change across sources. These distinctions necessitate different techniques than these works.

**Group-conditional conformal prediction.** Within conformal prediction, our setting is close to the group-conditional conformal prediction, sometimes framed for fairness (Romano et al., 2019a) and extended to multi-validity (Jung et al., 2022; Gibbs et al., 2025). MDCP can be viewed as addressing a related problem when a distribution represents a group-conditional distribution. In contrast, however, our method achieves so without observing the group label at test time, which can be particularly useful in sensitive scenarios with protected labels. In particular, Gibbs et al. (2025) view the group membership as part of the covariate, and cast the problem as protecting against a family of covariate shifts (due to changes in the group membership), which further relates to conformal prediction under covariate shift (Tibshirani et al., 2019). However, once the group membership is unknown, the distribution shift between labeled sources and test data cannot be captured by a covariate shift assumption. As such, the techniques we develop are in sharp contrast to those methods.

**Distribution robustness and multi-source/group learning.** This work is connected to a rich line of work on the robustness to distribution shift and heterogeneity across data sources. Classical domain adaptation and multi-source learning frameworks aim to generalize models across environments with distinct data-generating mechanisms (Crammer et al., 2008; Mansour et al., 2008; Ben-David et al., 2010). In the modern ML setting, robust optimization and distributionally robust optimization (DRO) offer a complementary worst-case perspective (Ben-Tal et al., 2009; Rahimian & Mehrotra, 2022), and group-robust variants (e.g., group-DRO or relatedly, agnostic federated learning) seek worst-case performance over groups or mixtures of source distributions (Hashimoto et al., 2018; Sagawa et al., 2019; Mohri et al., 2019; Santurkar et al., 2020; Lahoti et al., 2020; Martinez et al., 2020; Subbaswamy et al., 2021; Yang et al., 2023). Our approach parallels this perspective but operates in the space of coverage guarantees rather than loss minimization: MDCP ensures valid coverage holds for all source distributions, serving as a analogue of group-DRO in uncertainty quantification with exact finite-sample validity.

**Efficiency optimization in conformal prediction.** The efficiency optimization component of MDCP is connected to a body of literature in conformal prediction that optimizes the size/volume of prediction sets. The approach we take is to work out the population-level optimality results and plug in estimations of unknown quantities, similar to (Lei & Wasserman, 2014). Alternatively, another line of work use a subset of data to optimize an empirical estimate of prediction set size (Bai et al., 2022; Huang et al., 2023; Bars & Humbert, 2025), which may inspire alternative strategies to minimize MDCP set size.

## A.2. Optimality under Marginal Coverage

In this part, we study the optimal prediction set under marginal validity. We consider the following optimization problem:

$$\underset{C(X) \subseteq \mathcal{Y}, \text{ measurable}}{\text{minimize}} \quad \int_{\mathcal{X}} |C(x)| \, d\nu(x) \tag{11}$$

$$\text{subject to } \mathbb{P}^{(k)}(Y \in C(X)) \geq 1 - \alpha, \quad \forall k = 1, \ldots, K.$$

We integrate $|C(x)|$ over $\nu(\cdot)$ to ensure a scalar objective. By definition, (11) seeks the measurable prediction set with the smallest size that achieves uniform coverage. Rigorously speaking, by "measurable", we mean $\mathbb{1}\{y \in C(x)\}$ is a measurable function on $\mathcal{X} \times \mathcal{Y}$, or $C(x)$ is a measurable subset of $\mathcal{Y}$ for $\nu$-a.s. all $x \in \mathcal{X}$.

Solving (11) amounts to a change-of-variable via the indicator function $I(x, y) := \mathbb{1}\{y \in C(x)\}$. For a clear presentation, we relax the range of $I(x, y)$ to $[0, 1]$, so that $I(x, y)$ can be viewed as the probability of $y \in C(x)$ for a randomized prediction set. The optimization problem becomes

$$\underset{I(x,y)\in[0,1], \text{ measurable}}{\text{minimize}} \quad \iint_{\mathcal{X}\times\mathcal{Y}} I(x, y) d\rho(x, y) \tag{12}$$

$$\text{subject to} \quad \iint_{\mathcal{X}\times\mathcal{Y}} I(x, y) r_k(x) f_k(y \,|\, x) d\rho(x, y) \geq 1 - \alpha, \quad \forall k = 1, \ldots, K.$$

Theorem 7 characterizes the globally optimal prediction set with smallest size subject to uniform validity, whose proof is in Appendix B.2. Here, the coverage probability should be understood as that of a randomized prediction set with probability $I(x, y) \in [0, 1]$.

**Theorem 7** (Marginal optimality). *Consider the marginal size-minimization problem* (12). *There exists a vector of nonnegative constants* $\lambda^* = (\lambda_1^*, \ldots, \lambda_K^*) \in \mathbb{R}_+^K$ *such that with* $h_\lambda(x, y) = \sum_{k=1}^{K} \lambda_k \, r_k(x) \, f_k(y \,|\, x)$, *one optimal solution* $C^*$ *to* (12) *takes the following form:*

$$C^*(x) \;=\; \{\, y \in \mathcal{Y} \colon h_{\lambda^*}(x, y) > 1 \,\} \;\cup\; S(x), \qquad S(x) \subseteq \{\, y \in \mathcal{Y} \colon h_{\lambda^*}(x, y) = 1 \,\}.$$

*In particular,* $\lambda^* = (\lambda_1^*, \ldots, \lambda_K^*) \in \mathbb{R}_+^K$ *is the optimal solution to the dual problem*

$$\Phi(\lambda) = (1 - \alpha) \sum_{k=1}^{K} \lambda_k \;-\; \int_{\mathcal{X}} \int_{\mathcal{Y}} (h_\lambda(x, y) - 1)_+ \, d\mu(y) \, d\nu(x), \tag{13}$$

*where* $(h_\lambda(x, y) - 1)_+ = \max\{h_\lambda(x, y) - 1, 0\}$. *Moreover, the complementary slackness holds:*

(i) *If* $\lambda_k^* > 0$ *then the k-th constraint is active, with* $P^{(k)}(Y \in C^*(X)) = 1 - \alpha$;

(ii) *If* $\lambda_k^* = 0$ *then the k-th constraint is (weakly) inactive, with* $P^{(k)}(Y \in C^*(X)) \geq 1 - \alpha$;

(iii) *There exists at least one* $k^* \in [K]$ *such that* $\lambda_{k^*}^* > 0$ *and* $P^{(k^*)}(Y \in C^*(X)) = 1 - \alpha$.

*If additionally* $\mu(\{y : h_\lambda(x, y) = 1\}) = 0$ *for* $\nu$-a.e. $x$, *then* $C^*$ *is unique up to* $(\nu \otimes \mu)$-null sets.

Theorem 7 reveals the central role of a single score function $h_{\lambda^*}(x, y)$: the optimal solution $C^*(x)$ is determined by thresholding this score value. Whether to include the boundary set $\{y : h_{\lambda^*}(x, y) = 1\}$ in the prediction set is often subject to users' preference. If $h_{\lambda^*}(x, y)$ does not have point mass over $\nu \otimes \mu$, the inclusion of the boundary set does not affect the average size or coverage probability. Otherwise, one need to randomize the inclusion to achieve exact $1 - \alpha$ coverage, or include it with slight over-coverage.

The complementary slackness in statement (iii) of Theorem 7 is worth noting: there always exists one source distribution under which $C^*(X)$ achieves exact $1 - \alpha$ coverage. (In the presence of point mass, such coverage should be understood as that of a randomized prediction set with $\mathbb{P}(y \in C^*(x)) = I^*(x, y) \in [0, 1]$).

Finally, we remark that since the objective of (11) integrates over the base measure $\nu(\cdot)$, the solution does not necessarily aim for the smallest average size for *the* test distribution in a specific problem. Arguably, it would be more natural to study the conditional problem in the main paper for a fixed $x \in \mathcal{X}$ subject to conditional uniform coverage, in which case the objective is inherently a scalar.

## A.3. Sieve Estimation Theory

In this section, we present and theoretically justify such a procedure using the method of sieves (Geman & Hwang, 1982). The sieve analysis below is one concrete way to verify the high-level consistency assumptions in Theorem 3. Practically, we implement $\lambda(\cdot)$ using splines or neural networks (see both Sections 4.2, 4.3 and experiments in Sections 5 and 6).

Consider an increasing sequence $\Theta_1 \subset \Theta_2 \subset \cdots$ of spaces of smooth functions. To be consistent with the language of ERM, we write the loss function and our estimator via

$$\hat{\lambda}(\cdot) = \operatorname*{argmin}_{\lambda(\cdot) \in \Theta_n} \hat{\mathbb{E}}_n \big[\hat{\ell}(\lambda(\cdot), X, Y)\big], \tag{14}$$

$$\text{where} \quad \hat{\ell}(\lambda(\cdot), x, y) = -(1 - \hat{h}_\lambda(x, y))_- / \hat{p}_{\text{pool}}(y \mid x)$$
$$- (1 - \alpha) \textstyle\sum_{k=1}^{K} \lambda_k(x).$$

Here, $(X_i, Y_i)$ are from the distribution of the pooled data $p_{\text{pool}}(x, y)$, $\hat{p}_{\text{pool}}$ is a pre-trained estimator, and $\hat{h}_\lambda(x, y) := \sum_{k=1}^{K} \lambda_k(x) \hat{f}_k(y \mid x)$ is the plug-in estimate of $h_\lambda(x, y)$ given the per-source models.

We consider two examples of sieves inspired by (Yadlowsky et al., 2022; Jin et al., 2022).

**Example 8** (Polynomials). *Let $Pol(J, \epsilon)$ be the space of $J$-th order polynomials on $[0, 1]$ truncated at $\epsilon > 0$:*

$$Pol(J, \epsilon) = \Big\{ x \mapsto \max\{\epsilon, \textstyle\sum_{j=0}^{J} a_j x^j\} : a_j \in \mathbb{R} \Big\}.$$

*Then we define $\Theta_n = \Theta_{n,0}^K$, where $\Theta_{n,0} = \{x \mapsto \prod_{j=1}^{d} f_j(x_j) : f_j \in \text{Pol}(J_n, 0), j = 1, \ldots, d\}$ for $J_n \to \infty$.*

**Example 9** (Splines). *Let $0 = t_0 < \cdots < t_{J+1} = 1$ be knots that satisfy $\frac{\max_{0 \le j \le J}(t_{j+1} - t_j)}{\min_{0 \le j \le J}(t_{j+1} - t_j)} \le c$ for some $c > 0$. We define the space for $r$-th order truncated splines with $J$ knots as*

$$Spl(r, J) = \Big\{ x \mapsto \max \big\{\epsilon, \ \textstyle\sum_{k=0}^{r-1} a_k x^k + \sum_{j=1}^{J} b_j (x - t_j)_+^{r-1} \big\} : a_k, b_k \in \mathbb{R} \Big\}$$

*Then we define $\Theta_n = \Theta_{n,0}^K$, where $\Theta_{n,0} = \{x \mapsto \prod_{j=1}^{d} f_j(x_j) : f_j \in \text{Spl}(J_n, 0), j = 1, \ldots, d\}$ for $J_n \to \infty$.*

In both examples, we consider coordinate-wise function $\{f_j(x_j)\}_{j=1}^d$ for $\mathcal{X} \subseteq \mathbb{R}^d$ in a sieve series, so that $\prod_{j=1}^d f_j(x_j) \in \Theta_{n,0}$, and the optimal dual variables $\lambda^*(\cdot) \colon \mathcal{X} \to \mathbb{R}^K$ is approximated by elements in $\Theta_n = \Theta_{n,0}^K$. Here, we truncate the functions away from zero for simplicity. Note that if $\lambda_k^*(x)$ is always positive and continuous and $\mathcal{X}$ is a compact set, then there exists a positive $\epsilon > 0$ such that $\inf_{x \in \mathcal{X}} \lambda_k^*(x) \ge \epsilon$. In practice, we can set $\epsilon$ to be small enough, or let $\epsilon = \epsilon_n$ decays slowly to zero.

Next, we show the convergence of $\hat{\lambda}(\cdot) \in \Theta_n \to \lambda^*(\cdot)$. For $p_1 = \lceil p \rceil - 1$ and $p_2 = p - p_1$, we define

$$\Lambda_c^p = \left\{ h \in C^{p_1}(\mathcal{X}) \colon \sup_{\substack{x \in \mathcal{X} \\ \sum_{l=1}^d \alpha_l < p_1}} |D^\alpha h(x)| \quad + \sup_{\substack{x \not\in x' \in \mathcal{X} \\ \sum_{l=1}^d \beta_l = p_1}} \frac{|D^\beta h(x) - D^\beta h(x')|}{\|x - x'\|^{p_2}} \le c \right\}$$

To ensure non-negativeness, we define the truncated function class $\Lambda_{c,+}^p := \{x \mapsto \max\{f(x), 0\} : f \in \Lambda_c^p\}$. We denote the oracle minimizer and loss function as

$$\lambda^*(\cdot) = \operatorname*{argmin}_{\lambda \in \Theta = (\Lambda_{c,+}^p)^K} \mathbb{E}_{\text{pool}}[\ell(\lambda(\cdot), X, Y)], \tag{15}$$

$$\text{where} \quad \ell(\lambda(\cdot), x, y) = -(1 - h_\lambda(x, y))_- / p_{\text{pool}}(y \mid x) - (1 - \alpha) \textstyle\sum_{k=1}^K \lambda_k(x).$$

Throughout, $\mathbb{E}_{\text{pool}}[\cdot]$ denotes the expectation under the pooled distribution, and $\hat{\ell}(\cdot)$ is viewed as fixed. Assuming the optimal dual functions in Theorem 2 obeys $\lambda^* \in \Theta = (\Lambda_{c,+}^p)^K$, Theorem 5 ensures the minimizer $\lambda^*(\cdot)$ in (15) coincides with the optimal $\lambda^*(\cdot)$ in Theorem 2; we thus use the same notation.

Similar to Jin et al. (2022, Theorem 1), we can show that the solution (14) is close to $\lambda^*$ once $\hat{p}_{\text{pool}}$ and $\hat{f}_k$'s are accurate. Our formal results build on the following two assumptions. Assumption 10 is a natural condition that the estimation error in

$\hat{p}_{\text{pool}}$ translates to errors in population risk minimizer of the same order. Assumption 11 collects regularity conditions that are standard in the literature and hold for convex and smooth functions; it is needed to derive rates, but consistency holds under even weaker conditions.

**Assumption 10.** *Assume* $\|\lambda^* - \bar{\lambda}^*\|_{L_2} = O_P(\|\hat{p}_{\text{pool}} - p_{\text{pool}}\|_{L_2})$ *and* $\|\lambda^* - \bar{\lambda}^*\|_\infty = O_P(\|\hat{p}_{\text{pool}} - p_{\text{pool}}\|_\infty)$.

**Assumption 11.** *Suppose* $\mathcal{X} = \prod_{j=1}^d \mathcal{X}_j$ *is the Cartesian product of compact intervals, and* $\theta^* \in \Theta = (\Lambda_c^p)^K$ *for some* $c > 0$. *Suppose* $P_{pool}$ *has positive density on* $\mathcal{X}$. *We assume the function* $\mathbb{E}_{pool}[\hat{\ell}(\lambda, x, Y) \mid X = x]$ *is* $\eta$-*strongly convex at* $\bar{\lambda}^*(x)$ *for all* $x \in \mathcal{X}$. *Also,* $|\hat{\ell}(\theta, x, y) - \hat{\ell}(\bar{\lambda}^*, x, y)| \le \bar{\ell}(x, y)\|\theta(x) - \bar{\lambda}^*(x)\|_2$ *for* $\|\theta(x) - \bar{\lambda}^*(x)\|_2 < \epsilon$ *for sufficiently small* $\epsilon > 0$, *where* $\|\cdot\|_2$ *is the Euclidean norm, and* $\sup_{x \in \mathcal{X}} \mathbb{E}_{pool}[\bar{\ell}(x, Y)^2] < M$ *for some constant* $M > 0$. *Furthermore, there exists a constant* $C_1$ *such that* $\mathbb{E}_{pool}[\hat{\ell}(\theta, X, Y) - \hat{\ell}(\bar{\lambda}^*, X, Y)] \le C_1 \|\theta - \bar{\lambda}^*\|_{L_2}^2$ *when* $\theta \in (\Lambda_c^p)^K$ *and* $\|\theta - \bar{\lambda}^*\|_{L_2}$ *is sufficiently small.*

Theorem 12 justifies using sieve approximation and ERM to learn the functions $\lambda^*(\cdot)$. Its proof largely follows (Jin et al., 2022), and is included in Appendix B.6 for completeness.

**Theorem 12.** *Under Assumptions 10 and 11, We set* $J_n \asymp (n/\log n)^{1/(2p+d)}$ *for the sieve estimators in Examples 8 and 9, and suppose* $\hat{\lambda} = \text{argmin}_{\theta \in \Theta} \hat{\mathbb{E}}_{\text{pool}}[\hat{\ell}(\theta, X, Y)]$. *Then employing the function classes in the two examples, we have* $\|\hat{\lambda} - \lambda^*\|_{L_2} = O_P((\frac{\log n}{n})^{p/(2p+d)}) + O_P(\|\hat{p}_{\text{pool}} - p_{\text{pool}}\|_{L_2})$ *and* $\|\hat{\lambda} - \lambda^*\|_\infty = O_P((\frac{\log n}{n})^{2p^2/(2p+d)^2}) + O_P(\|\hat{p}_{\text{pool}} - p_{\text{pool}}\|_\infty)$.

Our results so far justify an ERM approach to learn the unknown $\lambda_k^*(x)$ and approximate the optimal MDCP set. Suppose that the true optimal dual functions $\lambda_k^*(x)$ in Theorem 2 is sufficiently smooth in $x$, and that we solve the ERM (14) with a suitable sieve class based on a consistent $\hat{p}_{\text{pool}}(y \mid x)$. Theorem 12 then ensures $\hat{\lambda}(\cdot)$ converges to $\lambda^*(\cdot)$. Thus, as long as the $\hat{f}_k(\cdot \mid \cdot)$'s are consistent, taking $s_k(x, y) = -\sum_{k=1}^K \hat{\lambda}_k(x) \hat{f}_k(y \mid x)$ yields an MDCP set that is asymptotically optimal.

### A.4. Detailed Algorithm

As in Section 3.2, to simplify boundary conditions, we adopt the randomized version of our general max-p aggregation which remains finite-sample valid. Specifically, for source $k \in [K]$ and a candidate label value $y \in \mathcal{Y}$ at a feature value $x \in \mathcal{X}$, we define the randomized p-value

$$p^{(k)}(x, y) := \frac{\sum_{i \in \mathcal{D}_{\text{calib}}^{(k)}} \mathbb{1}\{S_{k,i} > s_k(x, y)\} + (1 + \sum_{i \in \mathcal{D}_{\text{calib}}^{(k)}} \mathbb{1}\{S_{k,i} = s_k(x, y)\}) \cdot U_k}{n_k + 1}, \tag{16}$$

where $U_k \sim \text{Unif}([0, 1])$ are i.i.d. and independent of everything else, and we write $S_{k,i} = -\hat{h}(X_i^{(k)}, Y_i^{(k)})$, and $s_k(x, y) = -\hat{h}(x, y)$.

## B. Technical Proofs

### B.1. Proof of Theorem 1

*Proof of Theorem 1.* Since $\sup_{j \in [K]} p^{(j)}(y) \le \alpha$ implies $p^{(k)}(y) \le \alpha$, we have

$$\mathbb{P}\big(\sup_{j \in [K]} p^{(j)}(Y_{n+j}) \le \alpha\big) \le \mathbb{P}\big(p^{(k)}(Y_{n+1}) \le \alpha\big) \le \alpha,$$

under $P^{(k)}$ and $\mathcal{D}$. The coverage statement follows by complement. The equality $\hat{C} = \bigcup_k \hat{C}^{(k)}$ is immediate from the definition of the supremum and the threshold rule:

- If $y \in \{y \in \mathcal{Y} : \sup_{j \in [K]} p^{(j)}(y) > \alpha\}$, then $y \in \bigcup_{j=1}^K \{y \in \mathcal{Y} : p^{(j)}(y) > \alpha\}$, since $\exists j$ s.t. $p^{(j)}(y) > \alpha$.

- If $y \in \bigcup_{j=1}^K \{y \in \mathcal{Y} : p^{(j)}(y) > \alpha\}$, then $\exists k$ s.t. $p^{(j)}(y) > \alpha$, then $y \in \{y \in \mathcal{Y} : \sup_{j \in [K]} p^{(j)}(y) > \alpha\}$.

This concludes the proof of Theorem 1. $\qquad\square$

---

**Algorithm 1** Multi-Distribution Conformal Prediction (MDCP)

---

**Input:** Data $\mathcal{D} = \cup_{k=1}^{K} \mathcal{D}^{(k)}$ from $K$ sources, test input $X_{n+1}$, significance level $\alpha$, problem MODE.

1: Split the data $\mathcal{D}$ into $\mathcal{D}_{\text{train}} = \cup_{k=1}^{K} \mathcal{D}_{\text{train}}^{(k)}$ and $\mathcal{D}_{\text{calib}} = \cup_{k=1}^{K} \mathcal{D}_{\text{calib}}^{(k)}$.
2: // Train per-source models
3: **if** MODE = classification **then**
4:     Fit any classifier $\hat{p}_k(y \mid x)$ on $\mathcal{D}_{\text{train}}^{(k)}$ for $k \in [K]$ and $\hat{p}_{\text{pool}}(y \mid x)$ on $\mathcal{D}_{\text{train}}$.
5: **else if** MODE = regression **then**
6:     Fit conditional density estimator $\hat{f}_k(y \mid x)$ on $\mathcal{D}_{\text{train}}^{(k)}$ via, e.g., conditional gaussian model for $k \in [K]$ and a pooled estimator $\hat{f}_{\text{pool}}(y \mid x)$ on $\mathcal{D}_{\text{train}}$.
7: **end if**
8: // Fit Lagrange multiplier $\lambda(\cdot)$
9: Solve the empirical objective (10) on $\mathcal{D}_{\text{train}}$ to obtain spline parameters $\hat{\Theta}$.
10: // MDCP set on test point $x$
11: **if** MODE = classification **then**
12:     Set $s_k(x, y) = -\sum_{\ell=1}^{K} \lambda_\ell(x; \hat{\Theta}) \hat{p}_\ell(y \mid x)$ via (9) for all $k \in [K]$.
13:     Compute $s_k(X_{n+1}, y)$ for all $y \in \mathcal{Y}$, and $p^{(k)}(y)$ with $\mathcal{D}_{\text{calib}}^{(k)}$ using (16) for $k \in [K]$.
14:     Compute $\hat{C}(x) = \{y : p(y) \geq \alpha\}$ with $p(y) = \max_k p^{(k)}(y)$.
15: **else if** MODE = regression **then**
16:     Set $s_k(x, y) = -\sum_{\ell=1}^{K} \lambda_\ell(x; \hat{\Theta}) \hat{f}_\ell(y \mid x)$ via (9) for all $k \in [K]$.
17:     Generate $y$-grid and use a grid search to construct prediction set $\hat{C}(X_{n+1})$ (Appendix C.4).
18: **end if**

**Output:** Prediction set $\hat{C}(X_{n+1})$.

---

### B.2. Proof of Theorem 7

*Proof of Theorem 7.* Write the joint density function $w_k(x, y) := r_k(x) f_k(y \mid x)$. The primal problem can be expressed as

$$\min_{I \in \{0,1\}} \iint I \, d\mu d\nu \quad \text{s.t.} \quad \iint I \, w_k \, d\mu d\nu \geq 1 - \alpha, \quad k = 1, \dots, K.$$

Relax $I \in \{0, 1\}$ to $I \in [0, 1]$. Since functions $I \in [0, 1]$ form a vector space (Luenberger, 1997), we consider the Lagrangian with constant multipliers $\lambda_k \geq 0$:

$$\mathcal{L}(I, \lambda) = \iint I(x, y) \, d\mu d\nu - \sum_{k=1}^{K} \lambda_k \Big( \iint I(x, y) \, w_k(x, y) \, d\mu d\nu - (1 - \alpha) \Big).$$

Let $h_\lambda(x, y) := \sum_{k=1}^{K} \lambda_k w_k(x, y)$. Then we have

$$\mathcal{L}(I, \lambda) = \iint I(x, y) \, [1 - h_\lambda(x, y)] \, d\mu d\nu \; + \; (1 - \alpha) \sum_{k=1}^{K} \lambda_k.$$

For a fixed value of $\lambda$, minimizing over $I(x, y) \in [0, 1]$ pointwise in $(x, y)$ yields the minimizer

$$I_\lambda^*(x, y) \in \begin{cases} \{1\}, & h_\lambda(x, y) > 1, \\ [0, 1], & h_\lambda(x, y) = 1, \\ \{0\}, & h_\lambda(x, y) < 1, \end{cases}$$

which yields the threshold form

$$C_\lambda(x) = \{y : h_\lambda(x, y) > 1\} \cup S(x), \quad S(x) \subseteq \{y : h_\lambda(x, y) = 1\}. \tag{17}$$

After minimizing over $I$, the dual objective is

$$\Phi(\lambda) = (1 - \alpha) \sum_{k=1}^{K} \lambda_k \; - \; \iint (h_\lambda(x, y) - 1)_+ \, d\mu(y) \, d\nu(x),$$

this gives the marginal dual objective (13) mentioned in the theorem, and the dual problem is to maximize $\Phi(\lambda)$ over $\lambda \in \mathbb{R}_+^K$.

Note that Slater's condition holds (e.g., $C(x) \equiv \mathcal{Y}$ strictly satisfies each constraint for $\alpha \in (0,1)$), so strong duality applies and a dual maximizer $\lambda^*$ exists (Luenberger, 1997). Let $\lambda^*$ be a dual maximizer and define $h^*(x,y) = \sum_k \lambda_k^* r_k(x) f_k(y \mid x)$ and the tie set $T(x) = \{y : h^*(x,y) = 1\}$. There exists a primal optimizer

$$I^*(x,y) = \mathbb{1}\{h^* > 1\} + Z^*(x,y)\,\mathbb{1}\{y \in T(x)\},$$

with $Z^* : X \times Y \to [0,1]$ measurable, chosen so that

$$\lambda_k^* > 0 \quad \Rightarrow \quad \int_{\mathcal{X}} \int_{\mathcal{Y}} I^* r_k f_k \, d\mu(y) \, d\nu(x) = 1 - \alpha,$$

$$\lambda_k^* = 0 \quad \Rightarrow \quad \int_{\mathcal{X}} \int_{\mathcal{Y}} I^* r_k f_k \, d\mu(y) \, d\nu(x) \geq 1 - \alpha,$$

where the covariate distribution $P_X^{(k)}$ admits a density $r_k(x)$ with respect to $\nu$. Equivalently, writing

$$a_k := \int_{\mathcal{X}} \int_{\mathcal{Y}} \mathbb{1}\{h^* > 1\} r_k f_k \, d\mu(y) \, d\nu(x), \qquad b_k := \int_{\mathcal{X}} \int_{y \in T(x)} Z^* r_k f_k \, d\mu(y) \, d\nu(x),$$

$Z^*$ must satisfy $a_k + b_k = 1 - \alpha$, for all $k$ with $\lambda_k > 0$ and $a_k + b_k \geq 1 - \alpha$ for all $k$ with $\lambda_k = 0$. When multiple constraints with their Lagrangian multiplier $\lambda_k > 0$, achieving all equalities generally requires a non-constant $Z^*$ (for example, using randomized inclusion on the boundary). In cases where the boundary has measure zero $(\nu \otimes \mu)(T) = 0$, one can implement $Z^*$ deterministically as an indicator of a measurable subset of the tie set; in cases where the boundary has non-zero measure $(\nu \otimes \mu)(T) > 0$, this corresponds to randomized tie-breaking.

Accordingly, complementary slackness yields, with $g_k(C) := \iint_{\mathcal{X} \times \mathcal{Y}} I w_k \, d\mu \, d\nu - (1 - \alpha) \geq 0$, it must hold that

$$\lambda_k^* g_k(C^*) = 0, \quad \forall k. \tag{18}$$

Thus: (i) if $\lambda_k^* > 0$ then $P^{(k)}(Y \in C^*(X)) = 1 - \alpha$, and (ii) if $\lambda_k^* = 0$ then $P^{(k)}(Y \in C^*(X)) \geq 1 - \alpha$.

We show next that at least one coordinate of $\lambda^*$ is strictly positive, i.e., statement (iii). Let $\rho := \nu \otimes \mu$ and recall that for each $k$,

$$w_k(x,y) := r_k(x)\, f_k(y \mid x) \geq 0$$

is integrable with respect to $\rho$ and satisfies $\iint w_k \, d\rho = 1$ since it is the joint density of $(X,Y)$ under $P^{(k)}$ with respect to $\rho$.

Notice that for the dual objective (13) with $h_\lambda(x,y) = \sum_k \lambda_k w_k(x,y)$, we have $\Phi(0) = 0$. Fix any $j \in \{1, \dots, K\}$ and consider $\lambda = te_j$ with $t > 0$, where $e_j$ is the $j$-th unit vector. Then $h_\lambda = tw_j$ and

$$\Phi(te_j) = (1 - \alpha)t - \iint (tw_j - 1)_+ \, d\rho.$$

For any $a \geq 0$ and $t > 0$, $(ta - 1)_+ \leq ta\,\mathbb{1}\{a \geq 1/t\}$. Applying this pointwise with $a = w_j(x,y)$ and integrating,

$$\iint (tw_j - 1)_+ \, d\rho \leq t \iint w_j \,\mathbb{1}\{w_j \geq 1/t\} \, d\rho.$$

Define $T_j(t) := \iint w_j \,\mathbb{1}\{w_j \geq 1/t\} \, d\rho$. Since $w_j$ is integrable, $T_j(t) \to 0$ as $t \downarrow 0$ (the tail of an integrable function vanishes). Therefore,

$$\iint (tw_j - 1)_+ \, d\rho \leq tT_j(t) = o(t),$$

and hence

$$\Phi(te_j) \geq t\left[(1 - \alpha) - T_j(t)\right].$$

Because $T_j(t) \to 0$ and $1 - \alpha > 0$, there exists $t_0 > 0$ such that for all $t \in (0, t_0)$,

$$\Phi(te_j) \geq t\frac{1-\alpha}{2} > 0.$$

Thus $\sup_{\lambda \geq 0} \Phi(\lambda) > 0$, so a dual maximizer cannot be $\lambda^* = 0$. Consequently, $\sum_k \lambda_k^* > 0$ and there exists at least one $k^*$ with $\lambda_{k^*}^* > 0$. By complementary slackness (18),

$$\hat{P}^{(k^*)}(Y \in C^*(X)) = 1 - \alpha.$$

This proves item (iii).

Finally, if $\mu(\{y : h_{\lambda^*}(x, y) = 1\}) = 0$ for $\nu$-almost every $x$, then the boundary set is $\mu$-null almost surely, making the optimizer unique up to $(\nu \otimes \mu)$-null sets. □

### B.3. Proof of Theorem 2

*Proof of Theorem 2.* Fix $x \in X$ and write $I(y) := \mathbb{1}\{y \in C(x)\}$. The conditional program is

$$\min_{I \in \{0,1\}} \quad \int I(y)\, d\mu(y)$$
$$\text{subject to} \quad \int I(y)f_k(y \mid x)\, d\mu(y) \geq 1 - \alpha, \quad \text{for } k = 1, \ldots, K.$$

Relax $I \in \{0, 1\}$ to $I \in [0, 1]$. Similar to the marginal problem, we can form the Lagrangian with multipliers $\lambda(x) = (\lambda_1(x), \ldots, \lambda_K(x)) \in \mathbb{R}_+^K$ (here $x$ is treated as fixed, yet we write the argument in $x$ for clarity):

$$\mathcal{L}_x(I, \lambda(x)) = \int I(y)\, d\mu(y) - \sum_k \lambda_k(x)\left(\int I(y)f_k(y \mid x)\, d\mu(y) - (1 - \alpha)\right).$$

Let $h_{\lambda(x)}(y) := \sum_k \lambda_k(x)f_k(y \mid x)$. Then

$$\mathcal{L}_x(I, \lambda(x)) = \int I(y)[1 - h_{\lambda(x)}(y)]\, d\mu(y) + (1 - \alpha)\sum_k \lambda_k(x).$$

For fixed $\lambda(x)$, minimization over $I \in [0, 1]$ is pointwise in $y$. Any minimizer has the threshold form

$$C_{\lambda(x)}(x) = \{y : h_{\lambda(x)}(y) > 1\} \cup S(x), \quad \text{with } S(x) \subseteq \{y : h_{\lambda(x)}(y) = 1\}. \tag{19}$$

The dual function is

$$\Phi_x(\lambda(x)) = (1 - \alpha)\sum_k \lambda_k(x) - \int \left(h_{\lambda(x)}(y) - 1\right)_+ d\mu(y),$$

this gives the conditional dual objective (5) mentioned in the theorem, and the dual problem is to maximize $\Phi_x$ over $\lambda(x) \in \mathbb{R}_+^K$.

Slater's condition holds (e.g., $C(x) \equiv \mathcal{Y}$ yields strict feasibility since $\alpha \in (0, 1)$), so strong duality applies and a dual maximizer $\lambda^*(x)$ exists. Thresholding $h_{\lambda^*(x)}$ yields a primal optimum $C^*(x)$. Complementary slackness gives, for each $k$,

$$\lambda_k^*(x)\left[\int \mathbb{1}\{y \in C^*(x)\}f_k(y \mid x)\, d\mu(y) - (1 - \alpha)\right] = 0. \tag{20}$$

Hence:

- If $\lambda_k^*(x) > 0$, then $P^{(k)}(Y_{n+1} \in C^*(x) \mid X_{n+1} = x) = 1 - \alpha$.

- If $\lambda_k^*(x) = 0$, then $P^{(k)}(Y_{n+1} \in C^*(x) \mid X_{n+1} = x) \geq 1 - \alpha$.

We now show that at least one coordinate of $\lambda^*(x)$ is strictly positive. Note that $\Phi_x(0) = 0$. Fix any $j \in \{1, \ldots, K\}$ and consider $\lambda(x) = te_j$ with $t > 0$, where $e_j$ is the $j$-th unit vector. Then $h_{\lambda(x)}(y) = tf_j(y \mid x)$ and

$$\Phi_x(te_j) = (1 - \alpha)t - \int (tf_j(y \mid x) - 1)_+ d\mu(y).$$

For any $a \geq 0$ and $t > 0$, $(ta - 1)_+ \leq ta \cdot \mathbb{1}\{ta - 1 \geq 0\} = ta \cdot \mathbb{1}\{a \geq 1/t\}$. Applying this pointwise with $a = f_j(y \mid x)$ and integrating,

$$\int (tf_j - 1)_+ d\mu \leq t \int f_j \mathbb{1}\{f_j \geq 1/t\} d\mu.$$

Because $f_j(\cdot \mid x)$ is a density (or probability mass function), $\int f_j d\mu = 1$. The set $\{f_j \geq 1/t\}$ shrinks to the empty set as $t \downarrow 0$, and $0 \leq f_j \mathbb{1}\{f_j \geq 1/t\} \leq f_j$. By dominated convergence,

$$T_j(t) := \int f_j \mathbb{1}\{f_j \geq 1/t\} d\mu \to 0 \text{ as } t \downarrow 0.$$

Therefore,

$$\Phi_x(te_j) \geq (1 - \alpha)t - tT_j(t) = t\left[(1 - \alpha) - T_j(t)\right].$$

Since $T_j(t) \to 0$ and $1 - \alpha > 0$, there exists $t_0 > 0$ such that for all $t \in (0, t_0)$,

$$\Phi_x(te_j) \geq t(1 - \alpha)/2 > 0.$$

Thus $\sup_{\lambda(x) \geq 0} \Phi_x(\lambda(x)) > 0$, so a dual maximizer cannot be $\lambda^*(x) = 0$. Consequently, $\sum_k \lambda_k^*(x) > 0$ and there exists some $k^*$ with $\lambda_{k^*}^*(x) > 0$. By complementary slackness (20),

$$P^{(k^*)}(Y_{n+1} \in C^*(x) \mid X_{n+1} = x) = 1 - \alpha.$$

Additionally, if $\mu(\{y : h_{\lambda^*(x)}(y) = 1\}) = 0$, then the boundary set is $\mu$-null, making the optimizer unique up to $\mu$-null sets. $\qquad\square$

## B.4. Proof of Theorem 3

We begin by introducing the notation employed throughout the proofs, as well as several auxiliary lemmas that will be relied upon in the main results. Proofs of the lemmas are deferred to Appendix B.4.2. We begin with some useful definitions.

**Definition 13** (Generalized quantile). *For $\alpha \in (0, 1)$, define the generalized $\alpha$-quantile set of a CDF $G$ as*

$$Q_\alpha(G) := \left\{q \in \mathbb{R} : G(q^-) \leq \alpha \leq G(q)\right\}. \tag{21}$$

*We term each $q \in Q_\alpha(G)$ as a generalized $\alpha$-quantile.*

**Definition 14** (Randomized quantile). *Given scores $W_1, \ldots, W_n \in \mathbb{R}$ and an auxiliary $U \sim \mathrm{Unif}(0, 1)$ independent of the data, define the randomized empirical CDF*

$$\hat{G}_U(t) := \frac{\#\{i : W_i < t\} + (1 + \#\{i : W_i = t\})U}{n + 1}.$$

*We define the randomized empirical $\alpha$-quantile of $\{W_1, \ldots, W_n\}$ as*

$$\hat{q}_\alpha := \inf\left\{t \in \mathbb{R} : \hat{G}_U(t) \geq \alpha\right\}. \tag{22}$$

**Lemma 15** (Quantile stability under uniform CDF convergence). *Let $G$ be a CDF on $\mathbb{R}$ and let $G_n$ be CDFs with $\sup_t |G_n(t) - G(t)| \to 0$ as $n \to \infty$. Fix $\alpha \in (0, 1)$, and let $Q_\alpha(G)$ denote the generalized $\alpha$-quantile set defined in Equation (21). Suppose $q_n \in \mathbb{R}$ satisfies $G_n(q_n-) \leq \alpha \leq G_n(q_n)$, then*

$$\mathrm{dist}(q_n, Q_\alpha(G)) := \inf_{q \in Q_\alpha(G)} |q_n - q| \to 0.$$

*In particular, if $Q_\alpha(G) = \{q^*\}$ (i.e., $G$ is continuous at the $\alpha$-quantile and there is no flat segment at level $\alpha$), then $q_n \to q^*$.*

**Lemma 16** (Quantile-set stability under CDF perturbations). *Let $F$ and $G$ be CDFs on $\mathbb{R}$. Suppose that for some $\varepsilon \geq 0$,*

$$F(t - \varepsilon) \leq G(t) \leq F(t + \varepsilon), \qquad \forall t \in \mathbb{R}.$$

*If $Q_\alpha(F) = [a, b]$ for some $\alpha \in (0, 1)$, then every $q \in Q_\alpha(G)$ satisfies $q \in [a - \varepsilon, b + \varepsilon]$. In particular,*

$$\sup_{q \in Q_\alpha(G)} \mathrm{dist}\left(q, Q_\alpha(F)\right) \leq \varepsilon.$$

B.4.1. PROOF OF THEOREM 3

*Proof of Theorem 3.* To clearly denote the asymptotic regime as $n \to \infty$, throughout this proof we add the superscript $(n)$ to the estimated quantities $\hat{\lambda}$, $\hat{f}_k$, and $\hat{h}$.

Since both $f_k$ and $\hat{\lambda}^{(n)}$ are bounded, and by assumption, $\sup_x \|\hat{\lambda}^{(n)}(x) - \lambda^*(x)\|_\infty \xrightarrow{p} 0$ and $\sup_{x,y} |\hat{f}_k^{(n)}(y|x) - f_k(y|x)| \xrightarrow{p} 0$, decompose

$$\sup_{x,y} \left| \hat{h}^{(n)}(x,y) - h^*(x,y) \right| \leq \sum_k \left[ \sup_x \left| \hat{\lambda}_k^{(n)}(x) - \lambda_k^*(x) \right| \cdot \sup_{x,y} f_k(y|x) \right.$$
$$\left. + \sup_x \left| \hat{\lambda}_k^{(n)}(x) \right| \cdot \sup_{x,y} \left| \hat{f}_k^{(n)}(y|x) - f_k(y|x) \right| \right].$$

Each term tends to 0 in probability, hence

$$\sup_{x,y} \left| \hat{h}^{(n)}(x,y) - h^*(x,y) \right| \xrightarrow{p} 0. \tag{23}$$

Fix $k$ and condition on $\hat{h}^{(n)}$. Let $W_{k,i} := \hat{h}^{(n)}(X_i^{(k)}, Y_i^{(k)})$ and $W_{k,\text{test}} := \hat{h}^{(n)}(x,y)$. Since $V = -\hat{h}$, the randomized $p$-value from Section 3.2 (equivalently, Equation (16)) is the randomized empirical CDF of $W$ at the test point:

$$p_k^{(n)}(x,y) = \frac{\#\{i : W_{k,i} < W_{k,\text{test}}\} + (1 + \#\{i : W_{k,i} = W_{k,\text{test}}\}) U_k}{n_k + 1}, \quad \text{with } U_k \sim \text{Unif}(0,1).$$

Thus, the single-source prediction set is based on thresholding $\hat{h}^{(n)}$:

$$\{y : p_k^{(n)}(x,y) \geq \alpha\} = \{y : \hat{h}^{(n)}(x,y) > \hat{q}_{k,\alpha}^{(n)}\},$$

where $\hat{q}_{k,\alpha}^{(n)}$ is the randomized empirical $\alpha$-quantile of $W_{k,i}$:

$$\hat{q}_{k,\alpha}^{(n)} := \inf \left\{ t \in \mathbb{R} : \frac{\#\{i : W_{k,i} < t\} + (1 + \#\{i : W_{k,i} = t\}) \cdot U_k}{n_k + 1} \geq \alpha \right\}.$$

Aggregating $K$ sources yields

$$\hat{C}^{(n)}(x) = \{y : \hat{h}^{(n)}(x,y) \geq \hat{q}_{\min,\alpha}^{(n)}\},$$

where $\hat{q}_{\min,\alpha}^{(n)} := \min_k \hat{q}_{k,\alpha}^{(n)}$.

Let $F_k(t)$ be the CDF of $h^*(X,Y)$ under $P^{(k)}$ and $F_k^{(n)}(t)$ the CDF of $\hat{h}^{(n)}(X,Y)$. Conditional on the training data, the calibration scores are i.i.d. from a distribution with CDF $F_k^{(n)}$. By the DKW inequality,

$$\sup_t \left| \hat{F}_k^{(n)}(t) - F_k^{(n)}(t) \right| \xrightarrow{p} 0.$$

Define

$$\varepsilon_n := \sup_{x,y} |\hat{h}^{(n)}(x,y) - h^*(x,y)|.$$

Then $\varepsilon_n \xrightarrow{p} 0$ by (23). Moreover, for any $t \in \mathbb{R}$ we have

$$F_k(t - \varepsilon_n) \leq F_k^{(n)}(t) \leq F_k(t + \varepsilon_n),$$

because $|\hat{h}^{(n)} - h^*| \leq \varepsilon_n$ implies

$$\{h^*(X,Y) \leq t - \varepsilon_n\} \subseteq \{\hat{h}^{(n)}(X,Y) \leq t\} \subseteq \{h^*(X,Y) \leq t + \varepsilon_n\}.$$

Since our randomized $p$-value inversion selects an empirical generalized $\alpha$-quantile $\hat{q}_{k,\alpha}^{(n)} \in Q_\alpha(\hat{F}_k^{(n)})$, applying Lemma 15 with $G_n = \hat{F}_k^{(n)}$ and $G = F_k^{(n)}$ yields

$$\text{dist}\left(\hat{q}_{k,\alpha}^{(n)}, Q_\alpha(F_k^{(n)})\right) \xrightarrow{p} 0.$$

Next, applying Lemma 16 with $F = F_k$, $G = F_k^{(n)}$ and $\varepsilon = \varepsilon_n$ gives

$$\text{dist}\left(Q_\alpha(F_k^{(n)}), Q_\alpha(F_k)\right) \leq \varepsilon_n \xrightarrow{p} 0.$$

By the triangle inequality,

$$\text{dist}(\hat{q}_{k,\alpha}^{(n)}, Q_\alpha(F_k)) \leq \text{dist}(\hat{q}_{k,\alpha}^{(n)}, Q_\alpha(F_k^{(n)})) + \text{dist}(Q_\alpha(F_k^{(n)}), Q_\alpha(F_k)) \xrightarrow{p} 0.$$

That is,

$$\text{dist}\left(\hat{q}_{k,\alpha}^{(n)}, Q_\alpha(F_k)\right) \xrightarrow{p} 0.$$

In particular, if 1 is the unique generalized $\alpha$-quantile of $F_k$, then $\hat{q}_{k,\alpha}^{(n)} \xrightarrow{p} 1$.

By KKT conditions and strong duality, we know the optimal rule thresholds at 1; that is, it includes all $(x,y)$ with $h^*(x,y) > 1$ and, at most, a randomized fraction of those with $h^*(x,y) = 1$. Under $P^{(k)}$, the maximal coverage achievable without lowering the threshold is

$$\mathbb{P}^{(k)}\left(h^*(X,Y) \geq 1\right) = 1 - F_k(1^-).$$

The coverage constraint $\mathbb{P}^{(k)}(y \in C^*) \geq 1 - \alpha$ therefore forces $1 - F_k(1^-) \geq 1 - \alpha$, i.e., $F_k(1^-) \leq \alpha$ for every $k$; otherwise the threshold-1 solution would be infeasible, contradicting strong duality.

By conclusion (iii) of Theorem 2, there exists at least one $k$ with $\lambda_k > 0$, such that $\alpha$ lies in the jump $[F_k(1^-), F_k(1)]$, hence the generalized $\alpha$-quantile is unique and equals 1. Consequently, for each $k$, any $\alpha$-quantile of $F_k$ is no smaller than 1, and for at least one $k$, it is exactly equal to 1. Therefore,

$$\hat{q}_{\min,\alpha}^{(n)} \xrightarrow{p} 1. \tag{24}$$

Let

$$\delta_n := \left|\hat{q}_{\min,\alpha}^{(n)} - 1\right| + \sup_{x,y}\left|\hat{h}^{(n)}(x,y) - h^*(x,y)\right|.$$

Then $\delta_n \xrightarrow{p} 0$ by Equations (24) and (23). Also note that,

- If $h^*(x,y) > 1 + 2\delta_n$, then $\hat{h}^{(n)}(x,y) > 1 + \delta_n \geq \hat{q}_{\min,\alpha}^{(n)}$, so $(x,y) \in \hat{C}^{(n)}$.

- If $h^*(x,y) < 1 - 2\delta_n$, then $\hat{h}^{(n)}(x,y) < 1 - \delta_n \leq \hat{q}_{\min,\alpha}^{(n)}$, so $(x,y) \notin \hat{C}^{(n)}$.

Hence,

$$\{(x,y) : h^*(x,y) > 1 + 2\delta_n\} \subseteq \hat{C}^{(n)} \subseteq \{(x,y) : h^*(x,y) \geq 1 - 2\delta_n\} \cup \hat{B}_n,$$

where

$$\hat{B}_n \subseteq \{(x,y) : |\hat{h}^{(n)}(x,y) - \hat{q}_{\min,\alpha}^{(n)}| = 0\} \subseteq \{(x,y) : |h^*(x,y) - 1| \leq \delta_n\},$$

which follows from Equation (23) derived earlier. Taking symmetric differences with $\{(x,y) : h^*(x,y) \geq 1\}$ and letting $n \to \infty$,

$$\limsup_n \rho\left(\hat{C}^{(n)} \triangle \{(x,y) : h^*(x,y) \geq 1\}\right) \leq \limsup_n \rho\left(\{(x,y) : |h^*(x,y) - 1| \leq 2\delta_n\}\right) \leq |T| \tag{25}$$

since $T = \{(x,y) : h^*(x,y) = 1\}$ and $|T| = \rho(T) = \int_{\mathcal{X}} \mu(T(x))d\nu(x)$, and the measure of shrinking neighborhoods of $T$ tends to $|T|$.

Finally, write

$$|\hat{C}^{(n)}| - |C^*| = \left(|\hat{C}^{(n)}| - |\{h^* \geq 1\}|\right) + \left(|\{h^* \geq 1\}| - |C^*|\right).$$

The first bracket is bounded in absolute value by $\rho(\hat{C}^{(n)} \triangle \{h^* \geq 1\}) \leq |T|$ from Inequality (25). The second bracket equals $|T| - |S^*| + |\{h^* > 1\}| - |\{h^* > 1\}| = |T| - |S^*|$, whose absolute value is $\leq |T|$. Therefore,

$$\limsup_{n \to \infty} \left| |\hat{C}^{(n)}| - |C^*| \right| \leq |T|.$$

Moreover, Inequality (25) shows there exists a subsequence $\{n_j\}$ and a measurable set $S_\infty \subseteq T := \{(x, y) : h^*(x, y) = 1\}$ such that

$$\rho\left(\hat{C}^{(n_j)} \Delta \left(\{h^* > 1\} \cup S_\infty\right)\right) \to 0.$$

Consequently, choosing the oracle set $C^*(x) = \{y : h^*(x, y) > 1\} \cup S_\infty(x)$ yields $|\hat{C}^{(n_j)}| \to |C^*|$. $\qquad\square$

### B.4.2. PROOF OF LEMMA 15

*Proof of Lemma 15.* For a monotone right-continuous $H$, denote the left limit by $H(x-) := \sup_{t < x} H(t)$. If $\sup_t |H_n(t) - H(t)| \leq \varepsilon$, then also $\sup_x |H_n(x-) - H(x-)| \leq \varepsilon$, because

$$H_n(x-) = \sup_{t < x} H_n(t) \geq \sup_{t < x}[H(t) - \varepsilon] = H(x-) - \varepsilon,$$
$$H_n(x-) = \sup_{t < x} H_n(t) \leq \sup_{t < x}[H(t) + \varepsilon] = H(x-) + \varepsilon.$$

Let $a := \inf\{t : G(t) \geq \alpha\}$ and $b := \sup\{t : G(t) \leq \alpha\}$; then $a \leq b$ and $Q_\alpha(G) = [a, b]$. Then

- For any $\delta > 0$, $G(a - \delta) < \alpha$. Define $\gamma_L(\delta) := \alpha - G(a - \delta) > 0$.

- For any $\delta > 0$, for all $x \geq b + \delta$ we have $G(x-) > \alpha$. Indeed, for any such $x$ pick $s$ with $b < s < x$; then $G(s) > \alpha$ by definition of $b$, so $G(x-) \geq G(s) > \alpha$. Hence define $\gamma_R(\delta) := G(b + \delta/2) - \alpha > 0$.

**Left bound.** Fix $\delta > 0$ and choose $n$ large so that $\sup_t |G_n(t) - G(t)| \leq \varepsilon_n$ with $\varepsilon_n < \gamma_L(\delta)/2$. If $q \leq a - \delta$ then

$$G_n(q) \leq G(q) + \varepsilon_n \leq G(a - \delta) + \varepsilon_n = \alpha - \gamma_L(\delta) + \varepsilon_n < \alpha,$$

contradicting the requirement $\alpha \leq G_n(q)$ for $q \in Q_\alpha(G_n)$. Therefore any $q \in Q_\alpha(G_n)$ must satisfy $q > a - \delta$.

**Right bound.** With the same $n$ and $\varepsilon_n$ and the $\gamma_R(\delta)$ defined above, if $q \geq b + \delta$ then

$$G_n(q-) \geq G(q-) - \varepsilon_n \geq (\alpha + \gamma_R(\delta)) - \varepsilon_n > \alpha,$$

contradicting the requirement $G_n(q-) \leq \alpha$ for $q \in Q_\alpha(G_n)$. Therefore any $q \in Q_\alpha(G_n)$ must satisfy $q < b + \delta$.

Therefore, for any fixed $\delta > 0$ and all sufficiently large $n$ we have

$$Q_\alpha(G_n) \subset (a - \delta, b + \delta).$$

In particular, our selected $q_n \in Q_\alpha(G_n)$ lies within $\delta$ of the closed set $[a, b] = Q_\alpha(G)$, so $\mathrm{dist}(q_n, Q_\alpha(G)) \leq \delta$. Because $\delta > 0$ was arbitrary, $\mathrm{dist}(q_n, Q_\alpha(G)) \to 0$.

For the unique-quantile case $Q_\alpha(G) = \{q^*\}$, the distance convergence implies $q_n \to q^*$. $\qquad\square$

### B.4.3. PROOF OF LEMMA 16

*Proof of Lemma 16.* Let $Q_\alpha(F) = [a, b]$ and assume that

$$F(t - \varepsilon) \leq G(t) \leq F(t + \varepsilon), \qquad \forall t \in \mathbb{R}.$$

Take any $q \in Q_\alpha(G)$, so that $G(q^-) \leq \alpha \leq G(q)$.

From $G(q) \leq F(q + \varepsilon)$ and $\alpha \leq G(q)$ we obtain

$$F(q + \varepsilon) \geq \alpha.$$

By the definition $a = \inf\{t : F(t) \geq \alpha\}$, this implies $q + \varepsilon \geq a$, hence $q \geq a - \varepsilon$.

For the upper bound, note that for every $s < q$,

$$F(s - \varepsilon) \leq G(s).$$

Taking the supremum over $s < q$ yields

$$F((q - \varepsilon)^-) = \sup_{u < q - \varepsilon} F(u) = \sup_{s < q} F(s - \varepsilon) \leq \sup_{s < q} G(s) = G(q^-) \leq \alpha.$$

Suppose, for the sake of contradiction, that $q > b + \varepsilon$. Then $q - \varepsilon > b$. By the definition

$$b = \sup\{t : F(t) \leq \alpha\},$$

for any $u > b$ we must have $F(u) > \alpha$, and by monotonicity and right-continuity of $F$ this implies $F(u^-) > \alpha$ for all $u > b$. In particular,

$$F((q - \varepsilon)^-) > \alpha,$$

which contradicts $F((q - \varepsilon)^-) \leq \alpha$ above. Hence $q - \varepsilon \leq b$, i.e., $q \leq b + \varepsilon$.

We have shown that every $q \in Q_\alpha(G)$ lies in $[a - \varepsilon, b + \varepsilon]$, and therefore

$$\sup_{q \in Q_\alpha(G)} \text{dist}(q, Q_\alpha(F)) \leq \varepsilon.$$

$\square$

## B.5. Optimality of the Integrated Dual Problem

In this part, we formalize the discussion at the beginning of Section 4.1 on the optimal $\lambda^*(x)$ as the solution to an integrated dual objective.

**Proposition 17** (Equivalence of integrated dual and conditional dual)**.** *For $k = 1, \ldots, K$, let $f_k(\cdot \mid x)$ be the conditional density/pmf of $Y \mid X = x$ with respect to $\mu$. Fix $\alpha \in (0, 1)$. For $\lambda \in \mathbb{R}_+^K$ and $x \in \mathcal{X}$, define $h_\lambda(x, y) := \sum_{k=1}^K \lambda_k f_k(y \mid x)$, and*

$$\varphi_x(\lambda) := (1 - \alpha) \sum_{k=1}^K \lambda_k - \int_{\mathcal{Y}} (h_\lambda(x, y) - 1)_+ \, d\mu(y).$$

*Let $\tilde{\nu}$ be a $\sigma$-finite measure on $(\mathcal{X}, \mathcal{A})$ with Radon–Nikodym density $w(x) := \frac{d\tilde{\nu}}{d\nu}$ satisfying $0 < w(x) < \infty$ for $\nu$-a.e. $x$. Consider the integrated dual objective*

$$\Phi_{\tilde{\nu}}(\lambda(\cdot)) := \int_{\mathcal{X}} \left[ (1 - \alpha) \sum_{k=1}^K \lambda_k(x) - \int_{\mathcal{Y}} (h_\lambda(x, y) - 1)_+ \, d\mu(y) \right] d\tilde{\nu}(x).$$

*Then $\Phi_{\tilde{\nu}}(\lambda(\cdot)) = \int_{\mathcal{X}} w(x) \varphi_x(\lambda(x)) \, d\nu(x)$, and*

*(i) A measurable $\lambda^*(\cdot)$ maximizes $\Phi_{\tilde{\nu}}$ if and only if*

$$\lambda^*(x) \in \arg\max_{\lambda \in \mathbb{R}_+^K} \varphi_x(\lambda) \qquad \text{for } \nu\text{-a.e. } x.$$

*Hence the set of maximizers is independent of the particular choice of $\tilde{\nu}$, as long as $d\tilde{\nu}/d\nu > 0$ $\nu$-a.e.*

*(ii) For $\nu$-a.e. $x$, any maximizer $\lambda^*(x)$ is a dual maximizer of the $x$-conditional problem* (4). *Thresholding $h_{\lambda^*}$ at level 1 gives the conditionally optimal set*

$$C^*(x) = \{y \in Y : h_{\lambda^*}(x,y) > 1\} \cup S(x), \quad \text{with } S(x) \subseteq \{y : h_{\lambda^*}(x,y) = 1\},$$

*as in Theorem* 2. *Thus the optimal score and set do not depend on $\tilde{\nu}$.*

*Proof of Proposition* 17. By the Radon–Nikodym theorem, $d\tilde{\nu} = w\, d\nu$ with $w > 0$ $\nu$-a.e. Substituting,

$$\Phi_{\tilde{\nu}}(\lambda(\cdot)) = \int_{\mathcal{X}} \left[ (1-\alpha) \sum_k \lambda_k(x) - \int_{\mathcal{Y}} (h_\lambda(x,y) - 1)_+ \, d\mu(y) \right] w(x)\, d\nu(x) = \int_{\mathcal{X}} w(x)\, \varphi_x(\lambda(x))\, d\nu(x).$$

For any measurable $\lambda(\cdot)$,

$$\Phi_{\tilde{\nu}}(\lambda(\cdot)) = \int_{\mathcal{X}} w(x)\, \varphi_x(\lambda(x))\, d\nu(x) \leq \int_{\mathcal{X}} w(x) \sup_{\lambda \geq 0} \varphi_x(\lambda)\, d\nu(x), \tag{26}$$

with equality iff $\varphi_x(\lambda(x)) = \sup_{\lambda \geq 0} \varphi_x(\lambda)$ for $\nu$-a.e. $x$. Note that $\hat{\lambda}(\cdot)$ with $\hat{\lambda}(x) \in \operatorname{argmax} \varphi_x(\cdot)$ exists and attains the upper bound, for this $\hat{\lambda}$,

$$\Phi_{\tilde{\nu}}(\hat{\lambda}) = \int_{\mathcal{X}} w(x)\varphi_x(\hat{\lambda}(x))\, d\nu(x) = \int_{\mathcal{X}} w(x) \sup_{\lambda \geq 0} \varphi_x(\lambda)\, d\nu(x),$$

which matches the upper bound in Equation (26) and is therefore optimal. Moreover, if a candidate $\lambda(\cdot)$ fails to maximize $\varphi_x$ at a set of $x$ with positive $\tilde{\nu}$-measure, replacing it by a pointwise maximizer on that set always strictly increases $\Phi_{\tilde{\nu}}$, proving the necessity. Because $w(x) > 0$, multiplying by $w(x)$ does not change the pointwise argmax sets, so the maximizers are independent of $\tilde{\nu}$.

By definition, $\varphi_x(\cdot)$ is the dual objective of the $x$-conditional problem (4). Therefore, a pointwise maximizer $\lambda^*(x)$ is a dual maximizer for (4). By KKT conditions for (4), thresholding $h_{\lambda^*}(x, \cdot)$ at 1 yields the conditionally optimal set stated above. Independence from $\tilde{\nu}$ follows from (i). $\qquad\square$

### B.6. Proof of Theorem 12

*Proof of Theorem* 12. First, following exactly the same conditions and proof in Jin et al. (2022, Theorem 1) applied to the loss function $\hat{\ell}(\cdot)$, we can show that $\|\hat{\lambda} - \bar{\lambda}^*\|_{L_2} = O_P\big( (\frac{\log n}{n})^{p/(2p+d)} \big)$ and $\|\hat{\lambda} - \bar{\lambda}^*\|_\infty = O_P\big( (\frac{\log n}{n})^{2p^2/(2p+d)^2} \big)$. Then, by triangle inequality and Assumption 10, we obtain the desired results. $\qquad\square$

## C. Experimental Details

### C.1. Simulation Details

#### C.1.1. Classification Data-Generating Processes

Here we detail the data-generating processes used in the classification simulations of Section 5.2. We simulate $C = 6$ classes. For source $k \in [K]$ and class $c \in [C]$, the conditional class probability is given by a multinomial model $f_k(y = c \mid x) \propto \exp\{\eta_{kc}(x)\}$ with $\eta_{kc}(x) = \xi_k(b_{kc} + \beta_{kc}^\top x) + \mathbb{1}\{c > 1\}\, g(x)$. Here, with a temperature parameter $\tau \in \mathbb{R}$, the linear signal is $\xi_k = 2.5(1 + 0.25\tau \cdot u_k)$ with $u_k \overset{\text{i.i.d.}}{\sim} \text{Unif}([-1,1])$, and the heterogeneous intercept is independently sampled as $b_{kc} \sim \mathcal{N}(0, (0.4\tau)^2)$. The source-specific linear coefficients are $\beta_{kc} = \bar{\beta}_c + \tau \cdot \Delta_{kc}$ where, after a random sample of $\mathcal{I} \subseteq [d]$ with $|\mathcal{I}| = 4$, we independently sample $(\bar{\beta}_c)_j \sim \mathcal{N}(0, 1)$ and $(\Delta_{kc})_j \sim \mathcal{N}(0, 0.15^2)$ for each $j \in \mathcal{I}$ and set $(\bar{\beta}_c)_j = (\Delta_{kc})_j = 0$ for $j \notin \mathcal{I}$. Finally, the nonlinear component $g(x)$ is set as zero in the LINEAR experiments, and we vary its definition in three DGPs in the NONLINEAR experiments: (a) interaction: $g(x) = 2 \sum_{(u,v)} w_{uv}\, x_u x_v$; (b) sinusoid: $g(x) = 2 \sum_{r=1}^3 a_r \sin(u_r^\top x + b_r)$, (c) softplus: $g(x) = 2 \sum_{r=1}^3 a_r \log\big(1 + \exp(u_r^\top x + b_r)\big)$. At the beginning of each experiment, we sample the weights $w_{uv}$, the linear coefficients $a_r$, $u_r$, and $b_r$ (the detailed processes are in Appendix C.2); afterwards, we draw the labeled and unlabeled data conditional on them. In the LINEAR and NONLINEAR experiments, the temperature parameter is fixed at $\tau = 2.5$. In the evaluation of TEMPERATURE experiments, we focus only on the linear model with $g(x) \equiv 0$ and vary the temperature $\tau \in \{0.5, 1.5, 2.5, 3.5, 4.5\}$.

### C.1.2. CLASSIFICATION IMPLEMENTATION DETAILS

This subsection details the implementation of MDCP and baselines in the classification simulations of Section 5.2. We implement the three competing methods based on the same estimators (built from the training folds) for fairness. We train a gradient boosting classifier $\hat{p}_k(y \mid x)$ to estimate $P^{(k)}(Y = y \mid X = x)$ for each source $k$, and a separate gradient boosting classifier on the pooled training data to estimate $p_{\text{pool}}(y \mid x)$. Following Section 4.2, we specify the per-source scores

$$s_k(X_i, Y_i) := -\hat{h}_{\hat{\lambda}}(X_i, Y_i), \quad \text{where} \quad \hat{h}_{\hat{\lambda}}(x, y) := \sum_{k=1}^K \hat{\lambda}_k(x)\hat{p}_k(y \mid x),$$

where, following the procedure in Section 4.2, we parameterize the nonnegative weight functions $\lambda_k(x)$ as spline functions, and learn $\hat{\lambda}_k(x)$ by minimizing the empirical objective (8). Specifically, we use a cubic B-spline basis with 3 polynomial degree and 5 knots placed uniformly over the range of the observed covariates, constructed using the SPLINETRANSFORMER in the SCIKIT-LEARN Python package. The multipliers $\hat{\lambda}_k(x)$ are trained on the same training fold based on the fitted classifiers $\hat{p}_k(y \mid x)$ and $\hat{p}_{\text{pool}}(y \mid x)$, i.e., we reuse the training data. In both BASELINE-SRC-$k$ and BASELINE-AGG, we use the widely-used TPS score (Sadinle et al., 2019) with the same fitted probabilities $\hat{p}_k(y \mid x)$ to build single-source and aggregated prediction sets, thereby serving as baselines with the same fitted models without optimizing for multi-distribution efficiency.

### C.1.3. REGRESSION DATA-GENERATING PROCESSES

Here we detail the data-generating processes used in the regression simulations of Section 5.3. In all regression settings, for source $k \in [K]$, we sample the labels via $Y = \mu_k(X) + \varepsilon_k$, with independent noise $\varepsilon_k \sim \mathcal{N}(0, \sigma_k^2)$. Following similar design ideas as in the classification settings, given a temperature parameter $\tau \in \mathbb{R}$, the regression function is $\mu_k(x) = \beta_k^\top x + b_k + g(x)$, where the source-specific coefficient is given by $\beta_k = \bar{\beta} + 0.2\tau \cdot \delta_k$, with $\bar{\beta}_j \sim \mathcal{N}(0, 1)$ and $(\delta_k)_j \sim \mathcal{N}(0, 1)$ independently drawn for $j \in \mathcal{I}$ and $\bar{\beta}_j \equiv 0$ and $(\delta_k)_j \equiv 0$ for $j \notin \mathcal{I}$, and $\mathcal{I}$ is the randomly drawn set of signals. The source-specific intercept is given by $b_k = b + \tau \cdot v_k$ with independently drawn $b \sim \mathcal{N}(0, 0.5^2)$ and $v_k \sim \mathcal{N}(0, 0.5^2)$. In each run, we randomly sample a signal-to-noise ratio from Unif([5, 10]), and achieve it by adjusting the noise variance $\sigma_k^2$. Finally, the nonlinear component $g(x)$ is set to be zero in the LINEAR experiments, and we consider the same three choices of $g(x)$ in the NONLINEAR experiments as in the classification settings (Section 5.2), with the same sampling process of the hyper-parameters. We fix $\tau = 2.5$ in LINEAR and NONLINEAR experiments. In TEMPERATURE setting, we focus only on the linear model and vary $\tau \in \{0.5, 1.5, 2.5, 3.5, 4.5\}$; in addition, we sample $u \sim \text{Unif}([\{1 - \tau/4\}_+, 1 + \tau/4])$ and multiply the SNR-calibrated $\sigma_k$ by $u$, so that the temperature also affect the noise level.

### C.1.4. REGRESSION IMPLEMENTATION DETAILS

This subsection details the implementation of MDCP and baselines in the regression simulations of Section 5.3. In the regression procedure, the optimal score function relies on the condition density $f_k(y \mid x)$ in each source $k$. As mentioned in Section 4.3, to avoid the challenging conditional density estimation, we model the data as $P^{(k)}_{Y \mid X=x} \sim \mathcal{N}(\mu_k(x), \sigma_k(x))$ for some functions $\mu_k(x) = \mathbb{E}^{(k)}[Y \mid X = x]$ and $\sigma_k^2(x) = \text{Var}^{(k)}(Y \mid X = x)$, and obtain their estimates $\hat{\mu}_k(\cdot)$ and $\hat{\sigma}_k(\cdot)$ on the training fold via gradient boosting decision trees. Plugging in the two estimates leads to the estimated per-source conditional densities $\hat{f}_k(y \mid x)$ using gradient boosting decision trees. In the same way, we fit a pooled Gaussian working model on pooled training data to obtain $(\hat{\mu}_{\text{pool}}(x), \hat{\sigma}_{\text{pool}}(x))$ for the conditional density estimate $\hat{p}_{\text{pool}}(y \mid x)$. The conformity score for MDCP is then given by $s_k(X_i, Y_i) := -\sum_{k=1}^K \hat{\lambda}_k(X_i)\hat{f}_k(Y_i|X_i)$, where we parameterize $\lambda(x) \in \mathbb{R}_+^K$ as the spline function as in Section 5.2 and learn $\hat{\lambda}_k(x)$ by minimizing the empirical objective (8) on the same training fold. Both baselines use the conformity score $V_k(x, y) = (y - \hat{\mu}_k(x))/\hat{\sigma}_k(x)$ with the same estimated functions as in MDCP, paralleling the method in (Lei et al., 2018).

### C.2. Hyperparameter Sampling

We detail the sampling process of the hyperparameters in the simulations in Section 5.3.

For the interaction family, we draw the weights i.i.d. from $w_{uv} \sim \mathcal{N}(0, 1.1^2)$ for $(u, v) \in \mathcal{I} \times \mathcal{I}$. For both the sinusoidal and softplus families, each unit $r = 1, 2, 3$ uses a projection vector $u_r \in \mathbb{R}^d$ constructed as follows: we first sample a support $S_r \subset \mathcal{I}$ of size 3 uniformly at random and draw a magnitude $M_r \sim \text{Unif}(0.375, 0.875)$, sample a random unit vector $d_r \in \mathbb{R}^3$ and define $u_r[S_r] = M_r d_r$ with $u_r[\mathcal{I} \setminus S_r] = 0$. The sinusoidal component samples $b_r \sim \text{Unif}(-\pi/3, \pi/3)$ and

$a_r \sim \text{Unif}(0.5, 1.5)$, independently across $r$. The softplus component utilizes the same construction for $u_r$ as above, but with $b_r \sim \text{Unif}(-0.5, 0.5)$ and $a_r \sim \text{Unif}(0.75, 2.0)$, again independently across $r$.

### C.3. Algorithm Instantiation

This subsection includes omitted implementation details for the classification and regression algorithms used in our experiments.

**Classification algorithm** In Section 5.2, we use nonparametric methods to estimate class probabilities (in our case, gradient-boosted trees), and calibrate their probabilistic outputs using stratified cross-validation with isotonic regression. We then optimize the spline-based weights $\lambda(\cdot)$ using minibatch updates. The optimization is implemented in PyTorch with automatic differentiation: we precompute spline features, data densities, source weights, and related terms, form the minibatch objective using these quantities and a trainable spline parameter matrix, and update the parameters with Adam. After each epoch, we evaluate the objective on the full training fold and apply early stopping to improve efficiency and mitigate overfitting.

**Regression algorithm** In Section 5.3, for each source $k$ we fit a heteroskedastic Gaussian plug-in model for the conditional density. We first learn a regression function $\hat{\mu}_k(x)$ using flexible nonparametric estimators (again, gradient-boosted trees). To model dispersion, we obtain out-of-fold predictions $\tilde{\mu}$ for the mean model via $K$-fold cross-validation: we partition the data into $K$ folds, fit the model on the other $K-1$ folds, and predict on the held-out fold. We then compute squared residuals $\hat{r}^2 = (Y - \tilde{\mu})^2$ and fit a second regressor to the logarithm of the residual squares using all $K$ folds. At prediction time we evaluate $\hat{\sigma}_k(x)$ from this variance model and form

$$\hat{f}_k(y \mid x) = \frac{1}{\sqrt{2\pi}\,\hat{\sigma}_k(x)} \exp\left( -\frac{1}{2}\left( \frac{y - \hat{\mu}_k(x)}{\hat{\sigma}_k(x)} \right)^2 \right).$$

As in classification, we treat the marginal density of $X$ as constant and use $\hat{f}_k(x,y) := \hat{f}_k(y \mid x)$ throughout.

**Optimization hyperparameters.** For both classification and regression, we learn the spline parameterization of $\lambda(\cdot)$ by optimizing the empirical dual objective using Adam with a fixed learning rate of $10^{-3}$ and minibatch size 256. We cap training at 10,000 epochs and shuffle minibatches each epoch. For early stopping, after each epoch we evaluate the *full-data* objective value $\hat{\Phi}^{(t)}$ on the entire training fold and stop when the relative change between consecutive epochs satisfies

$$\frac{\left| \hat{\Phi}^{(t)} - \hat{\Phi}^{(t-1)} \right|}{\max\left\{ \left| \hat{\Phi}^{(t-1)} \right|, 10^{-8} \right\}} < 10^{-4}.$$

When $\left| \hat{\Phi}^{(t-1)} \right|$ is very small, this corresponds to an absolute tolerance of approximately $10^{-12}$. Unless otherwise specified, we use PyTorch's default Adam hyperparameters $(\beta_1, \beta_2, \epsilon) = (0.9, 0.999, 10^{-8})$ and no weight decay. For numerical stability, we clip $\hat{p}_{\text{data}}(y_i \mid x_i)$ below at $10^{-8}$.

### C.4. Grid Search Algorithm and Guarantee

Let $\mathcal{D}_{\text{train}}$ and $\mathcal{D}_{\text{calib}}$ be the training and calibration data pooled across all $K$ sources. Define

$$y_L := \min\{Y_i : (X_i, Y_i) \in \mathcal{D}_{\text{train}} \cup \mathcal{D}_{\text{calib}}\},$$
$$y_U := \max\{Y_i : (X_i, Y_i) \in \mathcal{D}_{\text{train}} \cup \mathcal{D}_{\text{calib}}\}.$$

Fix an integer $M \geq 2$ (e.g., $M = 100$). Define a uniform grid on $[y_L, y_U]$ with $\Delta := \frac{y_U - y_L}{M-1}$:

$$y^{(j)} := y_L + j\,\Delta, \quad \text{for } j = 0, 1, \ldots, M-1. \tag{27}$$

We call $y^{(j)}$ and $y^{(j+1)}$ adjacent grid points. A subset $B$ of indices is called *consecutive* if it contains no gaps; equivalently, $B$ can be written as $\{a, a+1, \ldots, b\}$ for some integers $a \leq b$. For example, $\{3, 4, 5\}$ is consecutive, while $\{3, 5\}$ is not. For a test covariate $x$, we include a candidate $y$-grid point $y^{(j)}$ if the aggregated MDCP $p$-value

$$p(y^{(j)}) := \max_k p^{(k)}(x, y^{(j)}) \geq \alpha,$$

---

**Algorithm 2** Grid-Search Algorithm (Regression)

---

**Input:** Number of sources $K$, pooled calibration data $\mathcal{D} = \cup_{k=1}^{K} \mathcal{D}^{(k)}$, test input $x$, grid endpoints $y_L, y_U$, grid size $M$, grid spacing $\Delta$, significance level $\alpha$.

1: // GRID CONSTRUCTION
2: Construct grid points $y^{(j)}$, $j = 0, \ldots, M-1$ over $[y_L, y_U]$ as in (27).
3: // EVALUATE AGGREGATED $p$-VALUES ON THE GRID
4: **for** $j = 0$ **to** $M-1$ **do**
5:    Compute $p(y^{(j)}) = \max_k p^{(k)}(x, y^{(j)})$.
6: **end for**
7: Collect included grid points, form $J$ following (28).
8: // MERGE INCLUDED GRID POINTS INTO BLOCKS
9: Decompose $J$ into maximal consecutive blocks $B_r = \{j_{r,L}, \ldots, j_{r,R}\}$ for $r = 1, \ldots, R$.
10: // EXTEND EACH BLOCK BY ONE GRID SPACING
11: **for** each block $B_r$ **do**
12:    Create interval $I_r := [y^{(j_{r,L})} - \Delta, \, y^{(j_{r,R})} + \Delta]$.
13: **end for**
14: Take the union of all intervals: $C_{\mathrm{grid}}(x) := \bigcup_r I_r$.
**Output:** Regression prediction set $C_{\mathrm{grid}}(x)$ for test input $x$.

---

where each $p^{(k)}$ is derived using formula (16). Let

$$J := \{j \in \{0, \ldots, M-1\} : p(y^{(j)}) \geq \alpha\} \tag{28}$$

be the set of included grid indices. We decompose $J$ into consecutive blocks $B_r = \{j_{r,L}, \ldots, j_{r,R}\}$ for $r = 1, \ldots, R$. We say a decomposition is *maximal* if the decomposed blocks are consecutive, disjoint, and cannot be enlarged by adding adjacent indices from $J$.

Let $C_{\mathrm{MDCP}}(x)$ denote the (exact) MDCP prediction set we want to construct with score

$$s(x, y) = -\sum_{k=1}^{K} \lambda_k(x; \hat{\Theta}) \, \hat{f}_k(y \mid x)$$

and randomized $p$-values given by (16). Let $C_{\mathrm{grid}}(x)$ denote the conformal set constructed from the grid-search Algorithm 2.

**Assumption 18** (Grid resolution). *For the fixed test covariate $x$, let $C := C_{\mathrm{MDCP}}(x) \cap [y_L, y_U]$. Every connected component of $C$ intersects the grid $\{y^{(j)}\}_{j=0}^{M-1}$; that is, for each connected component $[\ell, r] \subseteq C$ there exists some $j \in \{0, \ldots, M-1\}$ such that $y^{(j)} \in [\ell, r]$.*

**Remark 19** (Sufficient condition for Assumption 18). *A simple sufficient condition for Assumption 18 is that every connected component $[\ell, r] \subseteq C$ has Lebesgue length at least $\Delta$ (the grid spacing). In that case $[\ell, r]$ cannot lie strictly inside a single open cell $(y^{(j)}, y^{(j+1)})$ of length $\Delta$, so it must contain at least one grid point $y^{(j)}$.*

**Proposition 20** (Superset on the grid range). *Let $C := C_{\mathrm{MDCP}}(x) \cap [y_L, y_U]$ and suppose Assumption 18 holds. Then*

$$C \subseteq C_{\mathrm{grid}}(x) \subseteq [y_L - \Delta, \, y_U + \Delta].$$

*In particular,*

$$C \subseteq C_{\mathrm{grid}}(x) \cap [y_L - \Delta, \, y_U + \Delta],$$

*so within the observed $y$-range the grid merge-and-extend procedure never excludes any MDCP-accepted value and may only enlarge the set.*

*Proof of Proposition 20.* Under Assumption 18, fix a connected component $[\ell, r] \subseteq C$ with $[\ell, r] \cap \{y^{(j)}\}_{j=0}^{M-1} \neq \varnothing$. Let

$$j_1 := \min\{j : y^{(j)} \in [\ell, r]\}, \qquad j_2 := \max\{j : y^{(j)} \in [\ell, r]\}.$$

Since $y^{(j_1)}, y^{(j_2)} \in [\ell, r] \subseteq C$, by definition of $C_{\mathrm{MDCP}}(x)$ we have $p(y^{(j_1)}) \geq \alpha$ and $p(y^{(j_2)}) \geq \alpha$, so $j_1, j_2 \in J$ and all indices $j \in [j_1, j_2]$ belong to the same consecutive block $B_r$.

By the grid spacing, $y^{(j_1-1)} = y^{(j_1)} - \Delta$ (if $j_1 > 0$), and $y^{(j_2+1)} = y^{(j_2)} + \Delta$ (if $j_2 < M-1$).

(i). Because $j_1$ is the first grid index inside $[\ell, r]$, we have $y^{(j_1-1)} < \ell \leq y^{(j_1)}$, hence $\ell \geq y^{(j_1)} - \Delta$.

(ii). Because $j_2$ is the last grid index inside $[\ell, r]$, we have $y^{(j_2)} \leq r < y^{(j_2+1)}$, hence $r \leq y^{(j_2)} + \Delta$.

Therefore $[\ell, r] \subseteq [y^{(j_1)} - \Delta, y^{(j_2)} + \Delta] = I_r$, where $I_r$ is one of the intervals produced by Algorithm 2. Taking the union over all connected components of $C$ yields

$$C \subseteq \bigcup_r I_r = C_{\text{grid}}(x).$$

Finally, by construction $I_r \subseteq [y_L - \Delta, y_U + \Delta]$ for every $r$, so $C_{\text{grid}}(x) \subseteq [y_L - \Delta, y_U + \Delta]$ and the stated inclusions follow. $\qquad\square$

## C.5. Real-Data Application Details

### C.5.1. FUNCTIONAL CATEGORY OF SATELLITE IMAGE UNDER SUBPOPULATION SHIFT

Satellite ML has been widely used to detect functional land uses, allocate resources, and inform risk analyses. A key challenge here is the geographic heterogeneity and acquisition variability. Here, we use MDCP to protect subpopulation shift between data-rich locations to data-poor regions due to different materials, urban morphologies, and imaging conditions.

We leverage the 2016 time slice in the Functional Map of the World (FMoW) dataset (Christie et al., 2018) with over one million images from 249 countries/regions, and the label is one of 62 functional classes. We focus on uniform coverage across regions in Africa, the Americas, Asia, Europe, Oceania and *Other*. We treat each geographic region as a source. Let $X$ denote the image input and $Y$ the functional class label. In this context, uniform coverage ensures reliability under arbitrary changes in the composition of regions.

The data contains 140,459 samples in total. We allocate 37.5% as the model training fold $|\mathcal{D}_{\text{pre-train}}| = 52,531$. The training distribution is highly imbalanced, with 30.27% from Europe and 38.72% from the Americas, yet only 2.23% from Oceania and 0.05% from Other. Using the DENSENET-121 backbone (Huang et al., 2018) initialized with IMAGENET weights (Deng et al., 2009), we compute the penultimate representation $e(x)$ and fit a pooled probabilistic classifier $\hat{p}_{\text{pool}}(y \mid x)$ together with region-specific classifiers $\{\hat{p}_k(y \mid x)\}_{k=1}^K$ on top of $e(x)$. The models are trained on $\mathcal{D}_{\text{pre-train}}$ and these probability estimates are used by both the TPS baselines and MDCP. More specifically, after training the DENSENET-121 backbone on the pre-training split, we use its penultimate feature representation $e(x)$ for all subsequent conformal procedures. During the training, we fit a pooled multiclass probabilistic classifier on the model training fold $\mathcal{D}_{\text{pre-train}}$ and, in parallel, source-specific classifier per geographic region on the corresponding region-specific model training fold. Each classifier is trained via cross-entropy and yields estimated class probabilities, denoted by $\hat{p}_{\text{data}}(y \mid x)$ for the pooled model and $\hat{p}_k(y \mid x)$ for the $k$-th region. To improve estimation quality under imbalance, we let Europe and the Americas retain separate heads, while Africa, Asia, Oceania, and *Other* share one head during training, which stabilizes the conditional density estimation. These estimates are used to compute APS scores for the baselines and to form the MDCP score (Section 4.2) through $\hat{h}(x,y) = \sum_{k=1}^K \hat{\lambda}_k(x)\hat{p}_k(y \mid x)$, where $\hat{\lambda}(\cdot)$ is learned from the *auxiliary* training data. When fitting $\hat{\lambda}(\cdot)$, we apply PCA to $e(x)$ on the *auxiliary* training data and use the leading components as the input features.

Next, we perform 100 random partitions of the remaining data into auxiliary train (12.5%), calibration (37.5%), and test (50%) splits. Before fitting $\lambda(x)$, we apply PCA to the feature vectors $e(x)$ on the auxiliary training split and retain the first 16 components. We parameterize $\lambda(x) \in \mathbb{R}_+^K$ by a feedforward neural network containing two hidden layers of width 4 with ReLU activations and a $K$-dimensional output. We fit $\lambda(x)$ by optimizing the empirical dual objective in Section 4.2 on the auxiliary training split, then calibrate MDCP on the calibration split and evaluate on the test split. Baselines use the same fitted conditional models with TPS scores, calibrated on the 37.5% calibration split and evaluated on the 50% test split. The nominal coverage is set at $1 - \alpha = 0.9$, and the results are reported in Figure 8.

Due to nontrivial heterogeneity across regions, we observe unequal coverage for single-source baselines. Standard conformal prediction sets calibrated using data from the *Other* region achieve overall coverage above 0.9, yet still suffer from undercoverage in the worst-case. Notably, the baseline calibrated on data-rich regions (e.g., Europe and the Americas) exhibits worse worst-case coverage (despite the lower variability in coverage across runs). A possible explanation is that the abundant data allow the model to be well trained, producing prediction sets that are tightly tuned to those specific source distributions but perform poorly in others. In contrast, for data-scarce regions such as *Other*, the model performs poorly even in the original source, and thus must output wide sets. Consequently, models calibrated on scarce-data regions can yield better worst-case coverage than those calibrated on rich-data regions, albeit at the cost of larger prediction sets. In

comparison, MDCP remains valid across all sources, with near-tight worst-case coverage. Moreover, BASELINE-AGG admits any signal deemed useful by any source and is conservative. MDCP mitigates this issue by joint training across sources. Indeed, its set size is even smaller than single-source prediction sets, showing the significant benefit of efficiency optimization.

In Appendix D.1.2, we further examine the penalty-tuning approach similar to the simulations. In this task, we again observe negligible difference from standard MDCP, showing the stability of our procedure.

### C.5.2. POVERTY PREDICTION UNDER URBAN-RURAL SHIFT

Household surveys for mapping economic well-being are infrequent or missing especially in regions where nationally representative surveys are limited by local resources (Blumenstock et al., 2015). In these scenarios, satellite imagery offers a scalable proxy: a practical strategy is to learn from countries with the desired economic label then transfer to countries with images only (Abelson et al., 2014). In this part, we visit the subset of a modified release of the (Yeh et al., 2020) poverty-mapping dataset from 2014 to 2016 to show the application of MDCP to provide reliable uncertainty quantification across rural and urban areas. In this data, the features are the satellite image, and the label is a continuously-valued wealth index. Figure 9a visualizes the label density in the urban and rural areas which exhibits strong heterogeneity.

The dataset contains $n = 7,535$ samples, with 2,664 from urban areas and 4,871 from rural, each treated as a source. We reserve 37.5% of the data for training and fit a shared 8-channel RESNET-18 backbone (He et al., 2015) with random initialization. On top of the shared RESNET-18 representation $e(x)$, we fit both pooled and source-specific working models that output $(\hat{\mu}(x), \hat{\sigma}(x))$ as in Section 5.3 and hence a conditional density $\hat{f}(y \mid x) = \mathcal{N}(y; \hat{\mu}(x), \hat{\sigma}^2(x))$. Specifically, we train an 8-channel RESNET-18 backbone on the designated training split and denote its penultimate feature representation by $e(x)$. During the training, we fit (i) a pooled heteroskedastic Gaussian model on the auxiliary training split and (ii) two source-specific Gaussian models for Urban and Rural on their corresponding auxiliary training subsets. Each model outputs functions $\hat{\mu}(x) \in \mathbb{R}$ and $\hat{\sigma}(x) > 0$ (we use softplus as a monotone link to enforce positivity) and defines the conditional density $\hat{f}(y \mid x) = \mathcal{N}(y; \hat{\mu}(x), \hat{\sigma}^2(x))$. The models are trained by minimizing the Gaussian negative log-likelihood: for an observation $(x_i, y_i)$,

$$-\log \hat{f}(y_i \mid x_i) = \log \hat{\sigma}(x_i) + \frac{(y_i - \hat{\mu}(x_i))^2}{2\hat{\sigma}^2(x_i)} + \frac{1}{2}\log(2\pi).$$

The fitted source-specific densities $\{\hat{f}_k\}$ and the pooled density $\hat{f}_{\text{data}}$ are then used in both the regression baselines and MDCP via the score $\hat{h}(x, y) = \sum_{k=1}^K \hat{\lambda}_k(x)\hat{f}_k(y \mid x)$ and the max-$p$ aggregation (Sections 4.3). This yields two source models $\hat{f}^{\text{Rural}}(y \mid x)$ and $\hat{f}^{\text{Urban}}(y \mid x)$, as well as a pooled model $\hat{p}_{\text{pool}}(y \mid x)$. Next, we perform 100 random splits of the remaining data into auxiliary train (12.5%), calibration (37.5%), and test (50%) folds. As in FMoW, before fitting $\lambda(x)$ we apply PCA to $e(x)$ and keep the first 16 components. We use the same neural-network parameterization for $\lambda(x)$ as in FMoW, fit it on the auxiliary training fold, calibrate the MDCP set on the calibration fold, and evaluate on the test fold.

As shown in Figure 9b, single-source models calibrated with single-source data fail to achieve valid coverage on the other domain. We see from Figure 9a that the rural distribution is more skewed; under strong heterogeneity, despite the larger sample size from the rural source, single-source calibration still produces short intervals and low coverage. On the other hand, BASELINE-AGG, which naively combines single-source prediction sets, is overly conservative. MDCP maintains tight worst-case coverage with significant efficiency gains, striking a good balance in coverage allocation across sources.

Finally, we find that the penalty-tuning extension of MDCP still offers no clear advantage over MDCP. See Appendix D.1.2 for further discussion.

### C.5.3. MEDICAL SERVICES UTILIZATION ACROSS SENSITIVE GROUPS

Our last application revisits the Medical Expenditure Panel Survey (MEPS) dataset used in (Romano et al., 2019a), including Panels 19-21 (Agency for Healthcare Research and Quality, 2017; 2018; 2019), to address equalized coverage even without observing the sensitive group label. The dataset contains detailed individual-level information on demographics and health care utilization. The features include age, marital status, race, poverty level, and health status and insurance related covariates. The label is a continuously-valued medical service utilization score.

We follow the same pre-processing steps as (Romano et al., 2019a) with one-hot encoding of categorical variables. The feature dimension for $X$ is 139, consistent across panels. We apply a log transformation to the label due to its skewedness;

without this step, the estimated variance would be excessively inflated which drastically degrades the efficiency of single-source baselines. As reported by the (Romano et al., 2019b), predictive distributions vary across the sensitive attribute *race*: a neural-network predictor tends to predict higher utilization for non-White than for White individuals. Motivated by this finding, we treat *race* as the source label, assigning $k = 0$ to non-White and $k = 1$ to White, with sample sizes $n_0 = 9640$ and $n_1 = 6016$.

We split the data into training (60%), calibration (20%), and test (20%) folds. For both MDCP and the baselines, we follow the same modeling procedure as in Section 5.3: conditional densities are modeled as $P_{Y|X=x}^{(k)} \sim \mathcal{N}(\mu_k(x), \sigma_k(x)^2)$, with $\hat{\mu}_k(x)$ and $\hat{\sigma}_k(x)$ estimated via gradient-boosting decision trees trained on the source-specific training fold; in addition, we fit a pooled model on the union of the training data using the same approach as in Section 5.3. MDCP further fits $\lambda(x)$ using the same training data and calibrates prediction sets on the entire calibration fold. In contrast, the single-source baselines (NON-WHITE only and WHITE only) calibrate solely on their respective source-specific calibration fold. The BASELINE-AGG combines the two single-source calibrated sets. Finally, the three methods are all evaluated on the same test fold. The above protocol is applied independently to each panel, with results reported separately.

Figure 10 reports the performance of the competing methods. The single-source baseline trained and calibrated exclusively on the non-white group exhibits systematic undercoverage (both on average and worst-case) across panels. This is because the white group is more right-skewed and the single-source baseline from the non-white group fails to cover its heavy tail. On the other hand, single-source sets trained and calibrated exclusively on the White group approximately attains worst-case coverage, yet the width of the prediction sets is exceedingly high. We conjecture that this may be due to the unreliable estimation of the working models with the skewed data, since the models are not trained to optimize efficiency in the downstream conformal prediction set. Similarly, BASELINE-AGG is overly conservative and has wide prediction sets. Finally, MDCP achieves tight worst-case coverage, showing the role of approximate complementary slackness. Efficiency optimization lets MDCP achieve even shorter sets than the single-source baselines.

Finally, in Appendix D.1.2, we find that in this dataset, the penalty-tuning extension of MDCP again yields similar performance as MDCP, showing the robustness of the current implementation.

# D. Additional Experiments

## D.1. Ablation Study on Optimization Stability

For the ablation study, we examine the difficulty of optimizing the dual objective (7) and assess the stability and reliability of the optimization procedure. The motivating idea is to test whether off-the-shelf optimization via PYTORCH may lead to large fitted coefficients due to instability. To this end, we introduce the following penalty terms to encourage stability, recall in (9), $\lambda_j(x) = \text{softplus}\left(\Lambda(x)^\top \theta_j\right)$ for $j \in [K]$, where $\Lambda(x) \in \mathbb{R}^m$ is a vector of spline basis functions and $\theta_j \in \mathbb{R}^m$ are trainable coefficients:

$$\hat{\mathbb{E}}_{\text{train}}\left[\frac{(1 - h_\lambda)_-}{\hat{p}_{\text{data}}}\right] + (1 - \alpha)\hat{\mathbb{E}}_{\text{train}}\left[\sum_k \lambda_k\right] - \gamma \underbrace{\left(\hat{\mathbb{E}}_{\text{train}}\left[\sum_k \lambda_k^2\right] + \sum_k \|D\theta_k\|^2\right)}_{\text{Penalty}},$$

where $D$ is the second order difference operator: $(D\theta_k)_i = \theta_{k,i} - 2\theta_{k,i+1} + \theta_{k,i+2}$, with $i = 1, \ldots, m - 2$, which serves as a discrete analogue of penalizing the curvature of the underlying function $\lambda_k(\cdot)$, and $\theta_{k,i}$ is the $i$-th parameter in the spline feature space.

We select the hyperparameter $\gamma$ over the grid $[0.0, 0.001, 0.01, 0.1, 1.0, 10.0, 100.0, 1000.0]$. To use the data efficiently, we split the training data into a *mimic calibration set* and a *mimic test set*. For each individual run, we calibrate the method on the mimic calibration set for every candidate $\gamma$, evaluate performance on the mimic test set, and choose the $\gamma$ that yields the smallest average set size on this mimic test set. Denote this selected value by $\gamma^*$. We then fix $\gamma^*$ and run MDCP with the original calibration and test data. Since the calibration and test data are not involved in this optimization process, the uniform coverage guarantee of MDCP still follows. Moreover, we expect the selected hyperparameter to perform at least as well as, and potentially better than, the non-penalized version (i.e., $\gamma = 0$) in terms of the chosen efficiency criterion. We compare the results from the penalized MDCP with data-driven $\gamma^*$ side by side with the non-penalized version ($\gamma = 0$). We evaluate this approach across all the simulation studies and real data applications.

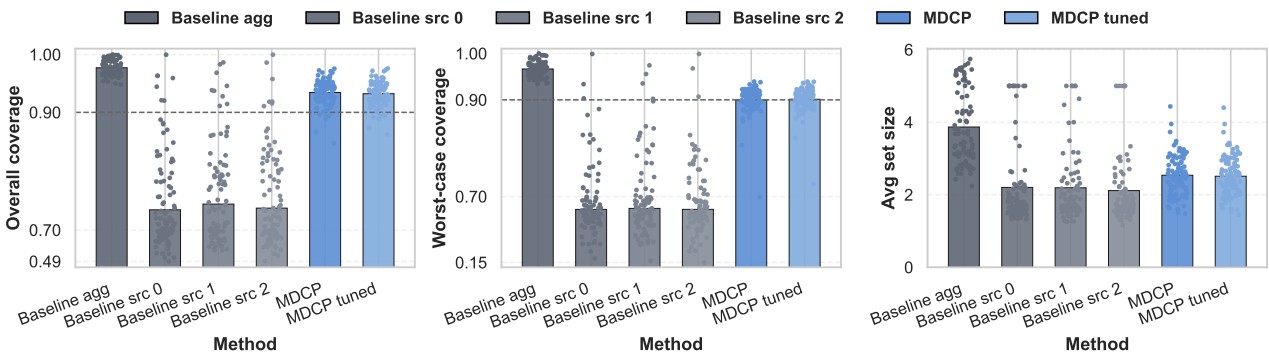

*Figure 11.* Evaluation on the classification LINEAR suites, where MDCP with data-driven $\gamma^*$ is labeled as "MDCP tuned". All other experimental settings are identical to those in Figure 2.

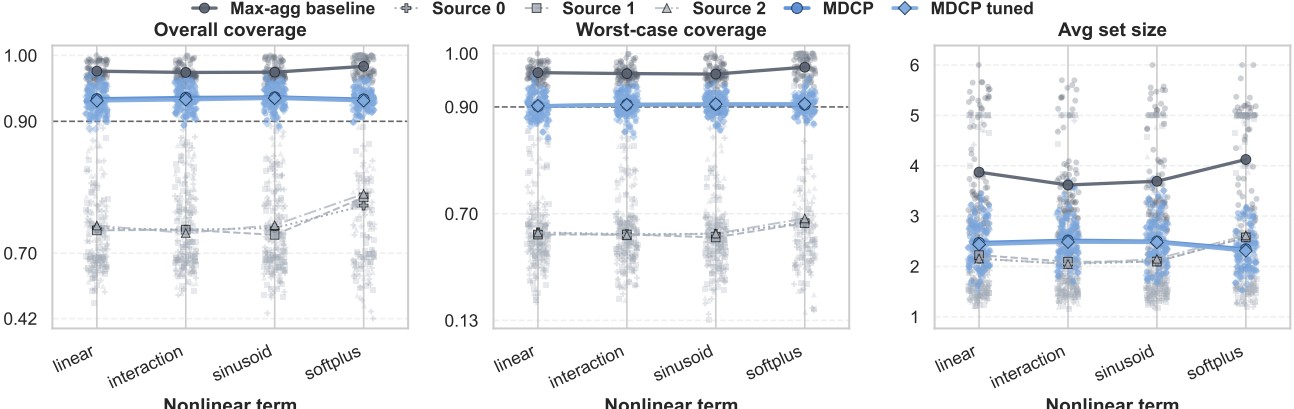

*Figure 12.* Evaluation on the classification NONLINEAR suites. Experimental settings are identical to Figure 3. The differences between vanilla MDCP and tuned MDCP are small across all nonlinear term settings.

### D.1.1. SIMULATION RESULTS

For the classification simulations, using the setup in Section 5.2, we evaluate performance on the three suites from Section 5.1: LINEAR (Figure 11), NONLINEAR (Figure 12), and TEMPERATURE (Figure 13). After the initial training step, we split the training data into equal-sized mimic calibration and mimic test sets (50%/50%) and apply the parameter-selection procedure described above. Across all three suites, tuning the penalty parameter $\gamma$ produces at most negligible gains in set efficiency. This suggests that the MDCP optimization step is already stable and no additional penalty is required in most of the simulation settings.

In the regression simulations, under the same setup as Section 5.3, we examine performance on the three suites defined in Section 5.1: LINEAR (Figure 14), NONLINEAR (Figure 15), and TEMPERATURE (Figure 16). Analogous to the classification experiments, once the model has been trained, we divide the training data evenly into a mimic calibration set and a mimic test set, and subsequently perform the parameter selection procedure described above. Across all three suites, data-driven tuning of the penalty parameter $\gamma$ produces, at best, marginal improvements in set efficiency. This finding indicates that the baseline MDCP optimization procedure is already sufficiently robust, and that, in most simulated scenarios, the dual optimization problem can be solved reliably without introducing an additional penalty term.

### D.1.2. REAL DATA RESULTS

We also assess the impact of $\gamma$ on the real-world datasets. Following Section 6, we repeat the procedures for the FMoW, PovertyMap, and MEPS datasets, now including $\gamma$ as an additional tuning parameter. The candidate values match those used above, ranging from 0.001 to 1000, with $\gamma = 0$ corresponding to the non-penalized version. For all three datasets, the training set is split 50%/50% into mimic calibration and mimic test subsets. For the FMoW and PovertyMap experiments, we use $\gamma$ to control an $\ell_2$ penalty on the magnitude of the learned weight functions, of the form $\gamma \, \hat{\mathbb{E}}_{\text{train}}[\sum_k \lambda_k(X)^2]$. In

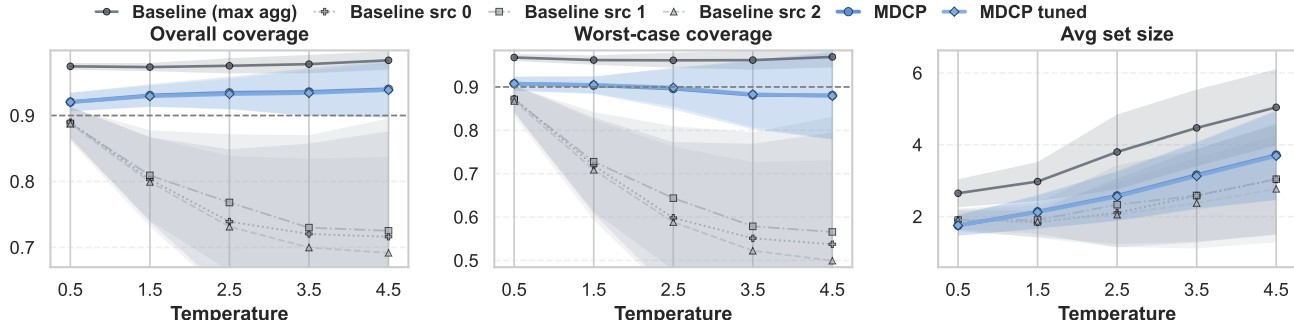

*Figure 13.* Evaluation on the classification TEMPERATURE suites. Experimental settings are identical to Figure 4. Vanilla MDCP and tuned MDCP exhibit only minor differences across all temperature parameter settings.

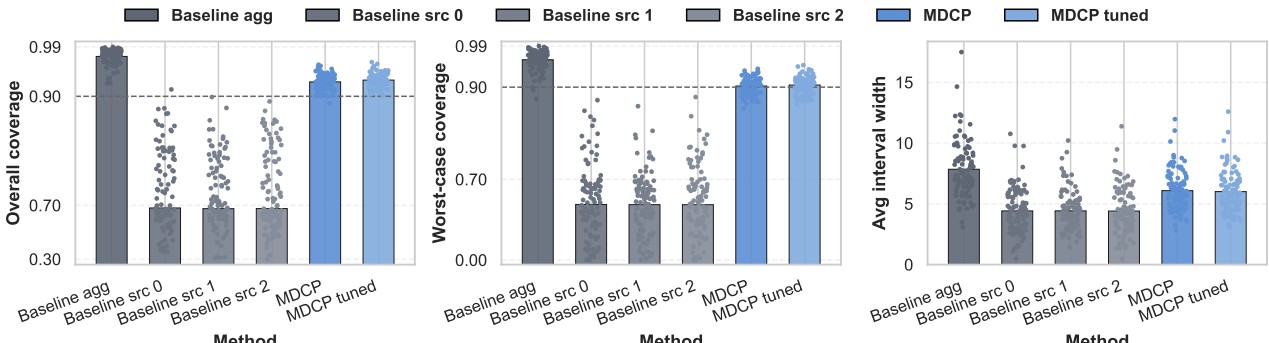

*Figure 14.* Results on the regression LINEAR suites, where MDCP with the selected penalty strength parameter $\gamma^*$ appears as "MDCP tuned". All other experimental settings match those in Figure 5.

particular, this tuning affects only the estimation of $\lambda(x)$; all subsequent calibration and evaluation steps remain unchanged. For the FMoW dataset (Figure 17), the original MDCP procedure is already stable, and introducing the penalty term yields little to no improvement. For the PovertyMap dataset (Figure 18), introducing $\gamma$ does not improve overall efficiency but does increase variability in the results. We attribute this to the $\gamma$-selection procedure: the chosen value is optimal for the *mimic calibration* and *mimic test* sets (Appendix D.1), but not necessarily for the true calibration and test sets. For the MEPS dataset (Figure 19), the low-density regions of the highly skewed target distribution are particularly challenging for baseline methods using score functions similar to Lei et al. (2018). In this setting, the penalty term still influences the behavior of the $\lambda_k$, helping prevent them from growing excessively large in low-density areas, but the resulting performance gains are modest. MDCP nonetheless maintains stable behavior while focusing more effectively on the higher-density and more practically relevant regions.

These results show that the mimic-split strategy can yield performance gains in cases where density estimation or optimization is particularly difficult, while remaining simple to implement with a 50%/50% calibration–test split. However, in most settings the vanilla MDCP procedure is already sufficiently robust, and the tuned MDCP variant offers little to no additional benefit.

### D.2. Effect of Randomization in $p$-Value

To assess the impact of randomly including points in the tie set $T(x)$, or equivalently, introducing a tie-breaking variable $U_k$ in the conformal $p$-value $p_k$ as described in Section 3.2, we run additional experiments with $U_k = 0$ and $U_k = 1$ during calibration of MDCP, and compare them with MDCP that uses randomized $p$-values. Across the LINEAR, TEMPERATURE, and NONLINEAR suites, for both classification and regression, we observe negligible differences between using randomized $U_k$ and fixing $U_k = 0$ or $U_k = 1$. The efficiency of MDCP with randomized $U_k$ lies between the two extremes, as expected: $U_k = 0$ excludes all points in the tie set, while $U_k = 1$ includes all of them. In Figures 21, 24, 22, and 25, the three curves are indistinguishable. But we do observe slight under-coverage in the worst case when setting $U_k = 0$, for example, on the LINEAR suite in Figures 20 and 23.

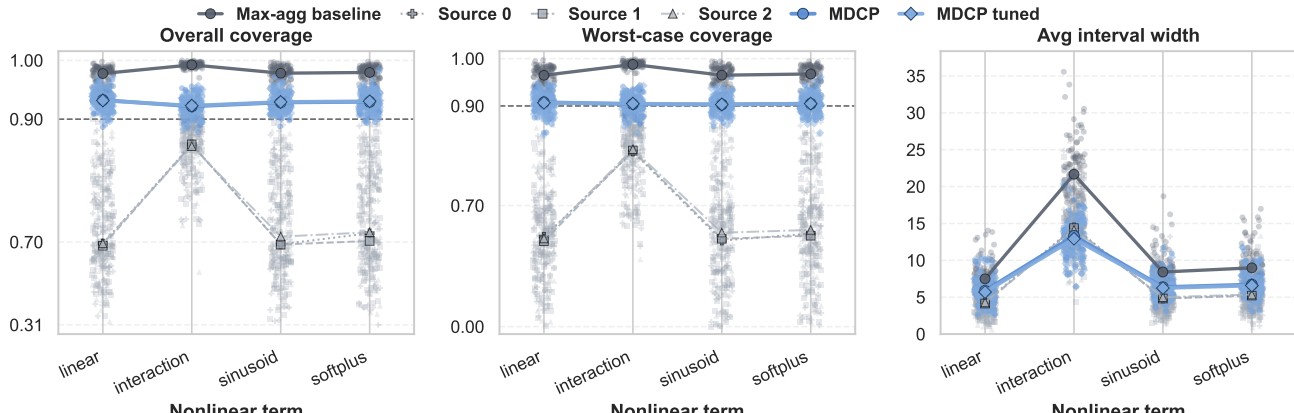

*Figure 15.* Results on the regression NONLINEAR suites Experimental settings match those in Figure 6. Across all choices of the nonlinear term, MDCP and tuned MDCP behave very similarly.

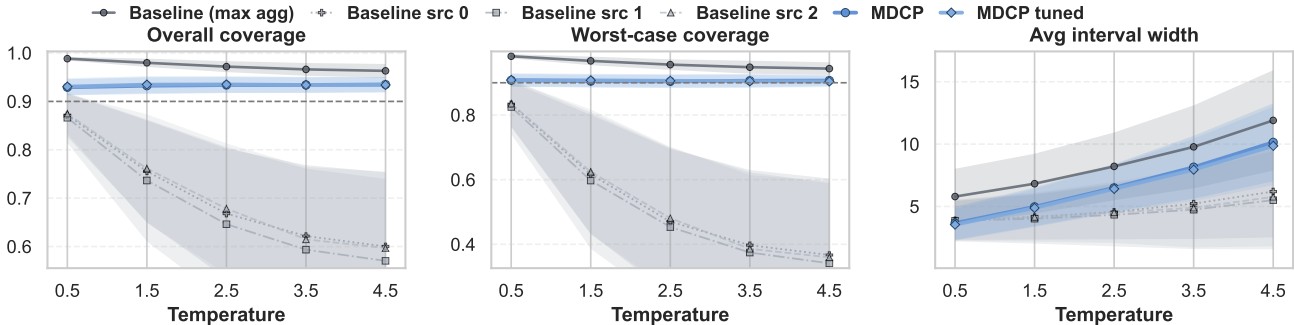

*Figure 16.* Results on the regression TEMPERATURE suites. Experimental settings match those in Figure 7. For all temperature parameter values, the gap between MDCP and tuned MDCP is negligible.

### D.3. Additional Simulations in Covariate Shift Settings

To evaluate MDCP in regimes where covariate shift contributes to the source heterogeneity, we introduce two additional suites of simulation settings:

(1). COVARIATE-SHIFT: In this suite, $P_X^{(k)}$ differs across sources but $P(Y \mid X)$ is shared.

(2). COVARIATE-AND-CONCEPT-SHIFT: In this suite, both $P_X^{(k)}$ and $P(Y \mid X)$ vary across sources.

These experiments follow the common protocol of Section 5.1: we consider $K = 3$ sources, feature dimension $d = 10$, and nominal miscoverage level $\alpha = 0.1$. For each source $k \in \{1, 2, 3\}$, we generate $n_k = 2000$ labeled samples. The pooled data are then randomly split into training (37.5%), calibration (12.5%), and test (50%) folds. For each suite, to focus on the effect of covariate shift, we fix the temperature parameter at $\tau = 2.5$, exclude nonlinear terms in both classification and regression settings, and sweep the covariate-shift magnitude parameter $\delta_X$ over $\delta_X \in \{0, 0.5, 1.5, 2.5, 3.5, 4.5\}$. For each configuration, we repeat the experiments for 100 independent trials. In each setting, we evaluate the following competing methods:

(i). BASELINE-SRC-$k$: The standard conformal prediction set $\hat{C}_{\text{src-}k}$ with calibration data from source $k$.

(ii). BASELINE-AGG: A simple max-$p$ aggregation of per-source prediction sets $\hat{C}_{\text{max-p}} := \cup_{k=1}^K \hat{C}_{\text{src-}k}$. This is the baseline without efficiency-oriented score learning.

(iii). MDCP: Our method in Algorithm 1.

(iv). MDCP-TUNED: The tuned variant of our method in Algorithm 1, employing a spline approximation for $\lambda$ and tuned penalty-term parameters, as detailed in Appendix D.1.

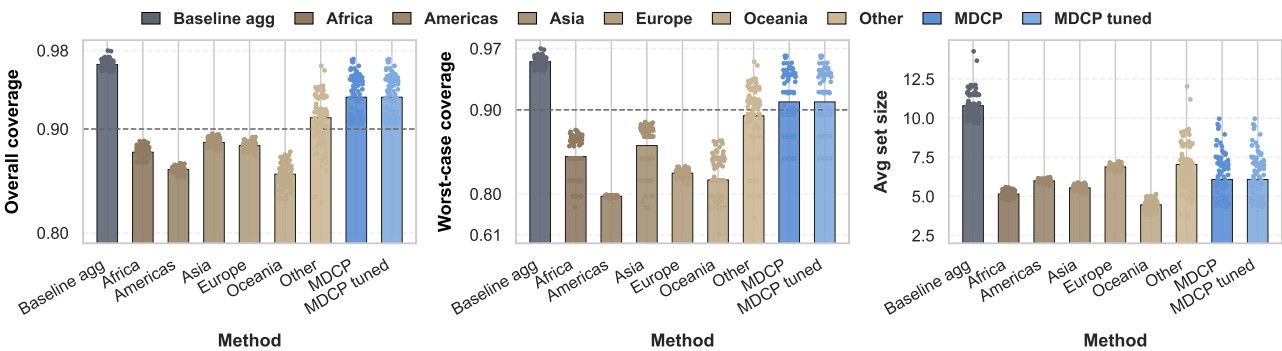

*Figure 17.* Results on the FMoW data, using the algorithmic procedure described in Section C.5.1. MDCP and tuned MDCP produce closely aligned performance in this case.

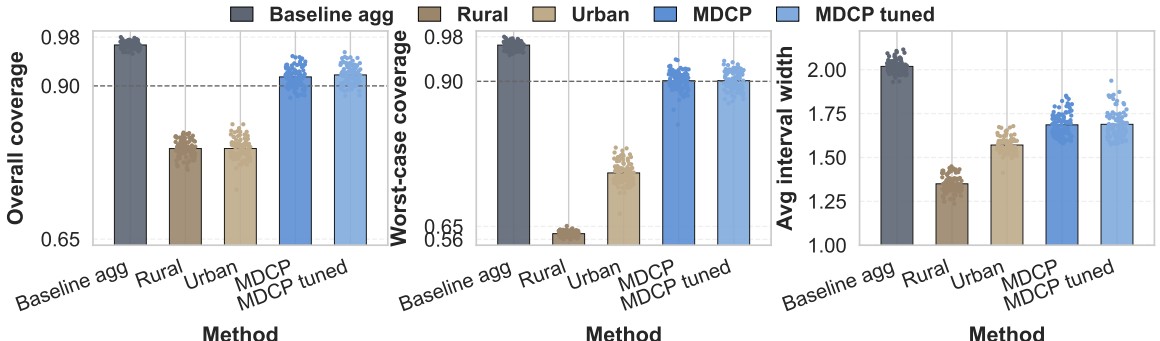

*Figure 18.* Results on the PovertyMap data, using the algorithmic procedure described in Section C.5.2. Introducing the parameter $\gamma$ increases the variability of efficiency.

As in Section 5.1, the single-source baseline is standard conformal prediction sets with the widely-used APS score (Romano et al., 2020) in classification and the variance-adaptive score of (Lei et al., 2018) in regression problems.

During each randomized individual run, we first sample an informative index set $I \subset \{1, \ldots, d\}$ uniformly at random with $|I| = 4$. We then construct a shared covariance matrix $\Sigma \in \mathbb{R}^{d \times d}$ for all sources using the equicorrelated form $\Sigma_{ij} = 0.2 + 0.8 \, \mathbb{1}\{i = j\}$. Next, we sample a shift direction $v \in \mathbb{R}^d$ supported on the informative coordinates: we draw $v_I \sim \mathcal{N}(0, I_{|I|})$, set $v_{I^c} = 0$, and normalize $v$. For a given shift magnitude $\delta_X$, we define the source-specific Gaussian means $\mu_1 = 0$, $\mu_2 = +\delta_X \, v$, $\mu_3 = -\delta_X \, v$, and generate covariates i.i.d. as

$$X_i^{(k)} \sim \mathcal{N}(\mu_k, \Sigma), \qquad i = 1, \ldots, n_k, \;\; k \in \{1, 2, 3\}.$$

Importantly, compared to the setup in Section 5.1, we do not standardize $X$ after sampling, so the mean shifts remain present in the observed covariates.

### D.3.1. ADDITIONAL SIMULATIONS IN CLASSIFICATION SETTINGS

**Data generating processes.** For classification, the label space is $\mathcal{Y} = \{1, 2, 3, 4, 5, 6\}$, with a total of $C = 6$ classes. We first draw class-specific base slopes $\{\bar{\beta}_c\}_{c=1}^C \subset \mathbb{R}^d$ supported on $I$, with $(\bar{\beta}_c)_j \sim \mathcal{N}(0, 1)$ for $j \in I$ and $(\bar{\beta}_c)_j = 0$ for $j \notin I$. We then generate $Y \mid X$ using a multinomial logit model with a fixed temperature $\tau = 2.5$.

In the COVARIATE-SHIFT suite, the conditional distribution $P(Y \mid X)$ is shared across sources. We draw shared intercepts $\bar{b}_c \sim \mathcal{N}(0, (0.4\tau)^2)$ and set $\xi_k \equiv \tau$, $\beta_{kc} \equiv \bar{\beta}_c$, and $b_{kc} \equiv \bar{b}_c$ for all sources $k$. Given a covariate value $x$, we compute logits $\eta_c(x) = \tau (\bar{b}_c + \bar{\beta}_c^\top x)$, and sample $Y$ from $\mathbb{P}(Y = c \mid X = x) = \frac{\exp\{\eta_c(x)\}}{\sum_{c'=1}^C \exp\{\eta_{c'}(x)\}}$, where $c \in [C]$. In the COVARIATE-AND-CONCEPT-SHIFT suite, we follow the linear concept-shift mechanism with fixed $\tau = 2.5$, while retaining the mean-shifted covariates described above. Specifically, for each source $k$, we draw $u_k \overset{\text{iid}}{\sim} \text{Unif}([-1, 1])$ and set $\xi_k = \tau (1 + 0.25\tau u_k)$. We also draw source-specific intercepts $b_{kc} \sim \mathcal{N}(0, (0.4\tau)^2)$ and perturb the slopes via $\beta_{kc} = \bar{\beta}_c + \tau \Delta_{kc}$, where

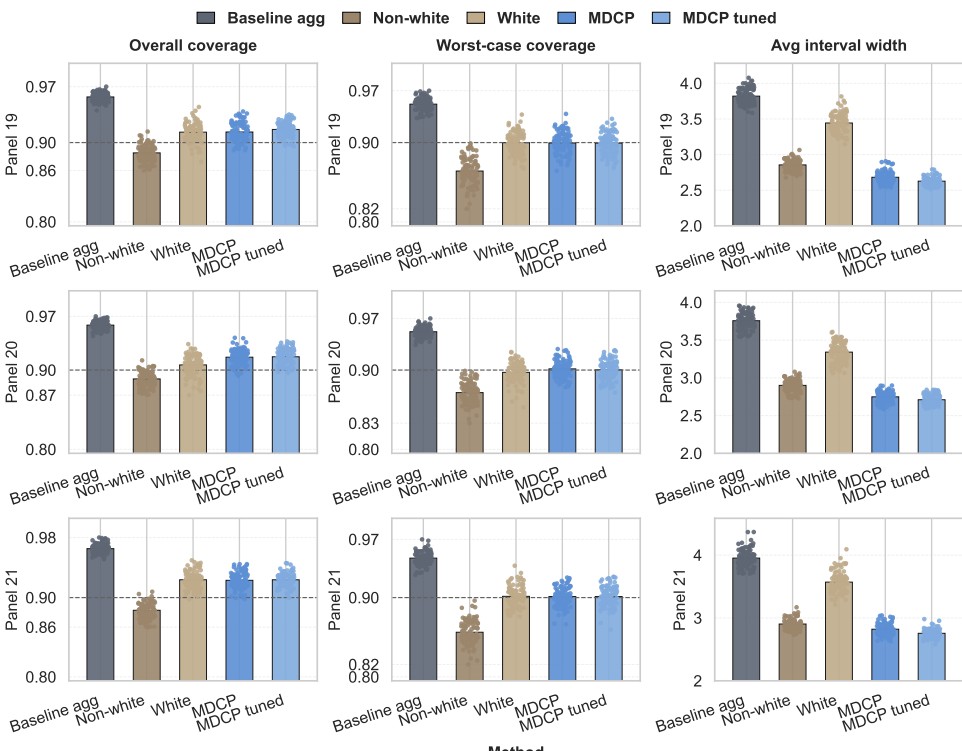

*Figure 19.* Results on the MEPS data, using the algorithmic procedure described in Section C.5.3. Introducing $\gamma$ provides a modest improvement to tuned MDCP, offering slightly better efficiency on this highly skewed dataset.

$(\Delta_{kc})_j \sim \mathcal{N}(0, 0.15^2)$ for $j \in I$ and $(\Delta_{kc})_j = 0$ otherwise. Given $x$ from source $k$, we compute $\eta_{kc}(x) = \xi_k(b_{kc} + \beta_{kc}^\top x)$ and sample $Y$ according to the multinomial probabilities $\mathbb{P}\left(Y = c \mid X = x, \text{ source} = k\right) = \frac{\exp\{\eta_{kc}(x)\}}{\sum_{c'=1}^C \exp\{\eta_{kc'}(x)\}}$, where $c \in [C]$.

**Method implementations.** The first three methods are implemented as in Section 5.2, while the MDCP-TUNED variant additionally uses the hyperparameter $\gamma$ for the penalty in the $\lambda$-optimization objective and follows the definitions and tuning procedures in Appendix D.1.

**Simulation results.** Figure 26 presents the results of the covariate-shift simulation in the classification setting. As the heterogeneity induced by the covariates across sources increases with the parameter $\delta_X$, MDCP maintains tight worst-case coverage, whereas the BASELINE-AGG method conservatively drives both the overall and worst-case coverage metrics to 1.0. MDCP also exhibits a more stable trajectory for the average set size (i.e., a smaller slope) compared with the baseline methods, while BASELINE-AGG shows a more rapid increase in average set size and a larger standard deviation at the same time.

Figure 27 presents the simulation results under the classification setting when both covariate shift and concept shift are present. MDCP remains robust, achieving tight worst-case coverage on average. It also maintains stability across different values of $\delta_X$, yielding a relatively flat average set size curve, whereas BASELINE-AGG degrades gradually as $\delta_X$ increases.

### D.3.2. ADDITIONAL SIMULATIONS IN REGRESSION SETTINGS

**Data generating processes.** For regression, we use a linear Gaussian model $Y = \beta^\top X + b + \varepsilon, \ \varepsilon \sim \mathcal{N}(0, \sigma^2)$. In each run, we randomly sample a signal-to-noise ratio from $\mathrm{Unif}([5, 10])$, and achieve it by adjusting the noise variance $\sigma_k^2$.

In the COVARIATE-SHIFT suite, $P(Y \mid X)$ is shared across sources and the noise level is also shared. We draw a shared slope vector $\bar{\beta} \in \mathbb{R}^d$ with $\bar{\beta}_j \sim \mathcal{N}(0, 1)$ for $j \in I$ and $\bar{\beta}_j = 0$ otherwise, and a shared intercept $\bar{b} \sim \mathcal{N}(0, 0.5^2)$. To enforce common noise, we compute $\sigma^2$ once using the realized covariates from source 1, and reuse this $\sigma^2$ for all sources. We then

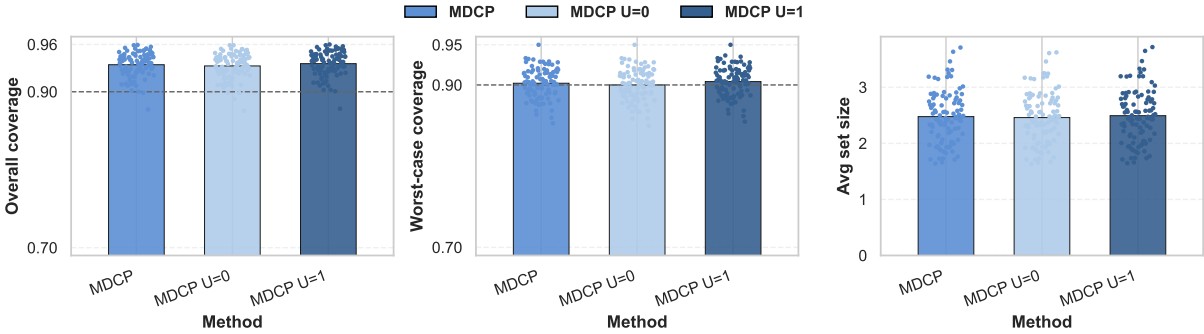

*Figure 20.* Performance of MDCP with randomized and deterministic $p$-values over the classification LINEAR suites; setting $U = 0$ excludes scores in the tie set, while $U = 1$ includes all points whose scores fall within the tie set.

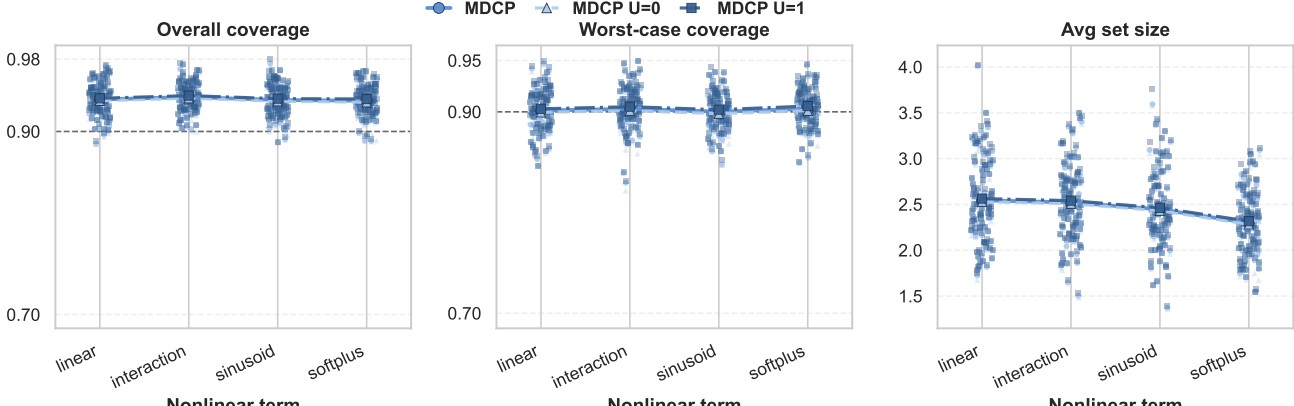

*Figure 21.* Performance of MDCP with randomized and deterministic $p$-values over the classification NONLINEAR suite. Lines show means over 100 runs, dots represent results of each run. Panels use the same settings as Figure 20.

generate, for each source $k$, $Y_i^{(k)} = \bar{\beta}^\top X_i^{(k)} + \bar{b} + \varepsilon_i^{(k)}$, where $\varepsilon_i^{(k)} \overset{\text{iid}}{\sim} \mathcal{N}(0, \sigma^2)$. In the COVARIATE-AND-CONCEPT-SHIFT suite, we introduce source-specific regression functions and calibrate noise per source. We draw a base slope $\bar{\beta} \in \mathbb{R}^d$ supported on $I$ same as the COVARIATE-SHIFT suite, and a base intercept $b \sim \mathcal{N}(0, 0.5^2)$. For each source $k$, we sample $\delta_k \in \mathbb{R}^d$ supported on $I$ with $(\delta_k)_j \sim \mathcal{N}(0, 1)$ for $j \in I$ and 0 otherwise, and set $\beta_k = \bar{\beta} + 0.2\tau\,\delta_k$, $b_k = b + \tau v_k$, $v_k \sim \mathcal{N}(0, 0.5^2)$. We then compute $\sigma_k^2$, and sample $Y_i^{(k)} = \beta_k^\top X_i^{(k)} + b_k + \varepsilon_i^{(k)}$, where $\varepsilon_i^{(k)} \overset{\text{iid}}{\sim} \mathcal{N}(0, \sigma_k^2)$.

**Method implementations.** The first three methods are implemented exactly as in Section 5.3. The MDCP-TUNED variant further introduces a hyperparameter $\gamma$ to control the penalty in the $\lambda$-optimization objective and adheres to the definitions and tuning procedures specified in Appendix D.1.

**Simulation results.** The evaluation results for the COVARIATE-SHIFT suite under the regression setting are shown in Figure 28. MDCP achieves tight worst-case coverage, while the performance of individual sources steadily degrades as $\delta_X$ increases, as expected. The average interval width of MDCP consistently stays below that of BASELINE-AGG, although the MDCP curve gradually increases and tends to be get closer with BASELINE-AGG. This behavior is also expected: as $\delta_X$, which approximately controls the separation among sources, continues to grow, in the extreme case where sources become perfectly separated, the optimal strategy is to optimize the per-source interval widths, and the efficiency gains from leveraging information across multiple sources become minimal.

The evaluation results for the COVARIATE-AND-CONCEPT-SHIFT suite under the regression setting are shown in Figure 29. Again, MDCP achieves tight worst-case coverage, while both the overall and worst-case coverage of the per-source-calibrated methods BASELINE-SRC-$k$ continually degrade. As explained earlier, the MDCP curve also tends to approach that of BASELINE-AGG as the shift magnitude increases.

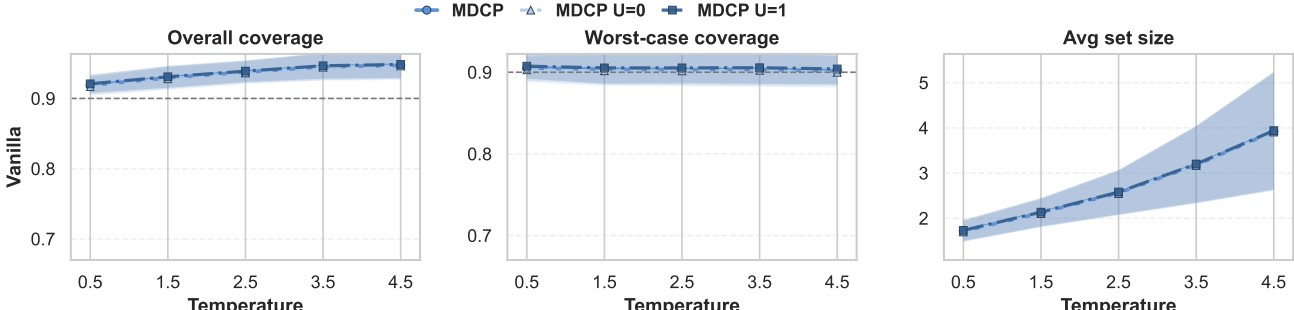

*Figure 22.* Performance of MDCP with randomized and deterministic $p$-values over the classification TEMPERATURE suite. Lines show means over 100 runs, with shaded $\pm 1$ standard deviation. Panels use the same settings as Figure 20.

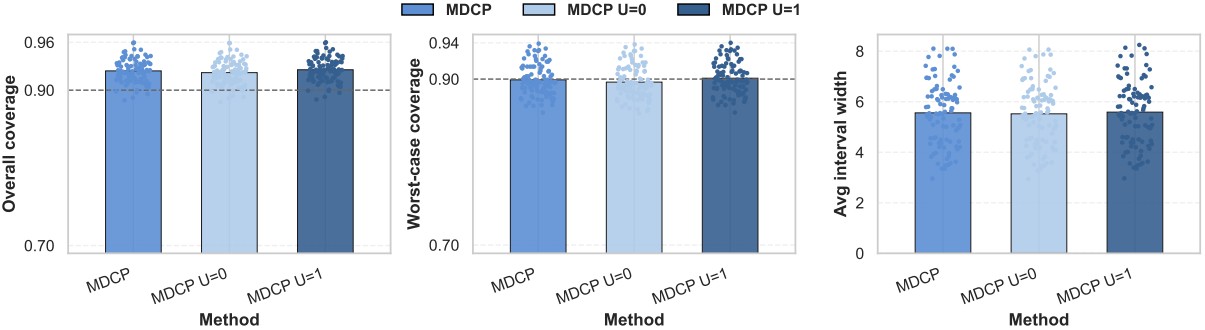

*Figure 23.* Performance of MDCP with randomized and deterministic $p$-values over the regression LINEAR suites; details are the same as in Figure 20.

### D.4. Runtime Breakdown and Analyses

To assess the time efficiency of our proposed algorithm, we decompose runtime across the LINEAR, NONLINEAR, and TEMPERATURE suites and further investigate computational bottlenecks. By decomposing both the MDCP pipeline and that of BASELINE-AGG (Figure 30), we find that the main bottleneck in MDCP is computing the weights $\lambda$. Since both methods share the same per-source conditional density estimation, the *Fit sources* cost is identical, while MDCP additionally requires fitting a conditional density on the pooled data (shown in light gray). The difference in calibration and evaluation time is negligible. Notably, by comparing the classification suites (Figure 30, 31, 32), and regression suites (Figure 33, 34, 35), MDCP is much slower for classification than for regression, and also slower than BASELINE-AGG. Empirically, we find that classification often requires many more iteration steps to solve the optimization problem.

### D.5. Scaling Behavior with Imbalance Ratio, Sample Size, and Number of Sources

In this subsection, we present additional empirical results to address the following questions: (1) How robust is MDCP when the sample size is small, and how does its efficiency change as the sample size increases? (2) Is MDCP sensitive to the number of sources? (3) Can MDCP be reliably deployed in scenarios where the sources are highly imbalanced?

#### D.5.1. SCALING STUDIES IN CLASSIFICATION SETTINGS

**Data generating processes.** As below, we vary three parameters of interest. All other aspects remain the same as in Section 5.2. We first vary the number of sources, $K \in \{2, 3, 5, 10\}$, while fixing the total sample size at $N = 6000$ and maintaining same size for each source. We next vary the balance-ratio $\rho = \max_k n_k / \min_k n_k \in \{1, 2, 4, 8\}$ with corresponding sample sizes $(2000, 2000, 2000), (3000, 1500, 1500), (4000, 1000, 1000), (4800, 600, 600)$ and fixes $K = 3$ and $N = 6000$. For $\rho > 1$, we randomly assign sample sizes for different source labels in each trial so the dominant source is not tied to a fixed source index. Finally, we vary the total sample size $N \in \{1500, 3000, 4500, 6000\}$ while fixing $K = 3$ and assuming balanced sources, that is, each source contains the same number of samples.

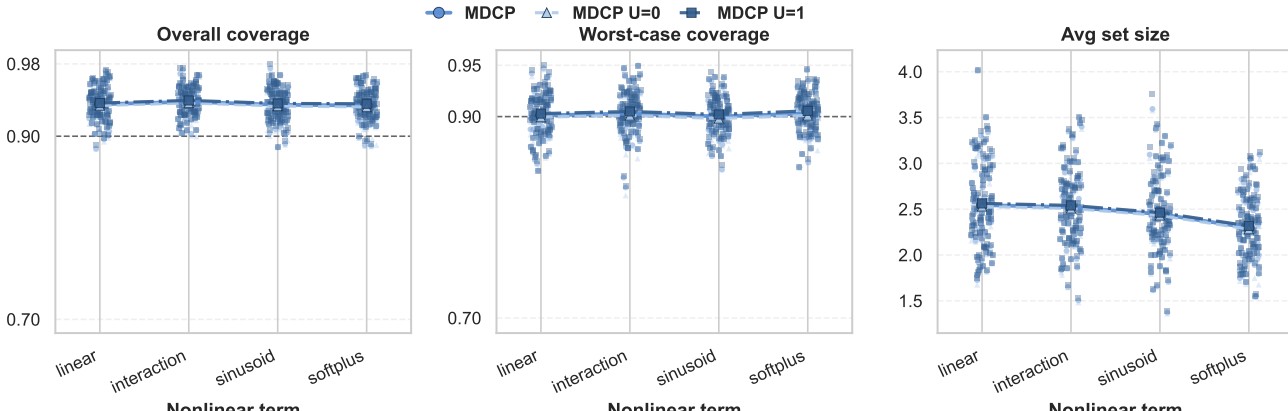

*Figure 24.* Performance of MDCP with randomized and deterministic $p$-values over the regression NONLINEAR suites; details are the same as in Figure 21.

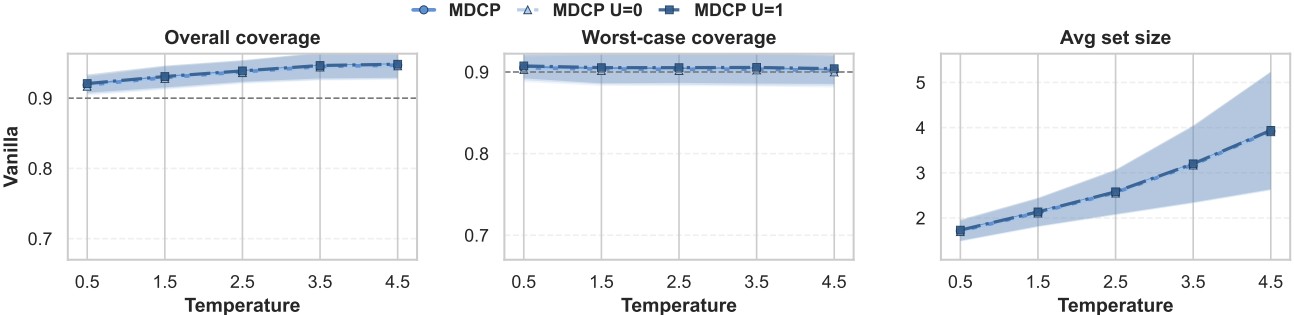

*Figure 25.* Performance of MDCP with randomized and deterministic $p$-values over the regression TEMPERATURE suites; details are the same as in Figure 22.

**Method implementations.** All methods follow Section 5.2. Additionally, we include an ORACLE method that has access to the ground-truth conditional density during both $\lambda$ optimization and score calculation. For simplicity we do not tune MDCP here.

**Simulation results.** Tables 1–3 collect the 120 classification configurations. Across all of them, MDCP keeps worst-case coverage in $[0.898, 0.922]$, much closer to the nominal 0.9 target than BASELINE-AGG, whose worst-case coverage ranges from 0.941 to 1.000 and is systematically conservative. The same pattern appears in overall coverage: MDCP ranges from 0.910 to 0.980, whereas BASELINE-AGG ranges from 0.953 to 1.000. Relative to BASELINE-AGG, MDCP reduces the prediction-set size by a median $32.1\%$, with reductions ranging from $8.9\%$ to $51.0\%$ across the 120 configurations. The runtime-breakdown tables, Tables 4–6, show that learning $\lambda$ is the dominant MDCP cost in classification, accounting for a median $77.9\%$ of the total runtime across the 120 configurations, whereas the time for calibration and test evaluation remain comparatively small. In contrast, BASELINE-AGG is faster since, by construction, it does not require fitting conditional density on pooled data and learning $\lambda$. Among the three parameters we vary, increasing $N$ leads to the most significant increase in MDCP's runtime, while the absolute changes are comparatively smaller when varying $K$ and $\rho$.

### D.5.2. SCALING STUDIES IN REGRESSION SETTINGS

**Data generating processes.** The DGP follows that in Section 5.3. The scaling scheme is the same as in classification.

**Method implementations.** All methods follow Section 5.3, with an additional ORACLE method as in the classification.

**Simulation results.** Tables 7–9 report the 120 regression configurations. Again MDCP stays close to the target worst-case coverage while BASELINE-AGG remains conservative. MDCP shortens the average interval width by a median $26.3\%$ relative to BASELINE-AGG, with reductions ranging from $8.8\%$ to $61.6\%$. The runtime breakdown tables (Tables 10–12) show that the runtime overhead is milder than in classification. This is mainly because the $\lambda$-optimization step is less

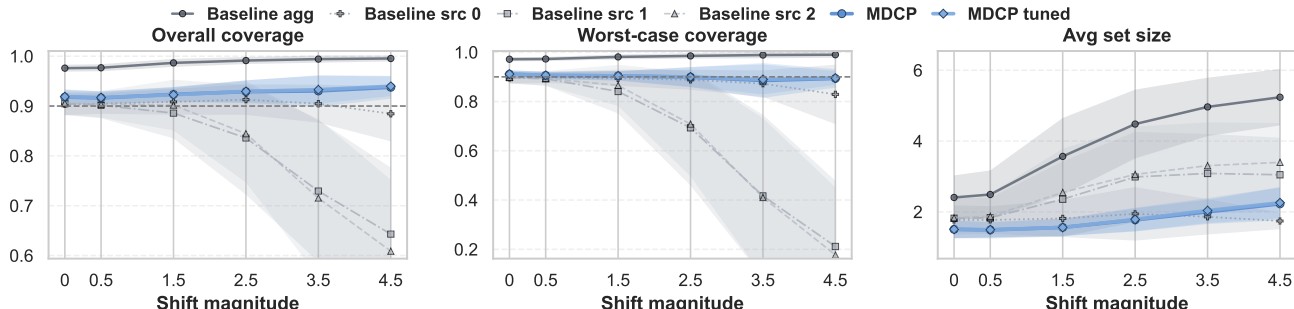

*Figure 26.* Performance of MDCP and baselines in the classification COVARIATE-SHIFT experiments. The x-axis is the covariate-shift magnitude $\delta_X$, which determines $P_X^{(k)}$ separation while keeping $P(Y \mid X)$ fixed. Each line reports the mean over 100 runs, and the shaded region indicates $\pm 1$ standard deviation across runs. Left: coverage over all test data. Middle: worst-case coverage over single-source test data. Right: average set size over all test data.

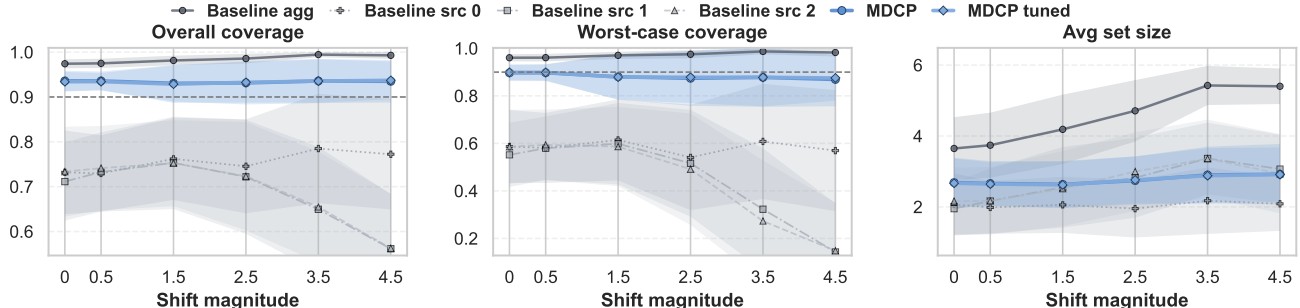

*Figure 27.* Performance of MDCP and baselines in the classification COVARIATE-AND-CONCEPT-SHIFT experiments. The x-axis is the covariate shift magnitude $\delta_X$, with both $P_X^{(k)}$ and $P^{(k)}(Y \mid X)$ varying across sources. Each line reports the mean over 100 runs, and the shaded region indicates $\pm 1$ standard deviation. Left: coverage over all test data. Middle: worst-case coverage over single-source test data. Right: average set size over all test data.

dominant than in classification, and we find that, during this step, regression generally requires fewer iterations to converge.

Finally, we find that the total runtime seems to scale linearly with the total sample size $N$, mildly increases with the number of sources $K$, and remains robust to the balance between groups.

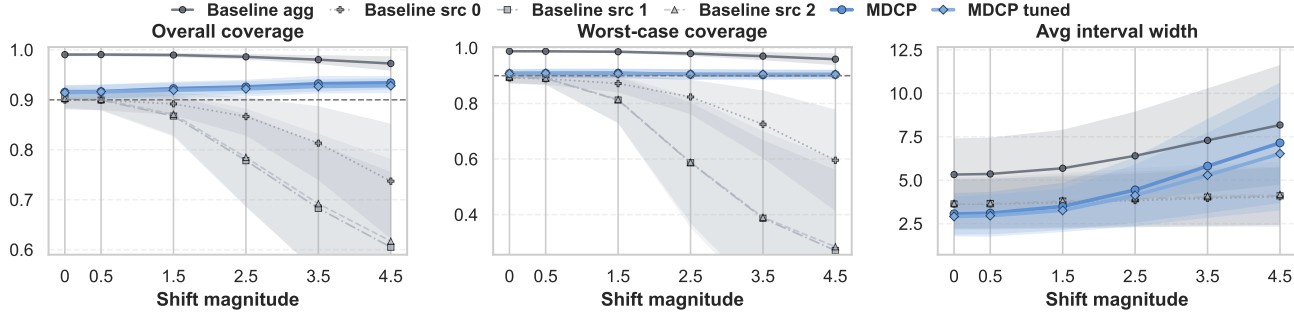

*Figure 28.* Evaluation with regression COVARIATE-SHIFT suites; details are otherwise the same as in Figure 26.

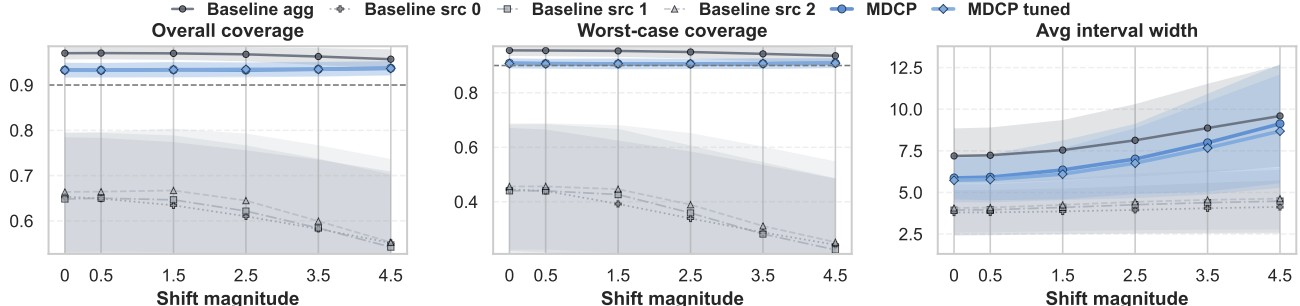

*Figure 29.* Evaluation with regression COVARIATE-AND-CONCEPT-SHIFT suites; details are otherwise the same as in Figure 27.

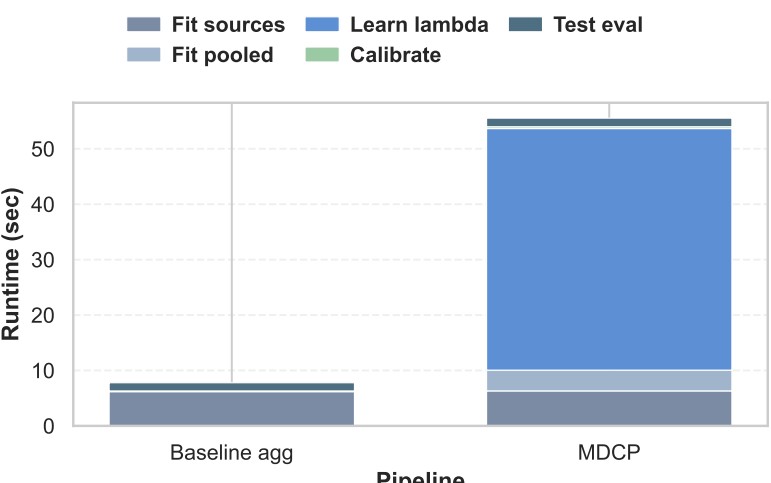

*Figure 30.* Runtime breakdown of MDCP on the classification LINEAR suites, compared with BASELINE-AGG introduced in Section 5.1. Here, *Fit sources* denotes the time to fit per-source conditional densities; *Fit pooled* the time to fit a conditional density for the pooled mixture; *Learn* $\lambda$ the time to solve the optimization problem for the weights $\lambda_k$; and *Calibrate* and *Test eval* the time for calibration and test evaluation, respectively.

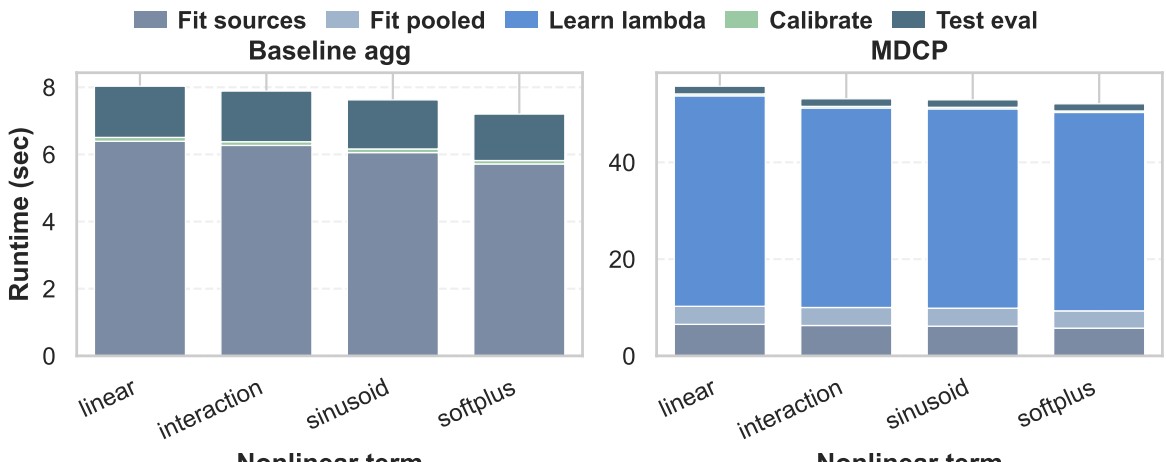

*Figure 31.* Runtime breakdown of MDCP and BASELINE-AGG on the classification NONLINEAR suites; details are as in Fig. 30.

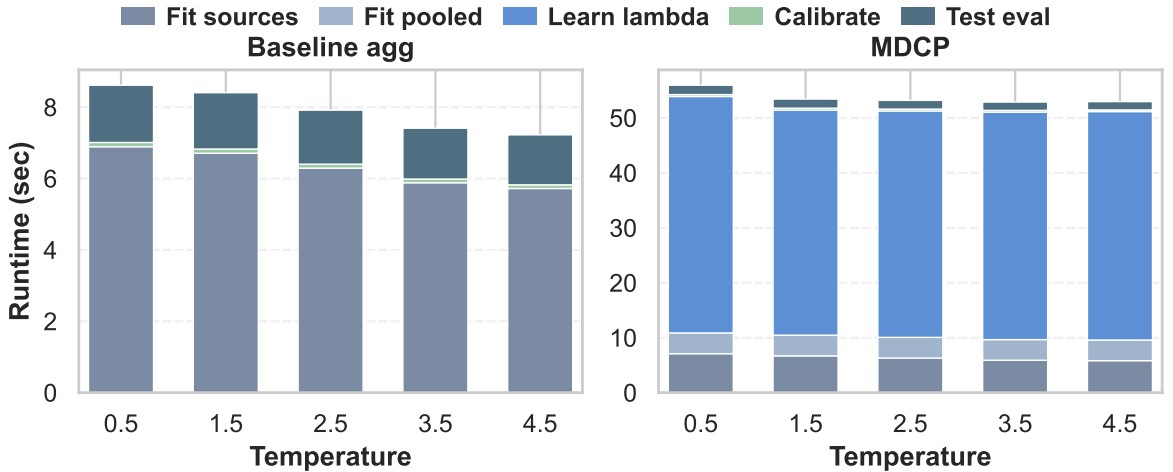

*Figure 32.* Runtime breakdown of MDCP and BASELINE-AGG on the classification TEMPERATURE suites; details are as in Fig. 30.

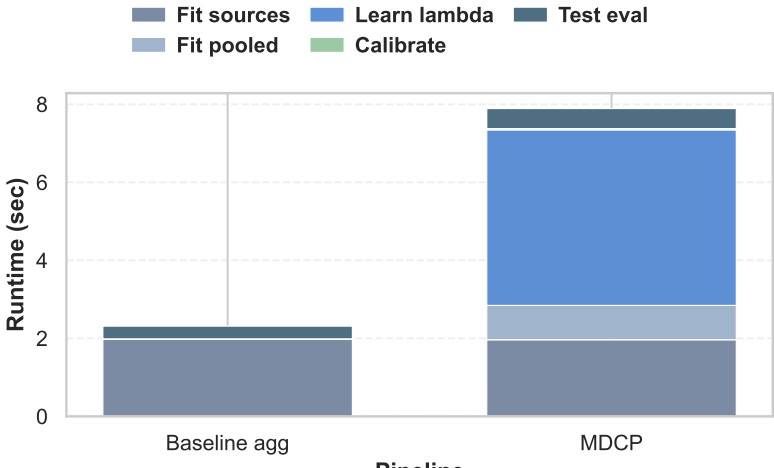

*Figure 33.* Runtime breakdown of MDCP and BASELINE-AGG on the regression LINEAR suites; details are as in Fig. 30.

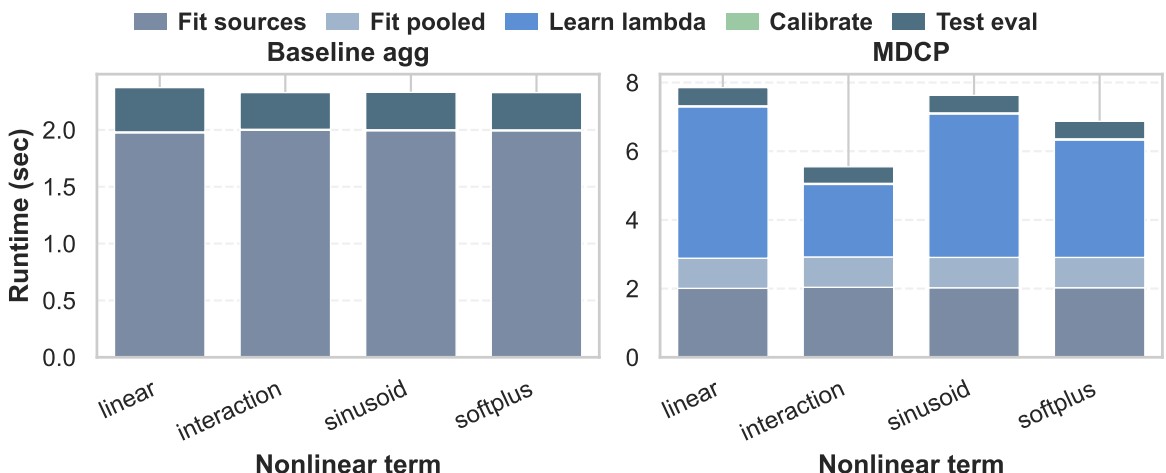

*Figure 34.* Runtime breakdown of MDCP and BASELINE-AGG on the regression NONLINEAR suites; details are as in Fig. 30.

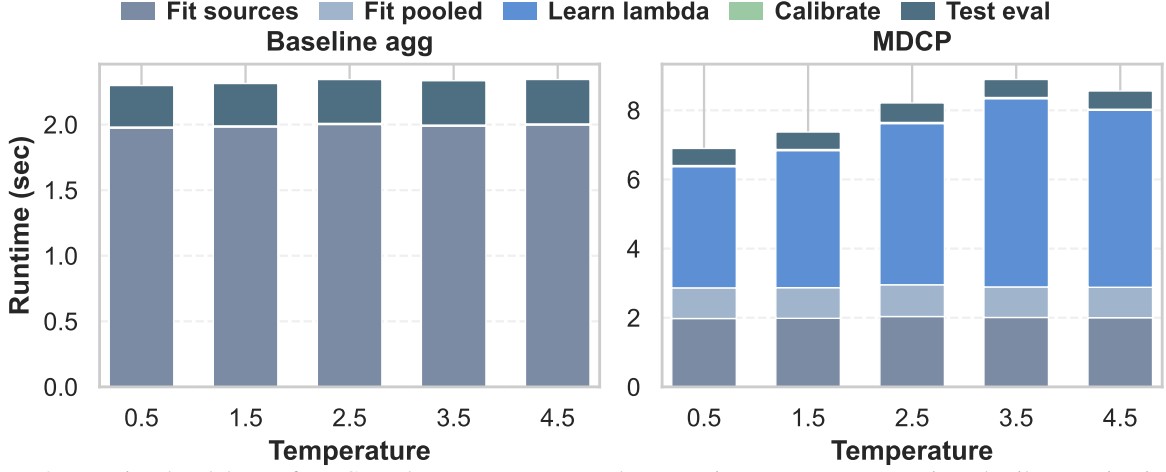

*Figure 35.* Runtime breakdown of MDCP and BASELINE-AGG on the regression TEMPERATURE suites; details are as in Fig. 30.

| Suite | Setting | $\rho$ | Overall coverage | Worst-case coverage | Avg set size | Total runtime |
|---|---|---|---|---|---|---|
| LINEAR | $\tau = 2.5$ | 1 | 0.975 / 0.936 / 0.921 | 0.961 / 0.903 / 0.908 | 3.78 / 2.53 / 2.26 | 7.1 / 53.3 / 2.3 |
| | | 2 | 0.975 / 0.934 / 0.922 | 0.960 / 0.903 / 0.908 | 3.72 / 2.67 / 2.32 | 7.0 / 51.3 / 2.3 |
| | | 4 | 0.975 / 0.927 / 0.922 | 0.963 / 0.901 / 0.907 | 3.90 / 3.44 / 2.32 | 6.1 / 50.6 / 2.8 |
| | | 8 | 0.981 / 0.916 / 0.925 | 0.970 / 0.902 / 0.910 | 4.32 / 3.74 / 2.44 | 5.4 / 50.1 / 3.5 |
| TEMPERATURE | $\tau = 0.5$ | 1 | 0.974 / 0.917 / 0.917 | 0.967 / 0.904 / 0.907 | 2.66 / 1.74 / 1.58 | 8.4 / 54.7 / 1.4 |
| | | 2 | 0.975 / 0.918 / 0.915 | 0.968 / 0.904 / 0.905 | 2.72 / 1.78 / 1.58 | 8.0 / 52.4 / 1.5 |
| | | 4 | 0.977 / 0.919 / 0.922 | 0.971 / 0.907 / 0.911 | 2.90 / 1.99 / 1.63 | 7.1 / 52.0 / 1.5 |
| | | 8 | 0.980 / 0.921 / 0.921 | 0.974 / 0.911 / 0.909 | 3.24 / 2.20 / 1.63 | 6.2 / 51.6 / 1.6 |
| | $\tau = 1.5$ | 1 | 0.972 / 0.926 / 0.916 | 0.962 / 0.903 / 0.904 | 2.98 / 2.08 / 1.88 | 8.1 / 54.7 / 2.2 |
| | | 2 | 0.973 / 0.922 / 0.918 | 0.963 / 0.900 / 0.906 | 2.97 / 2.10 / 1.89 | 7.7 / 52.4 / 2.4 |
| | | 4 | 0.976 / 0.922 / 0.919 | 0.964 / 0.901 / 0.906 | 3.29 / 2.70 / 1.93 | 6.8 / 51.7 / 2.4 |
| | | 8 | 0.979 / 0.915 / 0.923 | 0.967 / 0.902 / 0.910 | 3.49 / 2.89 / 2.00 | 6.0 / 51.3 / 3.0 |
| | $\tau = 2.5$ | 1 | 0.974 / 0.933 / 0.916 | 0.962 / 0.902 / 0.904 | 3.63 / 2.47 / 2.21 | 8.1 / 55.4 / 2.3 |
| | | 2 | 0.975 / 0.931 / 0.923 | 0.963 / 0.903 / 0.910 | 3.73 / 2.60 / 2.28 | 7.6 / 52.9 / 2.4 |
| | | 4 | 0.978 / 0.926 / 0.923 | 0.964 / 0.900 / 0.908 | 3.96 / 3.27 / 2.27 | 6.7 / 52.2 / 2.8 |
| | | 8 | 0.982 / 0.919 / 0.924 | 0.970 / 0.902 / 0.909 | 4.29 / 3.76 / 2.35 | 5.7 / 51.5 / 3.7 |
| | $\tau = 3.5$ | 1 | 0.977 / 0.939 / 0.923 | 0.964 / 0.901 / 0.908 | 4.30 / 2.96 / 2.70 | 7.4 / 55.0 / 2.1 |
| | | 2 | 0.978 / 0.939 / 0.924 | 0.964 / 0.904 / 0.905 | 4.48 / 3.22 / 2.83 | 7.0 / 52.5 / 2.1 |
| | | 4 | 0.980 / 0.932 / 0.925 | 0.965 / 0.904 / 0.907 | 4.46 / 3.86 / 2.70 | 6.4 / 52.0 / 2.6 |
| | | 8 | 0.982 / 0.923 / 0.926 | 0.969 / 0.905 / 0.907 | 4.77 / 4.19 / 2.82 | 5.4 / 51.1 / 3.5 |
| | $\tau = 4.5$ | 1 | 0.981 / 0.944 / 0.920 | 0.966 / 0.901 / 0.902 | 4.86 / 3.51 / 3.24 | 7.3 / 54.7 / 1.8 |
| | | 2 | 0.984 / 0.939 / 0.925 | 0.971 / 0.900 / 0.906 | 5.11 / 3.87 / 3.53 | 6.9 / 52.3 / 1.8 |
| | | 4 | 0.986 / 0.934 / 0.925 | 0.973 / 0.905 / 0.905 | 5.14 / 4.38 / 3.23 | 6.1 / 51.5 / 2.1 |
| | | 8 | 0.985 / 0.923 / 0.928 | 0.972 / 0.902 / 0.905 | 5.17 / 4.71 / 3.47 | 5.3 / 50.9 / 3.1 |
| NONLINEAR | LINEAR | 1 | 0.974 / 0.936 / 0.919 | 0.962 / 0.903 / 0.907 | 3.64 / 2.42 / 2.15 | 7.9 / 54.8 / 2.3 |
| | | 2 | 0.975 / 0.931 / 0.922 | 0.963 / 0.902 / 0.908 | 3.75 / 2.60 / 2.22 | 7.5 / 52.7 / 2.4 |
| | | 4 | 0.977 / 0.926 / 0.922 | 0.964 / 0.902 / 0.907 | 3.82 / 3.34 / 2.30 | 6.7 / 52.1 / 2.9 |
| | | 8 | 0.981 / 0.920 / 0.925 | 0.969 / 0.904 / 0.909 | 4.24 / 3.67 / 2.38 | 5.7 / 51.3 / 3.4 |
| | INTERACTION | 1 | 0.973 / 0.936 / 0.919 | 0.961 / 0.902 / 0.905 | 3.56 / 2.45 / 2.11 | 7.8 / 52.9 / 2.4 |
| | | 2 | 0.974 / 0.931 / 0.916 | 0.961 / 0.903 / 0.902 | 3.56 / 2.47 / 2.07 | 7.6 / 53.1 / 2.4 |
| | | 4 | 0.975 / 0.926 / 0.921 | 0.963 / 0.901 / 0.907 | 3.69 / 3.19 / 2.08 | 6.8 / 52.5 / 2.8 |
| | | 8 | 0.983 / 0.920 / 0.926 | 0.972 / 0.905 / 0.912 | 4.28 / 3.63 / 2.21 | 6.0 / 51.9 / 3.4 |
| | SINUSOID | 1 | 0.973 / 0.934 / 0.917 | 0.961 / 0.903 / 0.906 | 3.57 / 2.43 / 2.17 | 8.5 / 56.4 / 2.4 |
| | | 2 | 0.975 / 0.931 / 0.917 | 0.961 / 0.903 / 0.904 | 3.66 / 2.45 / 2.14 | 8.1 / 54.1 / 2.4 |
| | | 4 | 0.976 / 0.926 / 0.919 | 0.963 / 0.901 / 0.905 | 3.74 / 3.18 / 2.11 | 7.1 / 53.4 / 2.7 |
| | | 8 | 0.983 / 0.917 / 0.928 | 0.969 / 0.900 / 0.913 | 4.29 / 3.43 / 2.26 | 6.1 / 52.5 / 3.6 |
| | SOFTPLUS | 1 | 0.982 / 0.934 / 0.921 | 0.973 / 0.906 / 0.908 | 4.02 / 2.37 / 2.13 | 7.8 / 53.4 / 2.3 |
| | | 2 | 0.985 / 0.927 / 0.917 | 0.976 / 0.899 / 0.903 | 4.20 / 2.41 / 2.11 | 7.4 / 53.4 / 2.5 |
| | | 4 | 0.987 / 0.928 / 0.921 | 0.979 / 0.903 / 0.908 | 4.41 / 3.11 / 2.15 | 6.6 / 52.7 / 2.7 |
| | | 8 | 0.991 / 0.917 / 0.922 | 0.984 / 0.900 / 0.907 | 4.67 / 3.22 / 2.15 | 5.7 / 51.9 / 3.6 |

*Table 1.* Classification results when varying the balance ratio $\rho$. Each metric cell reports the mean over 100 trials in the order BASELINE-AGG / MDCP / ORACLE. Runtime values are reported in seconds. Except for the varied parameters, the LINEAR suite setup follows that of Figure 2; the NONLINEAR suite setup follows that of Figure 3; and the TEMPERATURE suite setup follows that of Figure 4.

| Suite | Setting | $K$ | Overall coverage | Worst-case coverage | Avg set size | Total runtime |
|-------|---------|-----|------------------|---------------------|--------------|---------------|
| LINEAR | $\tau = 2.5$ | 2 | 0.953 / 0.923 / 0.909 | 0.942 / 0.901 / 0.901 | 2.97 / 2.25 / 1.95 | 6.2 / 50.1 / 1.7 |
| | | 3 | 0.975 / 0.935 / 0.921 | 0.962 / 0.902 / 0.908 | 3.86 / 2.51 / 2.25 | 7.0 / 51.2 / 2.3 |
| | | 5 | 0.990 / 0.949 / 0.932 | 0.978 / 0.905 / 0.910 | 4.63 / 2.96 / 2.66 | 8.1 / 54.3 / 3.7 |
| | | 10 | 0.999 / 0.963 / 0.954 | 0.996 / 0.913 / 0.917 | 5.66 / 3.39 / 3.19 | 10.1 / 59.6 / 10.3 |
| TEMPERATURE | $\tau = 0.5$ | 2 | 0.958 / 0.910 / 0.908 | 0.953 / 0.902 / 0.902 | 2.31 / 1.68 / 1.51 | 7.2 / 49.1 / 1.3 |
| | | 3 | 0.974 / 0.917 / 0.917 | 0.967 / 0.904 / 0.907 | 2.66 / 1.74 / 1.58 | 8.3 / 52.6 / 1.4 |
| | | 5 | 0.987 / 0.931 / 0.929 | 0.981 / 0.912 / 0.913 | 3.25 / 1.90 / 1.70 | 9.7 / 56.5 / 1.8 |
| | | 10 | 0.998 / 0.949 / 0.949 | 0.993 / 0.922 / 0.926 | 4.56 / 2.24 / 1.92 | 12.1 / 62.7 / 2.9 |
| | $\tau = 1.5$ | 2 | 0.954 / 0.916 / 0.910 | 0.945 / 0.901 / 0.904 | 2.49 / 1.93 / 1.71 | 7.0 / 50.7 / 1.7 |
| | | 3 | 0.972 / 0.927 / 0.915 | 0.963 / 0.903 / 0.903 | 3.00 / 2.08 / 1.88 | 8.0 / 52.6 / 2.2 |
| | | 5 | 0.987 / 0.941 / 0.931 | 0.976 / 0.906 / 0.911 | 3.72 / 2.40 / 2.17 | 9.2 / 56.1 / 3.3 |
| | | 10 | 0.998 / 0.958 / 0.952 | 0.993 / 0.918 / 0.922 | 5.10 / 2.84 / 2.59 | 11.3 / 61.8 / 7.3 |
| | $\tau = 2.5$ | 2 | 0.954 / 0.920 / 0.908 | 0.941 / 0.900 / 0.901 | 2.86 / 2.19 / 1.94 | 6.9 / 51.5 / 2.8 |
| | | 3 | 0.973 / 0.933 / 0.917 | 0.962 / 0.903 / 0.905 | 3.61 / 2.48 / 2.21 | 7.8 / 52.8 / 2.3 |
| | | 5 | 0.990 / 0.946 / 0.931 | 0.979 / 0.902 / 0.911 | 4.65 / 2.92 / 2.65 | 8.8 / 56.3 / 3.9 |
| | | 10 | 0.999 / 0.963 / 0.955 | 0.995 / 0.917 / 0.920 | 5.66 / 3.42 / 3.22 | 10.8 / 61.7 / 10.4 |
| | $\tau = 3.5$ | 2 | 0.955 / 0.924 / 0.912 | 0.941 / 0.900 / 0.902 | 3.50 / 2.69 / 2.36 | 6.6 / 51.4 / 1.5 |
| | | 3 | 0.978 / 0.939 / 0.922 | 0.966 / 0.901 / 0.908 | 4.41 / 2.99 / 2.72 | 7.4 / 52.5 / 2.1 |
| | | 5 | 0.993 / 0.957 / 0.933 | 0.983 / 0.903 / 0.906 | 5.26 / 3.58 / 3.21 | 8.6 / 55.9 / 3.4 |
| | | 10 | 1.000 / 0.972 / 0.961 | 0.998 / 0.908 / 0.919 | 5.94 / 4.22 / 4.04 | 10.2 / 60.9 / 11.0 |
| | $\tau = 4.5$ | 2 | 0.961 / 0.925 / 0.917 | 0.946 / 0.904 / 0.905 | 4.05 / 3.25 / 2.88 | 6.7 / 51.8 / 1.5 |
| | | 3 | 0.982 / 0.943 / 0.921 | 0.968 / 0.898 / 0.903 | 5.01 / 3.57 / 3.32 | 7.2 / 52.5 / 1.8 |
| | | 5 | 0.995 / 0.963 / 0.935 | 0.987 / 0.901 / 0.903 | 5.68 / 4.43 / 4.18 | 8.1 / 55.6 / 2.6 |
| | | 10 | 1.000 / 0.980 / 0.962 | 1.000 / 0.912 / 0.916 | 6.00 / 5.17 / 5.11 | 9.9 / 60.6 / 6.7 |
| NONLINEAR | LINEAR | 2 | 0.955 / 0.918 / 0.908 | 0.947 / 0.901 / 0.900 | 2.87 / 2.05 / 1.84 | 6.8 / 51.5 / 1.7 |
| | | 3 | 0.973 / 0.935 / 0.920 | 0.961 / 0.903 / 0.908 | 3.55 / 2.44 / 2.19 | 7.7 / 52.9 / 2.3 |
| | | 5 | 0.988 / 0.949 / 0.934 | 0.977 / 0.908 / 0.912 | 4.51 / 2.97 / 2.70 | 8.9 / 56.3 / 4.0 |
| | | 10 | 0.999 / 0.964 / 0.954 | 0.996 / 0.914 / 0.919 | 5.73 / 3.45 / 3.24 | 10.6 / 61.7 / 9.7 |
| | INTERACTION | 2 | 0.953 / 0.921 / 0.910 | 0.944 / 0.903 / 0.903 | 2.76 / 2.06 / 1.78 | 6.8 / 49.6 / 1.7 |
| | | 3 | 0.974 / 0.936 / 0.919 | 0.961 / 0.901 / 0.905 | 3.56 / 2.45 / 2.10 | 7.8 / 53.0 / 2.3 |
| | | 5 | 0.989 / 0.947 / 0.933 | 0.978 / 0.905 / 0.914 | 4.47 / 2.85 / 2.49 | 8.7 / 56.4 / 4.0 |
| | | 10 | 0.999 / 0.963 / 0.950 | 0.995 / 0.909 / 0.915 | 5.60 / 3.47 / 3.03 | 10.9 / 62.0 / 10.4 |
| | SINUSOID | 2 | 0.956 / 0.923 / 0.910 | 0.946 / 0.905 / 0.903 | 2.85 / 2.08 / 1.80 | 7.1 / 52.2 / 1.7 |
| | | 3 | 0.973 / 0.934 / 0.917 | 0.960 / 0.903 / 0.905 | 3.48 / 2.38 / 2.12 | 8.3 / 54.4 / 2.5 |
| | | 5 | 0.990 / 0.945 / 0.931 | 0.978 / 0.901 / 0.909 | 4.49 / 2.80 / 2.55 | 9.4 / 57.8 / 3.8 |
| | | 10 | 0.999 / 0.966 / 0.956 | 0.995 / 0.919 / 0.922 | 5.60 / 3.46 / 3.22 | 11.2 / 63.5 / 10.9 |
| | SOFTPLUS | 2 | 0.967 / 0.921 / 0.909 | 0.960 / 0.902 / 0.902 | 3.39 / 2.03 / 1.74 | 6.5 / 48.6 / 1.7 |
| | | 3 | 0.984 / 0.931 / 0.919 | 0.976 / 0.903 / 0.906 | 4.09 / 2.27 / 2.06 | 7.7 / 53.4 / 2.4 |
| | | 5 | 0.995 / 0.942 / 0.933 | 0.987 / 0.905 / 0.912 | 4.94 / 2.63 / 2.47 | 8.6 / 57.0 / 3.9 |
| | | 10 | 0.999 / 0.964 / 0.954 | 0.994 / 0.914 / 0.919 | 5.43 / 3.20 / 3.01 | 10.3 / 62.0 / 10.6 |

*Table 2.* Classification results when varying the number of sources $K$. Details are otherwise the same as in Table 1.

| Suite | Setting | $N$ | Overall coverage | Worst-case coverage | Avg set size | Total runtime |
|---|---|---|---|---|---|---|
| LINEAR | $\tau = 2.5$ | 1500 | 0.984 / 0.941 / 0.934 | 0.974 / 0.911 / 0.913 | 4.59 / 3.00 / 2.47 | 2.0 / 19.3 / 1.2 |
| | | 3000 | 0.980 / 0.938 / 0.925 | 0.968 / 0.905 / 0.908 | 4.20 / 2.72 / 2.36 | 3.7 / 29.4 / 1.7 |
| | | 4500 | 0.978 / 0.940 / 0.924 | 0.966 / 0.907 / 0.909 | 3.95 / 2.57 / 2.26 | 5.4 / 40.6 / 2.1 |
| | | 6000 | 0.973 / 0.933 / 0.920 | 0.960 / 0.901 / 0.908 | 3.64 / 2.45 / 2.20 | 6.9 / 51.1 / 2.3 |
| TEMPERATURE | $\tau = 0.5$ | 1500 | 0.982 / 0.931 / 0.933 | 0.975 / 0.915 / 0.919 | 3.60 / 2.20 / 1.71 | 2.4 / 19.9 / 0.6 |
| | | 3000 | 0.977 / 0.923 / 0.923 | 0.971 / 0.908 / 0.912 | 3.04 / 1.92 / 1.65 | 4.5 / 30.3 / 0.9 |
| | | 4500 | 0.976 / 0.919 / 0.917 | 0.970 / 0.907 / 0.906 | 2.80 / 1.80 / 1.58 | 6.4 / 41.8 / 1.2 |
| | | 6000 | 0.974 / 0.917 / 0.917 | 0.967 / 0.904 / 0.907 | 2.66 / 1.74 / 1.58 | 8.3 / 52.5 / 1.4 |
| | $\tau = 1.5$ | 1500 | 0.981 / 0.937 / 0.931 | 0.970 / 0.912 / 0.912 | 3.89 / 2.58 / 2.04 | 2.3 / 19.6 / 1.0 |
| | | 3000 | 0.976 / 0.929 / 0.926 | 0.965 / 0.902 / 0.912 | 3.36 / 2.24 / 1.96 | 4.1 / 30.0 / 1.5 |
| | | 4500 | 0.975 / 0.931 / 0.923 | 0.965 / 0.908 / 0.910 | 3.15 / 2.17 / 1.93 | 6.1 / 41.6 / 1.9 |
| | | 6000 | 0.972 / 0.926 / 0.915 | 0.962 / 0.902 / 0.903 | 2.97 / 2.08 / 1.88 | 8.0 / 52.4 / 2.2 |
| | $\tau = 2.5$ | 1500 | 0.985 / 0.942 / 0.936 | 0.975 / 0.912 / 0.917 | 4.77 / 2.97 / 2.41 | 2.1 / 19.8 / 1.1 |
| | | 3000 | 0.980 / 0.936 / 0.922 | 0.968 / 0.903 / 0.905 | 4.24 / 2.68 / 2.32 | 4.1 / 30.6 / 1.7 |
| | | 4500 | 0.978 / 0.936 / 0.922 | 0.966 / 0.904 / 0.907 | 3.91 / 2.51 / 2.23 | 5.9 / 42.1 / 2.0 |
| | | 6000 | 0.973 / 0.933 / 0.917 | 0.961 / 0.902 / 0.905 | 3.59 / 2.49 / 2.23 | 7.9 / 53.1 / 2.3 |
| | $\tau = 3.5$ | 1500 | 0.988 / 0.949 / 0.940 | 0.977 / 0.916 / 0.915 | 5.15 / 3.46 / 2.91 | 2.1 / 19.8 / 1.1 |
| | | 3000 | 0.983 / 0.943 / 0.927 | 0.970 / 0.902 / 0.908 | 4.70 / 3.13 / 2.74 | 3.8 / 30.3 / 1.7 |
| | | 4500 | 0.981 / 0.942 / 0.923 | 0.967 / 0.902 / 0.906 | 4.58 / 3.08 / 2.73 | 5.7 / 42.0 / 1.8 |
| | | 6000 | 0.977 / 0.939 / 0.922 | 0.964 / 0.901 / 0.907 | 4.34 / 2.95 / 2.68 | 7.3 / 52.7 / 2.1 |
| | $\tau = 4.5$ | 1500 | 0.990 / 0.952 / 0.937 | 0.979 / 0.909 / 0.906 | 5.50 / 4.19 / 3.78 | 2.1 / 19.8 / 1.0 |
| | | 3000 | 0.987 / 0.949 / 0.929 | 0.975 / 0.902 / 0.903 | 5.23 / 3.86 / 3.55 | 3.7 / 30.2 / 1.3 |
| | | 4500 | 0.985 / 0.944 / 0.926 | 0.973 / 0.902 / 0.905 | 5.10 / 3.59 / 3.34 | 5.4 / 41.6 / 1.6 |
| | | 6000 | 0.983 / 0.942 / 0.922 | 0.969 / 0.899 / 0.903 | 5.00 / 3.61 / 3.38 | 7.3 / 52.6 / 1.8 |
| NONLINEAR | LINEAR | 1500 | 0.985 / 0.944 / 0.936 | 0.975 / 0.914 / 0.916 | 4.57 / 2.87 / 2.30 | 2.2 / 19.8 / 1.2 |
| | | 3000 | 0.979 / 0.938 / 0.926 | 0.968 / 0.907 / 0.908 | 4.01 / 2.63 / 2.26 | 3.9 / 30.2 / 1.8 |
| | | 4500 | 0.975 / 0.937 / 0.921 | 0.962 / 0.905 / 0.905 | 3.84 / 2.57 / 2.25 | 5.8 / 41.8 / 2.0 |
| | | 6000 | 0.973 / 0.935 / 0.920 | 0.960 / 0.902 / 0.908 | 3.59 / 2.45 / 2.20 | 7.7 / 52.8 / 2.4 |
| | INTERACTION | 1500 | 0.985 / 0.936 / 0.932 | 0.975 / 0.906 / 0.912 | 4.66 / 2.88 / 2.20 | 2.1 / 17.8 / 1.3 |
| | | 3000 | 0.978 / 0.937 / 0.923 | 0.966 / 0.906 / 0.905 | 4.00 / 2.66 / 2.18 | 4.1 / 30.5 / 1.8 |
| | | 4500 | 0.974 / 0.936 / 0.919 | 0.961 / 0.901 / 0.904 | 3.72 / 2.51 / 2.13 | 6.1 / 42.1 / 2.2 |
| | | 6000 | 0.972 / 0.934 / 0.919 | 0.960 / 0.900 / 0.904 | 3.45 / 2.40 / 2.08 | 7.8 / 53.0 / 2.3 |
| | SINUSOID | 1500 | 0.986 / 0.939 / 0.937 | 0.976 / 0.908 / 0.915 | 4.63 / 2.78 / 2.29 | 2.2 / 19.9 / 1.3 |
| | | 3000 | 0.979 / 0.935 / 0.924 | 0.967 / 0.903 / 0.908 | 4.04 / 2.51 / 2.19 | 4.1 / 31.0 / 1.9 |
| | | 4500 | 0.976 / 0.932 / 0.920 | 0.963 / 0.899 / 0.905 | 3.72 / 2.43 / 2.18 | 6.1 / 42.7 / 2.1 |
| | | 6000 | 0.973 / 0.934 / 0.917 | 0.961 / 0.904 / 0.905 | 3.62 / 2.43 / 2.16 | 8.2 / 54.2 / 2.4 |
| | SOFTPLUS | 1500 | 0.993 / 0.937 / 0.934 | 0.986 / 0.909 / 0.916 | 4.96 / 2.66 / 2.20 | 2.0 / 17.5 / 1.2 |
| | | 3000 | 0.988 / 0.935 / 0.926 | 0.980 / 0.904 / 0.911 | 4.52 / 2.43 / 2.15 | 4.0 / 30.6 / 1.7 |
| | | 4500 | 0.986 / 0.933 / 0.921 | 0.977 / 0.898 / 0.904 | 4.27 / 2.36 / 2.12 | 5.9 / 42.0 / 2.1 |
| | | 6000 | 0.981 / 0.935 / 0.920 | 0.972 / 0.905 / 0.907 | 3.90 / 2.31 / 2.09 | 7.8 / 53.7 / 2.3 |

*Table 3.* Classification results when varying the total sample size $N$. Details are otherwise the same as in Table 1.

| Suite | Setting | $\rho$ | Fit sources | Fit pooled | Learn $\lambda$ | Calibrate | Test eval | Total runtime |
|---|---|---|---|---|---|---|---|---|
| LINEAR | $\tau = 2.5$ | 1 | 5.6 / 5.5 / 0.0 | – / 3.4 / 0.0 | – / 42.7 / 2.0 | 0.1 / 0.3 / 0.0 | 1.4 / 1.5 / 0.3 | 7.1 / 53.3 / 2.3 |
| | | 2 | 5.5 / 5.5 / 0.0 | – / 3.4 / 0.0 | – / 40.6 / 2.0 | 0.1 / 0.3 / 0.0 | 1.4 / 1.5 / 0.3 | 7.0 / 51.3 / 2.3 |
| | | 4 | 4.7 / 4.7 / 0.0 | – / 3.4 / 0.0 | – / 40.8 / 2.5 | 0.1 / 0.3 / 0.0 | 1.3 / 1.4 / 0.3 | 6.1 / 50.6 / 2.8 |
| | | 8 | 4.1 / 4.1 / 0.0 | – / 3.4 / 0.0 | – / 41.1 / 3.2 | 0.1 / 0.3 / 0.0 | 1.2 / 1.4 / 0.3 | 5.4 / 50.1 / 3.5 |
| TEMPERATURE | $\tau = 0.5$ | 1 | 6.7 / 6.6 / 0.0 | – / 3.6 / 0.0 | – / 42.6 / 1.1 | 0.1 / 0.3 / 0.0 | 1.6 / 1.6 / 0.3 | 8.4 / 54.7 / 1.4 |
| | | 2 | 6.4 / 6.4 / 0.0 | – / 3.6 / 0.0 | – / 40.5 / 1.2 | 0.1 / 0.3 / 0.0 | 1.5 / 1.7 / 0.4 | 8.0 / 52.4 / 1.5 |
| | | 4 | 5.5 / 5.5 / 0.0 | – / 3.6 / 0.0 | – / 41.0 / 1.2 | 0.1 / 0.3 / 0.0 | 1.5 / 1.5 / 0.3 | 7.1 / 52.0 / 1.5 |
| | | 8 | 4.7 / 4.7 / 0.0 | – / 3.6 / 0.0 | – / 41.5 / 1.3 | 0.1 / 0.3 / 0.0 | 1.3 / 1.4 / 0.3 | 6.2 / 51.6 / 1.6 |
| | $\tau = 1.5$ | 1 | 6.5 / 6.4 / 0.0 | – / 3.6 / 0.0 | – / 42.8 / 1.9 | 0.1 / 0.3 / 0.0 | 1.5 / 1.6 / 0.3 | 8.1 / 54.7 / 2.2 |
| | | 2 | 6.1 / 6.1 / 0.0 | – / 3.6 / 0.0 | – / 40.7 / 2.0 | 0.1 / 0.3 / 0.0 | 1.5 / 1.6 / 0.3 | 7.7 / 52.4 / 2.4 |
| | | 4 | 5.3 / 5.3 / 0.0 | – / 3.6 / 0.0 | – / 41.1 / 2.1 | 0.1 / 0.3 / 0.0 | 1.4 / 1.5 / 0.3 | 6.8 / 51.7 / 2.4 |
| | | 8 | 4.6 / 4.6 / 0.0 | – / 3.6 / 0.0 | – / 41.4 / 2.7 | 0.1 / 0.3 / 0.0 | 1.3 / 1.4 / 0.3 | 6.0 / 51.3 / 3.0 |
| | $\tau = 2.5$ | 1 | 6.5 / 6.3 / 0.0 | – / 3.7 / 0.0 | – / 43.4 / 2.0 | 0.1 / 0.3 / 0.0 | 1.5 / 1.6 / 0.3 | 8.1 / 55.4 / 2.3 |
| | | 2 | 6.0 / 6.0 / 0.0 | – / 3.7 / 0.0 | – / 41.3 / 2.1 | 0.1 / 0.3 / 0.0 | 1.5 / 1.6 / 0.3 | 7.6 / 52.9 / 2.4 |
| | | 4 | 5.2 / 5.2 / 0.0 | – / 3.7 / 0.0 | – / 41.6 / 2.5 | 0.1 / 0.3 / 0.0 | 1.4 / 1.4 / 0.3 | 6.7 / 52.2 / 2.8 |
| | | 8 | 4.4 / 4.4 / 0.0 | – / 3.7 / 0.0 | – / 41.7 / 3.4 | 0.1 / 0.3 / 0.0 | 1.2 / 1.4 / 0.3 | 5.7 / 51.5 / 3.7 |
| | $\tau = 3.5$ | 1 | 5.9 / 5.8 / 0.0 | – / 3.7 / 0.0 | – / 43.7 / 1.8 | 0.1 / 0.3 / 0.0 | 1.4 / 1.5 / 0.3 | 7.4 / 55.0 / 2.1 |
| | | 2 | 5.5 / 5.5 / 0.0 | – / 3.7 / 0.0 | – / 41.5 / 1.8 | 0.1 / 0.3 / 0.0 | 1.4 / 1.5 / 0.3 | 7.0 / 52.5 / 2.1 |
| | | 4 | 5.0 / 4.9 / 0.0 | – / 3.7 / 0.0 | – / 41.7 / 2.3 | 0.1 / 0.3 / 0.0 | 1.3 / 1.4 / 0.3 | 6.4 / 52.0 / 2.6 |
| | | 8 | 4.2 / 4.2 / 0.0 | – / 3.7 / 0.0 | – / 41.7 / 3.2 | 0.1 / 0.3 / 0.0 | 1.2 / 1.3 / 0.3 | 5.4 / 51.1 / 3.5 |
| | $\tau = 4.5$ | 1 | 5.8 / 5.7 / 0.0 | – / 3.7 / 0.0 | – / 43.6 / 1.6 | 0.1 / 0.3 / 0.0 | 1.4 / 1.5 / 0.3 | 7.3 / 54.7 / 1.8 |
| | | 2 | 5.5 / 5.5 / 0.0 | – / 3.7 / 0.0 | – / 41.4 / 1.5 | 0.1 / 0.3 / 0.0 | 1.4 / 1.5 / 0.3 | 6.9 / 52.3 / 1.8 |
| | | 4 | 4.8 / 4.7 / 0.0 | – / 3.7 / 0.0 | – / 41.5 / 1.9 | 0.1 / 0.3 / 0.0 | 1.3 / 1.3 / 0.3 | 6.1 / 51.5 / 2.1 |
| | | 8 | 4.0 / 4.0 / 0.0 | – / 3.7 / 0.0 | – / 41.6 / 2.8 | 0.1 / 0.3 / 0.0 | 1.2 / 1.3 / 0.3 | 5.3 / 50.9 / 3.1 |
| NONLINEAR | LINEAR | 1 | 6.3 / 6.1 / 0.0 | – / 3.7 / 0.0 | – / 43.2 / 2.1 | 0.1 / 0.3 / 0.0 | 1.5 / 1.5 / 0.3 | 7.9 / 54.8 / 2.3 |
| | | 2 | 5.9 / 5.9 / 0.0 | – / 3.7 / 0.0 | – / 41.3 / 2.1 | 0.1 / 0.3 / 0.0 | 1.4 / 1.5 / 0.3 | 7.5 / 52.7 / 2.4 |
| | | 4 | 5.2 / 5.2 / 0.0 | – / 3.7 / 0.0 | – / 41.5 / 2.6 | 0.1 / 0.3 / 0.0 | 1.4 / 1.4 / 0.3 | 6.7 / 52.1 / 2.9 |
| | | 8 | 4.4 / 4.4 / 0.0 | – / 3.7 / 0.0 | – / 41.6 / 3.2 | 0.1 / 0.3 / 0.0 | 1.2 / 1.3 / 0.3 | 5.7 / 51.3 / 3.4 |
| | INTERACTION | 1 | 6.2 / 6.1 / 0.0 | – / 3.7 / 0.0 | – / 41.1 / 2.1 | 0.1 / 0.3 / 0.0 | 1.5 / 1.6 / 0.3 | 7.8 / 52.9 / 2.4 |
| | | 2 | 6.0 / 6.0 / 0.0 | – / 3.7 / 0.0 | – / 41.3 / 2.1 | 0.1 / 0.3 / 0.0 | 1.5 / 1.6 / 0.3 | 7.6 / 53.1 / 2.4 |
| | | 4 | 5.3 / 5.3 / 0.0 | – / 3.7 / 0.0 | – / 41.6 / 2.5 | 0.1 / 0.3 / 0.0 | 1.4 / 1.5 / 0.3 | 6.8 / 52.5 / 2.8 |
| | | 8 | 4.6 / 4.6 / 0.0 | – / 3.7 / 0.0 | – / 41.8 / 3.1 | 0.1 / 0.3 / 0.0 | 1.3 / 1.4 / 0.3 | 6.0 / 51.9 / 3.4 |
| | SINUSOID | 1 | 6.8 / 6.7 / 0.0 | – / 4.0 / 0.0 | – / 43.8 / 2.1 | 0.1 / 0.4 / 0.0 | 1.5 / 1.6 / 0.3 | 8.5 / 56.4 / 2.4 |
| | | 2 | 6.5 / 6.4 / 0.0 | – / 4.0 / 0.0 | – / 41.8 / 2.1 | 0.1 / 0.3 / 0.0 | 1.5 / 1.6 / 0.3 | 8.1 / 54.1 / 2.4 |
| | | 4 | 5.6 / 5.6 / 0.0 | – / 4.0 / 0.0 | – / 42.0 / 2.4 | 0.1 / 0.3 / 0.0 | 1.4 / 1.5 / 0.3 | 7.1 / 53.4 / 2.7 |
| | | 8 | 4.7 / 4.7 / 0.0 | – / 4.0 / 0.0 | – / 42.2 / 3.3 | 0.1 / 0.3 / 0.0 | 1.3 / 1.3 / 0.3 | 6.1 / 52.5 / 3.6 |
| | SOFTPLUS | 1 | 6.3 / 6.2 / 0.0 | – / 3.8 / 0.0 | – / 41.4 / 2.0 | 0.1 / 0.3 / 0.0 | 1.5 / 1.6 / 0.3 | 7.8 / 53.4 / 2.3 |
| | | 2 | 5.9 / 5.9 / 0.0 | – / 3.9 / 0.0 | – / 41.7 / 2.2 | 0.1 / 0.3 / 0.0 | 1.4 / 1.6 / 0.3 | 7.4 / 53.4 / 2.5 |
| | | 4 | 5.2 / 5.2 / 0.0 | – / 3.8 / 0.0 | – / 42.0 / 2.4 | 0.1 / 0.3 / 0.0 | 1.3 / 1.5 / 0.3 | 6.6 / 52.7 / 2.7 |
| | | 8 | 4.4 / 4.4 / 0.0 | – / 3.7 / 0.0 | – / 42.2 / 3.3 | 0.1 / 0.3 / 0.0 | 1.2 / 1.4 / 0.3 | 5.7 / 51.9 / 3.6 |

*Table 4.* Classification runtime breakdown when varying the balance ratio $\rho$. Each cell reports the mean runtime in seconds over 100 trials in the order BASELINE-AGG / MDCP / ORACLE; "–" indicates that the method does not use that stage. Details are otherwise the same as in Table 1.

| Suite | Setting | $K$ | Fit sources | Fit pooled | Learn $\lambda$ | Calibrate | Test eval | Total runtime |
|---|---|---|---|---|---|---|---|---|
| LINEAR | $\tau = 2.5$ | 2 | 5.1 / 5.0 / 0.0 | – / 3.3 / 0.0 | – / 40.5 / 1.5 | 0.1 / 0.2 / 0.0 | 1.0 / 1.1 / 0.2 | 6.2 / 50.1 / 1.7 |
| | | 3 | 5.5 / 5.5 / 0.0 | – / 3.4 / 0.0 | – / 40.6 / 2.0 | 0.1 / 0.3 / 0.0 | 1.4 / 1.5 / 0.3 | 7.0 / 51.2 / 2.3 |
| | | 5 | 5.9 / 5.8 / 0.0 | – / 3.4 / 0.0 | – / 42.3 / 3.3 | 0.1 / 0.6 / 0.0 | 2.1 / 2.2 / 0.4 | 8.1 / 54.3 / 3.7 |
| | | 10 | 6.3 / 6.3 / 0.0 | – / 3.4 / 0.0 | – / 44.4 / 9.5 | 0.2 / 1.8 / 0.0 | 3.6 / 3.7 / 0.8 | 10.1 / 59.6 / 10.3 |
| TEMPERATURE | $\tau = 0.5$ | 2 | 6.0 / 5.9 / 0.0 | – / 3.6 / 0.0 | – / 38.2 / 1.1 | 0.1 / 0.2 / 0.0 | 1.1 / 1.2 / 0.2 | 7.2 / 49.1 / 1.3 |
| | | 3 | 6.6 / 6.6 / 0.0 | – / 3.6 / 0.0 | – / 40.4 / 1.1 | 0.1 / 0.4 / 0.0 | 1.6 / 1.6 / 0.3 | 8.3 / 52.6 / 1.4 |
| | | 5 | 7.2 / 7.1 / 0.0 | – / 3.6 / 0.0 | – / 42.6 / 1.3 | 0.1 / 0.7 / 0.0 | 2.4 / 2.5 / 0.4 | 9.7 / 56.5 / 1.8 |
| | | 10 | 7.8 / 7.7 / 0.0 | – / 3.6 / 0.0 | – / 45.0 / 2.1 | 0.2 / 2.1 / 0.0 | 4.1 / 4.3 / 0.8 | 12.1 / 62.7 / 2.9 |
| | $\tau = 1.5$ | 2 | 5.8 / 5.7 / 0.0 | – / 3.6 / 0.0 | – / 40.1 / 1.5 | 0.1 / 0.2 / 0.0 | 1.1 / 1.1 / 0.2 | 7.0 / 50.7 / 1.7 |
| | | 3 | 6.4 / 6.4 / 0.0 | – / 3.6 / 0.0 | – / 40.7 / 1.9 | 0.1 / 0.3 / 0.0 | 1.5 / 1.6 / 0.3 | 8.0 / 52.6 / 2.2 |
| | | 5 | 6.8 / 6.7 / 0.0 | – / 3.6 / 0.0 | – / 42.7 / 2.9 | 0.1 / 0.7 / 0.0 | 2.3 / 2.4 / 0.4 | 9.2 / 56.1 / 3.3 |
| | | 10 | 7.2 / 7.1 / 0.0 | – / 3.6 / 0.0 | – / 45.1 / 6.5 | 0.2 / 1.9 / 0.0 | 3.9 / 4.1 / 0.8 | 11.3 / 61.8 / 7.3 |
| | $\tau = 2.5$ | 2 | 5.7 / 5.6 / 0.0 | – / 3.7 / 0.0 | – / 40.9 / 2.6 | 0.1 / 0.2 / 0.0 | 1.1 / 1.1 / 0.2 | 6.9 / 51.5 / 2.8 |
| | | 3 | 6.2 / 6.2 / 0.0 | – / 3.7 / 0.0 | – / 41.0 / 2.0 | 0.1 / 0.3 / 0.0 | 1.5 / 1.6 / 0.3 | 7.8 / 52.8 / 2.3 |
| | | 5 | 6.5 / 6.4 / 0.0 | – / 3.7 / 0.0 | – / 43.2 / 3.4 | 0.1 / 0.7 / 0.0 | 2.2 / 2.3 / 0.4 | 8.8 / 56.3 / 3.9 |
| | | 10 | 6.8 / 6.8 / 0.0 | – / 3.7 / 0.0 | – / 45.4 / 9.6 | 0.2 / 1.9 / 0.0 | 3.8 / 3.9 / 0.8 | 10.8 / 61.7 / 10.4 |
| | $\tau = 3.5$ | 2 | 5.5 / 5.4 / 0.0 | – / 3.7 / 0.0 | – / 41.1 / 1.3 | 0.1 / 0.2 / 0.0 | 1.0 / 1.1 / 0.2 | 6.6 / 51.4 / 1.5 |
| | | 3 | 5.8 / 5.8 / 0.0 | – / 3.7 / 0.0 | – / 41.2 / 1.8 | 0.1 / 0.3 / 0.0 | 1.4 / 1.5 / 0.3 | 7.4 / 52.5 / 2.1 |
| | | 5 | 6.3 / 6.2 / 0.0 | – / 3.7 / 0.0 | – / 43.1 / 2.9 | 0.1 / 0.6 / 0.0 | 2.2 / 2.2 / 0.4 | 8.6 / 55.9 / 3.4 |
| | | 10 | 6.4 / 6.4 / 0.0 | – / 3.7 / 0.0 | – / 45.3 / 10.1 | 0.2 / 1.8 / 0.0 | 3.6 / 3.7 / 0.8 | 10.2 / 60.9 / 11.0 |
| | $\tau = 4.5$ | 2 | 5.5 / 5.4 / 0.0 | – / 3.7 / 0.0 | – / 41.4 / 1.3 | 0.1 / 0.2 / 0.0 | 1.0 / 1.1 / 0.2 | 6.7 / 51.8 / 1.5 |
| | | 3 | 5.7 / 5.7 / 0.0 | – / 3.7 / 0.0 | – / 41.4 / 1.5 | 0.1 / 0.3 / 0.0 | 1.4 / 1.5 / 0.3 | 7.2 / 52.5 / 1.8 |
| | | 5 | 5.9 / 5.9 / 0.0 | – / 3.7 / 0.0 | – / 43.2 / 2.1 | 0.1 / 0.6 / 0.0 | 2.1 / 2.2 / 0.4 | 8.1 / 55.6 / 2.6 |
| | | 10 | 6.2 / 6.2 / 0.0 | – / 3.7 / 0.0 | – / 45.4 / 5.9 | 0.2 / 1.7 / 0.0 | 3.5 / 3.6 / 0.8 | 9.9 / 60.6 / 6.7 |
| NONLINEAR | LINEAR | 2 | 5.7 / 5.5 / 0.0 | – / 3.7 / 0.0 | – / 40.9 / 1.5 | 0.1 / 0.2 / 0.0 | 1.0 / 1.1 / 0.2 | 6.8 / 51.5 / 1.7 |
| | | 3 | 6.0 / 6.1 / 0.0 | – / 3.7 / 0.0 | – / 41.2 / 2.0 | 0.1 / 0.3 / 0.0 | 1.5 / 1.5 / 0.3 | 7.7 / 52.9 / 2.3 |
| | | 5 | 6.5 / 6.4 / 0.0 | – / 3.7 / 0.0 | – / 43.2 / 3.6 | 0.1 / 0.6 / 0.0 | 2.2 / 2.3 / 0.4 | 8.9 / 56.3 / 4.0 |
| | | 10 | 6.7 / 6.8 / 0.0 | – / 3.7 / 0.0 | – / 45.5 / 8.9 | 0.2 / 1.8 / 0.0 | 3.8 / 3.9 / 0.8 | 10.6 / 61.7 / 9.7 |
| | INTERACTION | 2 | 5.6 / 5.6 / 0.0 | – / 3.7 / 0.0 | – / 39.0 / 1.5 | 0.1 / 0.2 / 0.0 | 1.1 / 1.1 / 0.2 | 6.8 / 49.6 / 1.7 |
| | | 3 | 6.2 / 6.1 / 0.0 | – / 3.7 / 0.0 | – / 41.3 / 2.1 | 0.1 / 0.3 / 0.0 | 1.5 / 1.6 / 0.3 | 7.8 / 53.0 / 2.3 |
| | | 5 | 6.4 / 6.5 / 0.0 | – / 3.7 / 0.0 | – / 43.3 / 3.5 | 0.1 / 0.6 / 0.0 | 2.2 / 2.3 / 0.4 | 8.7 / 56.4 / 4.0 |
| | | 10 | 6.9 / 6.9 / 0.0 | – / 3.7 / 0.0 | – / 45.5 / 9.6 | 0.2 / 1.9 / 0.0 | 3.8 / 4.0 / 0.8 | 10.9 / 62.0 / 10.4 |
| | SINUSOID | 2 | 6.0 / 5.8 / 0.0 | – / 4.0 / 0.0 | – / 41.0 / 1.5 | 0.1 / 0.2 / 0.0 | 1.1 / 1.1 / 0.2 | 7.1 / 52.2 / 1.7 |
| | | 3 | 6.6 / 6.6 / 0.0 | – / 4.0 / 0.0 | – / 41.8 / 2.2 | 0.1 / 0.3 / 0.0 | 1.6 / 1.6 / 0.3 | 8.3 / 54.4 / 2.5 |
| | | 5 | 6.9 / 6.9 / 0.0 | – / 4.0 / 0.0 | – / 43.9 / 3.4 | 0.1 / 0.7 / 0.0 | 2.3 / 2.4 / 0.4 | 9.4 / 57.8 / 3.8 |
| | | 10 | 7.1 / 7.2 / 0.0 | – / 4.0 / 0.0 | – / 46.3 / 10.1 | 0.2 / 2.0 / 0.0 | 3.9 / 4.1 / 0.8 | 11.2 / 63.5 / 10.9 |
| | SOFTPLUS | 2 | 5.4 / 5.4 / 0.0 | – / 3.8 / 0.0 | – / 38.2 / 1.5 | 0.1 / 0.2 / 0.0 | 1.0 / 1.1 / 0.2 | 6.5 / 48.6 / 1.7 |
| | | 3 | 6.2 / 6.1 / 0.0 | – / 3.8 / 0.0 | – / 41.6 / 2.1 | 0.1 / 0.3 / 0.0 | 1.4 / 1.5 / 0.3 | 7.7 / 53.4 / 2.4 |
| | | 5 | 6.3 / 6.4 / 0.0 | – / 3.9 / 0.0 | – / 43.9 / 3.5 | 0.1 / 0.7 / 0.0 | 2.1 / 2.2 / 0.4 | 8.6 / 57.0 / 3.9 |
| | | 10 | 6.5 / 6.5 / 0.0 | – / 3.8 / 0.0 | – / 46.2 / 9.8 | 0.2 / 1.7 / 0.0 | 3.7 / 3.8 / 0.8 | 10.3 / 62.0 / 10.6 |

*Table 5.* Classification runtime breakdown when varying the number of sources $K$. Details are otherwise the same as in Table 4.

| Suite | Setting | $N$ | Fit sources | Fit pooled | Learn $\lambda$ | Calibrate | Test eval | Total runtime |
|---|---|---|---|---|---|---|---|---|
| LINEAR | $\tau = 2.5$ | 1500 | 1.6 / 1.5 / 0.0 | – / 1.7 / 0.0 | – / 15.6 / 1.1 | 0.0 / 0.1 / 0.0 | 0.3 / 0.3 / 0.1 | 2.0 / 19.3 / 1.2 |
| | | 3000 | 3.0 / 3.0 / 0.0 | – / 2.9 / 0.0 | – / 22.6 / 1.6 | 0.1 / 0.2 / 0.0 | 0.7 / 0.7 / 0.1 | 3.7 / 29.4 / 1.7 |
| | | 4500 | 4.3 / 4.3 / 0.0 | – / 3.2 / 0.0 | – / 31.7 / 1.9 | 0.1 / 0.3 / 0.0 | 1.0 / 1.2 / 0.2 | 5.4 / 40.6 / 2.1 |
| | | 6000 | 5.4 / 5.4 / 0.0 | – / 3.4 / 0.0 | – / 40.5 / 2.0 | 0.1 / 0.3 / 0.0 | 1.4 / 1.5 / 0.3 | 6.9 / 51.1 / 2.3 |
| TEMPERATURE | $\tau = 0.5$ | 1500 | 2.0 / 1.9 / 0.0 | – / 1.8 / 0.0 | – / 15.7 / 0.5 | 0.1 / 0.2 / 0.0 | 0.4 / 0.4 / 0.1 | 2.4 / 19.9 / 0.6 |
| | | 3000 | 3.6 / 3.6 / 0.0 | – / 3.1 / 0.0 | – / 22.5 / 0.8 | 0.1 / 0.2 / 0.0 | 0.8 / 0.8 / 0.1 | 4.5 / 30.3 / 0.9 |
| | | 4500 | 5.1 / 5.1 / 0.0 | – / 3.4 / 0.0 | – / 31.7 / 1.0 | 0.1 / 0.3 / 0.0 | 1.2 / 1.3 / 0.2 | 6.4 / 41.8 / 1.2 |
| | | 6000 | 6.6 / 6.6 / 0.0 | – / 3.6 / 0.0 | – / 40.3 / 1.1 | 0.1 / 0.3 / 0.0 | 1.6 / 1.7 / 0.3 | 8.3 / 52.5 / 1.4 |
| | $\tau = 1.5$ | 1500 | 1.9 / 1.8 / 0.0 | – / 1.8 / 0.0 | – / 15.5 / 0.9 | 0.1 / 0.2 / 0.0 | 0.3 / 0.4 / 0.1 | 2.3 / 19.6 / 1.0 |
| | | 3000 | 3.3 / 3.3 / 0.0 | – / 3.1 / 0.0 | – / 22.6 / 1.4 | 0.1 / 0.2 / 0.0 | 0.7 / 0.7 / 0.1 | 4.1 / 30.0 / 1.5 |
| | | 4500 | 4.9 / 4.9 / 0.0 | – / 3.4 / 0.0 | – / 31.8 / 1.7 | 0.1 / 0.3 / 0.0 | 1.1 / 1.2 / 0.2 | 6.1 / 41.6 / 1.9 |
| | | 6000 | 6.3 / 6.3 / 0.0 | – / 3.6 / 0.0 | – / 40.6 / 1.9 | 0.1 / 0.3 / 0.0 | 1.5 / 1.6 / 0.3 | 8.0 / 52.4 / 2.2 |
| | $\tau = 2.5$ | 1500 | 1.7 / 1.6 / 0.0 | – / 1.8 / 0.0 | – / 15.8 / 1.0 | 0.1 / 0.2 / 0.0 | 0.3 / 0.3 / 0.1 | 2.1 / 19.8 / 1.1 |
| | | 3000 | 3.3 / 3.3 / 0.0 | – / 3.2 / 0.0 | – / 23.1 / 1.6 | 0.1 / 0.2 / 0.0 | 0.7 / 0.7 / 0.1 | 4.1 / 30.6 / 1.7 |
| | | 4500 | 4.8 / 4.8 / 0.0 | – / 3.6 / 0.0 | – / 32.3 / 1.8 | 0.1 / 0.3 / 0.0 | 1.1 / 1.2 / 0.2 | 5.9 / 42.1 / 2.0 |
| | | 6000 | 6.3 / 6.3 / 0.0 | – / 3.7 / 0.0 | – / 41.2 / 2.0 | 0.1 / 0.3 / 0.0 | 1.5 / 1.6 / 0.3 | 7.9 / 53.1 / 2.3 |
| | $\tau = 3.5$ | 1500 | 1.7 / 1.6 / 0.0 | – / 1.8 / 0.0 | – / 15.8 / 1.1 | 0.1 / 0.2 / 0.0 | 0.3 / 0.3 / 0.1 | 2.1 / 19.8 / 1.1 |
| | | 3000 | 3.1 / 3.1 / 0.0 | – / 3.2 / 0.0 | – / 23.0 / 1.6 | 0.1 / 0.2 / 0.0 | 0.7 / 0.7 / 0.1 | 3.8 / 30.3 / 1.7 |
| | | 4500 | 4.5 / 4.5 / 0.0 | – / 3.6 / 0.0 | – / 32.4 / 1.6 | 0.1 / 0.3 / 0.0 | 1.0 / 1.2 / 0.2 | 5.7 / 42.0 / 1.8 |
| | | 6000 | 5.8 / 5.8 / 0.0 | – / 3.7 / 0.0 | – / 41.4 / 1.8 | 0.1 / 0.3 / 0.0 | 1.4 / 1.5 / 0.3 | 7.3 / 52.7 / 2.1 |
| | $\tau = 4.5$ | 1500 | 1.7 / 1.6 / 0.0 | – / 1.8 / 0.0 | – / 15.9 / 0.9 | 0.0 / 0.1 / 0.0 | 0.3 / 0.3 / 0.1 | 2.1 / 19.8 / 1.0 |
| | | 3000 | 2.9 / 3.0 / 0.0 | – / 3.2 / 0.0 | – / 23.1 / 1.1 | 0.1 / 0.2 / 0.0 | 0.6 / 0.7 / 0.1 | 3.7 / 30.2 / 1.3 |
| | | 4500 | 4.3 / 4.3 / 0.0 | – / 3.5 / 0.0 | – / 32.3 / 1.4 | 0.1 / 0.3 / 0.0 | 1.0 / 1.1 / 0.2 | 5.4 / 41.6 / 1.6 |
| | | 6000 | 5.8 / 5.7 / 0.0 | – / 3.6 / 0.0 | – / 41.5 / 1.5 | 0.1 / 0.3 / 0.0 | 1.4 / 1.5 / 0.3 | 7.3 / 52.6 / 1.8 |
| NONLINEAR | LINEAR | 1500 | 1.8 / 1.7 / 0.0 | – / 1.9 / 0.0 | – / 15.7 / 1.1 | 0.1 / 0.2 / 0.0 | 0.3 / 0.4 / 0.1 | 2.2 / 19.8 / 1.2 |
| | | 3000 | 3.2 / 3.2 / 0.0 | – / 3.2 / 0.0 | – / 22.9 / 1.7 | 0.1 / 0.2 / 0.0 | 0.7 / 0.7 / 0.1 | 3.9 / 30.2 / 1.8 |
| | | 4500 | 4.7 / 4.7 / 0.0 | – / 3.5 / 0.0 | – / 32.2 / 1.8 | 0.1 / 0.3 / 0.0 | 1.1 / 1.1 / 0.2 | 5.8 / 41.8 / 2.0 |
| | | 6000 | 6.1 / 6.1 / 0.0 | – / 3.7 / 0.0 | – / 41.1 / 2.1 | 0.1 / 0.3 / 0.0 | 1.5 / 1.5 / 0.3 | 7.7 / 52.8 / 2.4 |
| | INTERACTION | 1500 | 1.7 / 1.7 / 0.0 | – / 1.8 / 0.0 | – / 13.7 / 1.2 | 0.1 / 0.2 / 0.0 | 0.3 / 0.4 / 0.1 | 2.1 / 17.8 / 1.3 |
| | | 3000 | 3.3 / 3.3 / 0.0 | – / 3.2 / 0.0 | – / 23.0 / 1.6 | 0.1 / 0.2 / 0.0 | 0.7 / 0.7 / 0.1 | 4.1 / 30.5 / 1.8 |
| | | 4500 | 4.9 / 4.9 / 0.0 | – / 3.5 / 0.0 | – / 32.2 / 1.9 | 0.1 / 0.3 / 0.0 | 1.1 / 1.2 / 0.2 | 6.1 / 42.1 / 2.2 |
| | | 6000 | 6.2 / 6.2 / 0.0 | – / 3.7 / 0.0 | – / 41.2 / 2.0 | 0.1 / 0.3 / 0.0 | 1.5 / 1.6 / 0.3 | 7.8 / 53.0 / 2.3 |
| | SINUSOID | 1500 | 1.8 / 1.7 / 0.0 | – / 1.9 / 0.0 | – / 15.7 / 1.2 | 0.1 / 0.2 / 0.0 | 0.3 / 0.4 / 0.1 | 2.2 / 19.9 / 1.3 |
| | | 3000 | 3.3 / 3.4 / 0.0 | – / 3.5 / 0.0 | – / 23.2 / 1.7 | 0.1 / 0.2 / 0.0 | 0.7 / 0.7 / 0.1 | 4.1 / 31.0 / 1.9 |
| | | 4500 | 4.9 / 4.9 / 0.0 | – / 3.8 / 0.0 | – / 32.6 / 1.9 | 0.1 / 0.3 / 0.0 | 1.1 / 1.2 / 0.2 | 6.1 / 42.7 / 2.1 |
| | | 6000 | 6.6 / 6.6 / 0.0 | – / 4.0 / 0.0 | – / 41.7 / 2.1 | 0.1 / 0.3 / 0.0 | 1.5 / 1.6 / 0.3 | 8.2 / 54.2 / 2.4 |
| | SOFTPLUS | 1500 | 1.6 / 1.6 / 0.0 | – / 1.8 / 0.0 | – / 13.6 / 1.1 | 0.1 / 0.2 / 0.0 | 0.3 / 0.3 / 0.1 | 2.0 / 17.5 / 1.2 |
| | | 3000 | 3.2 / 3.3 / 0.0 | – / 3.3 / 0.0 | – / 23.1 / 1.6 | 0.1 / 0.2 / 0.0 | 0.7 / 0.7 / 0.1 | 4.0 / 30.6 / 1.7 |
| | | 4500 | 4.7 / 4.7 / 0.0 | – / 3.6 / 0.0 | – / 32.4 / 1.9 | 0.1 / 0.3 / 0.0 | 1.1 / 1.1 / 0.2 | 5.9 / 42.0 / 2.1 |
| | | 6000 | 6.2 / 6.2 / 0.0 | – / 3.9 / 0.0 | – / 41.7 / 2.0 | 0.1 / 0.3 / 0.0 | 1.5 / 1.6 / 0.3 | 7.8 / 53.7 / 2.3 |

*Table 6.* Classification runtime breakdown when varying the total sample size $N$. Details are otherwise the same as in Table 4.

| Suite | Setting | $\rho$ | Overall coverage | Worst-case coverage | Avg interval width | Total runtime |
|---|---|---|---|---|---|---|
| LINEAR | $\tau = 2.5$ | 1 | 0.973 / 0.927 / 0.920 | 0.961 / 0.903 / 0.905 | 7.50 / 5.87 / 4.79 | 2.3 / 9.2 / 1.8 |
| | | 2 | 0.973 / 0.930 / 0.921 | 0.959 / 0.903 / 0.907 | 7.85 / 6.23 / 5.01 | 2.1 / 7.3 / 1.9 |
| | | 4 | 0.975 / 0.931 / 0.924 | 0.959 / 0.909 / 0.909 | 7.92 / 6.19 / 4.91 | 1.9 / 6.6 / 2.2 |
| | | 8 | 0.977 / 0.933 / 0.923 | 0.960 / 0.908 / 0.906 | 8.63 / 6.64 / 5.01 | 1.7 / 7.5 / 2.7 |
| TEMPERATURE | $\tau = 0.5$ | 1 | 0.988 / 0.925 / 0.922 | 0.981 / 0.901 / 0.904 | 5.35 / 3.38 / 2.78 | 2.4 / 9.0 / 1.4 |
| | | 2 | 0.988 / 0.928 / 0.927 | 0.981 / 0.906 / 0.907 | 5.55 / 3.50 / 2.85 | 2.2 / 7.2 / 1.5 |
| | | 4 | 0.988 / 0.925 / 0.924 | 0.981 / 0.904 / 0.906 | 5.77 / 3.51 / 2.83 | 2.0 / 6.9 / 1.5 |
| | | 8 | 0.991 / 0.932 / 0.928 | 0.985 / 0.909 / 0.906 | 6.61 / 3.78 / 2.89 | 1.7 / 6.0 / 1.7 |
| | $\tau = 1.5$ | 1 | 0.979 / 0.931 / 0.922 | 0.964 / 0.901 / 0.905 | 6.14 / 4.56 / 3.75 | 2.5 / 9.9 / 1.8 |
| | | 2 | 0.978 / 0.931 / 0.926 | 0.965 / 0.902 / 0.907 | 6.48 / 4.77 / 3.90 | 2.2 / 7.3 / 1.9 |
| | | 4 | 0.979 / 0.930 / 0.923 | 0.966 / 0.901 / 0.906 | 6.65 / 4.71 / 3.81 | 1.9 / 7.3 / 2.2 |
| | | 8 | 0.985 / 0.934 / 0.924 | 0.972 / 0.906 / 0.904 | 7.53 / 4.99 / 3.84 | 1.7 / 7.4 / 2.7 |
| | $\tau = 2.5$ | 1 | 0.970 / 0.930 / 0.922 | 0.952 / 0.901 / 0.905 | 7.30 / 5.94 / 4.76 | 2.4 / 10.7 / 1.8 |
| | | 2 | 0.970 / 0.931 / 0.925 | 0.953 / 0.905 / 0.906 | 7.73 / 6.26 / 4.97 | 2.2 / 7.6 / 2.0 |
| | | 4 | 0.970 / 0.930 / 0.921 | 0.953 / 0.905 / 0.904 | 7.81 / 6.12 / 4.83 | 1.9 / 7.5 / 2.4 |
| | | 8 | 0.980 / 0.935 / 0.924 | 0.963 / 0.905 / 0.904 | 8.94 / 6.46 / 4.95 | 1.7 / 7.3 / 3.0 |
| | $\tau = 3.5$ | 1 | 0.960 / 0.931 / 0.922 | 0.940 / 0.904 / 0.902 | 8.57 / 7.41 / 5.73 | 2.4 / 10.3 / 1.8 |
| | | 2 | 0.964 / 0.932 / 0.924 | 0.945 / 0.906 / 0.903 | 9.21 / 7.86 / 6.01 | 2.2 / 7.8 / 2.0 |
| | | 4 | 0.965 / 0.933 / 0.921 | 0.946 / 0.906 / 0.905 | 9.27 / 7.60 / 5.83 | 2.0 / 7.5 / 2.5 |
| | | 8 | 0.974 / 0.937 / 0.924 | 0.954 / 0.904 / 0.904 | 10.47 / 8.01 / 6.06 | 1.8 / 6.6 / 3.0 |
| | $\tau = 4.5$ | 1 | 0.956 / 0.929 / 0.922 | 0.935 / 0.901 / 0.901 | 10.20 / 8.99 / 6.90 | 2.5 / 11.1 / 1.8 |
| | | 2 | 0.962 / 0.931 / 0.924 | 0.942 / 0.904 / 0.904 | 11.28 / 9.63 / 7.34 | 2.3 / 8.3 / 1.9 |
| | | 4 | 0.961 / 0.930 / 0.924 | 0.941 / 0.902 / 0.906 | 11.23 / 9.26 / 7.13 | 2.0 / 8.2 / 2.4 |
| | | 8 | 0.974 / 0.937 / 0.926 | 0.954 / 0.906 / 0.905 | 13.00 / 10.01 / 7.43 | 1.8 / 7.6 / 2.7 |
| NONLINEAR | LINEAR | 1 | 0.971 / 0.926 / 0.917 | 0.959 / 0.901 / 0.904 | 7.54 / 6.01 / 4.89 | 2.3 / 9.1 / 1.8 |
| | | 2 | 0.975 / 0.927 / 0.919 | 0.961 / 0.902 / 0.905 | 7.85 / 6.04 / 4.92 | 2.1 / 7.2 / 1.8 |
| | | 4 | 0.976 / 0.929 / 0.919 | 0.962 / 0.905 / 0.906 | 8.05 / 6.24 / 4.92 | 2.0 / 6.6 / 2.0 |
| | | 8 | 0.980 / 0.928 / 0.925 | 0.965 / 0.905 / 0.908 | 8.81 / 6.43 / 4.92 | 1.7 / 7.3 / 2.7 |
| | INTERACTION | 1 | 0.988 / 0.921 / 0.923 | 0.983 / 0.903 / 0.906 | 22.16 / 13.55 / 9.88 | 2.2 / 5.3 / 1.6 |
| | | 2 | 0.989 / 0.921 / 0.920 | 0.985 / 0.905 / 0.904 | 22.86 / 13.55 / 9.67 | 2.1 / 4.8 / 1.6 |
| | | 4 | 0.990 / 0.922 / 0.923 | 0.985 / 0.904 / 0.908 | 24.60 / 14.20 / 9.73 | 1.9 / 4.7 / 1.6 |
| | | 8 | 0.992 / 0.930 / 0.928 | 0.987 / 0.912 / 0.909 | 29.15 / 15.73 / 9.92 | 1.7 / 4.9 / 1.8 |
| | SINUSOID | 1 | 0.976 / 0.931 / 0.919 | 0.964 / 0.904 / 0.905 | 9.16 / 6.99 / 5.66 | 2.6 / 9.0 / 1.7 |
| | | 2 | 0.978 / 0.926 / 0.920 | 0.968 / 0.902 / 0.907 | 9.20 / 6.75 / 5.51 | 2.4 / 7.0 / 1.7 |
| | | 4 | 0.979 / 0.923 / 0.924 | 0.967 / 0.901 / 0.909 | 9.37 / 6.79 / 5.49 | 2.2 / 6.6 / 1.9 |
| | | 8 | 0.983 / 0.933 / 0.922 | 0.968 / 0.909 / 0.909 | 10.46 / 7.48 / 5.68 | 1.9 / 7.0 / 2.4 |
| | SOFTPLUS | 1 | 0.975 / 0.926 / 0.919 | 0.963 / 0.901 / 0.906 | 9.15 / 6.90 / 5.67 | 2.5 / 6.7 / 1.6 |
| | | 2 | 0.979 / 0.927 / 0.920 | 0.967 / 0.904 / 0.906 | 9.15 / 6.75 / 5.52 | 2.4 / 6.5 / 1.7 |
| | | 4 | 0.979 / 0.928 / 0.921 | 0.966 / 0.905 / 0.907 | 9.43 / 6.87 / 5.48 | 2.1 / 6.8 / 2.0 |
| | | 8 | 0.982 / 0.931 / 0.924 | 0.967 / 0.907 / 0.910 | 10.49 / 7.38 / 5.69 | 1.9 / 7.0 / 2.5 |

*Table 7.* Regression results when varying the balance ratio $\rho$. Each metric cell reports the mean over 100 trials in the order BASELINE-AGG / MDCP / ORACLE. Runtime values are reported in seconds. Except for the varied parameters, the LINEAR suite setup follows that of Figure 5; the NONLINEAR suite setup follows that of Figure 6; and the TEMPERATURE suite setup follows that of Figure 7.

| Suite | Setting | $K$ | Overall coverage | Worst-case coverage | Avg interval width | Total runtime |
|---|---|---|---|---|---|---|
| LINEAR | $\tau = 2.5$ | 2 | 0.950 / 0.913 / 0.912 | 0.941 / 0.902 / 0.904 | 6.03 / 5.18 / 4.14 | 1.9 / 8.0 / 1.1 |
| | | 3 | 0.973 / 0.927 / 0.920 | 0.961 / 0.903 / 0.905 | 7.50 / 5.87 / 4.79 | 2.2 / 7.1 / 1.8 |
| | | 5 | 0.989 / 0.943 / 0.930 | 0.974 / 0.902 / 0.906 | 9.76 / 6.94 / 5.87 | 2.7 / 7.7 / 3.1 |
| | | 10 | 0.998 / 0.968 / 0.956 | 0.988 / 0.909 / 0.918 | 13.72 / 8.49 / 7.29 | 3.9 / 9.5 / 8.0 |
| TEMPERATURE | $\tau = 0.5$ | 2 | 0.970 / 0.913 / 0.913 | 0.964 / 0.901 / 0.903 | 4.47 / 3.27 / 2.66 | 1.9 / 8.5 / 1.2 |
| | | 3 | 0.988 / 0.925 / 0.922 | 0.981 / 0.901 / 0.904 | 5.35 / 3.38 / 2.78 | 2.3 / 6.9 / 1.5 |
| | | 5 | 0.997 / 0.940 / 0.938 | 0.993 / 0.909 / 0.910 | 6.91 / 3.61 / 3.03 | 2.7 / 8.2 / 1.7 |
| | | 10 | 1.000 / 0.959 / 0.955 | 0.998 / 0.920 / 0.914 | 10.20 / 4.03 / 3.30 | 4.0 / 9.3 / 3.8 |
| | $\tau = 1.5$ | 2 | 0.957 / 0.914 / 0.914 | 0.947 / 0.899 / 0.905 | 5.10 / 4.12 / 3.34 | 1.9 / 8.6 / 1.3 |
| | | 3 | 0.979 / 0.931 / 0.922 | 0.964 / 0.901 / 0.905 | 6.14 / 4.56 / 3.75 | 2.2 / 7.6 / 1.8 |
| | | 5 | 0.993 / 0.948 / 0.938 | 0.983 / 0.908 / 0.910 | 8.09 / 5.17 / 4.37 | 2.7 / 8.1 / 2.9 |
| | | 10 | 0.999 / 0.966 / 0.958 | 0.994 / 0.907 / 0.912 | 11.41 / 6.01 / 5.14 | 4.0 / 10.2 / 8.1 |
| | $\tau = 2.5$ | 2 | 0.946 / 0.916 / 0.912 | 0.935 / 0.901 / 0.903 | 5.90 / 5.12 / 4.02 | 1.9 / 9.7 / 1.3 |
| | | 3 | 0.970 / 0.930 / 0.922 | 0.952 / 0.901 / 0.905 | 7.30 / 5.94 / 4.76 | 2.3 / 8.5 / 1.8 |
| | | 5 | 0.989 / 0.948 / 0.938 | 0.973 / 0.906 / 0.912 | 9.89 / 7.01 / 5.90 | 2.8 / 8.2 / 3.1 |
| | | 10 | 0.997 / 0.970 / 0.960 | 0.986 / 0.907 / 0.918 | 13.78 / 8.45 / 7.25 | 4.0 / 9.9 / 9.5 |
| | $\tau = 3.5$ | 2 | 0.939 / 0.918 / 0.912 | 0.928 / 0.902 / 0.902 | 6.81 / 6.14 / 4.67 | 2.0 / 8.9 / 1.4 |
| | | 3 | 0.960 / 0.931 / 0.922 | 0.940 / 0.904 / 0.902 | 8.57 / 7.41 / 5.73 | 2.4 / 8.2 / 1.8 |
| | | 5 | 0.985 / 0.949 / 0.937 | 0.965 / 0.907 / 0.910 | 11.96 / 9.14 / 7.41 | 2.8 / 8.9 / 3.1 |
| | | 10 | 0.997 / 0.970 / 0.960 | 0.983 / 0.906 / 0.916 | 16.67 / 11.00 / 9.37 | 4.2 / 10.4 / 9.7 |
| | $\tau = 4.5$ | 2 | 0.937 / 0.918 / 0.912 | 0.925 / 0.901 / 0.901 | 8.23 / 7.50 / 5.63 | 2.0 / 10.3 / 1.4 |
| | | 3 | 0.956 / 0.929 / 0.922 | 0.935 / 0.901 / 0.901 | 10.20 / 8.99 / 6.90 | 2.4 / 8.9 / 1.8 |
| | | 5 | 0.984 / 0.949 / 0.937 | 0.964 / 0.907 / 0.909 | 14.80 / 11.33 / 9.20 | 2.8 / 8.8 / 2.8 |
| | | 10 | 0.996 / 0.970 / 0.960 | 0.981 / 0.907 / 0.914 | 20.57 / 13.71 / 11.67 | 4.1 / 10.5 / 9.1 |
| NONLINEAR | LINEAR | 2 | 0.950 / 0.915 / 0.912 | 0.941 / 0.902 / 0.904 | 5.97 / 5.12 / 4.07 | 1.9 / 8.4 / 1.2 |
| | | 3 | 0.971 / 0.926 / 0.917 | 0.959 / 0.901 / 0.904 | 7.54 / 6.01 / 4.89 | 2.2 / 7.2 / 1.8 |
| | | 5 | 0.989 / 0.947 / 0.933 | 0.975 / 0.908 / 0.910 | 9.71 / 6.99 / 5.93 | 2.8 / 7.6 / 3.0 |
| | | 10 | 0.998 / 0.968 / 0.956 | 0.989 / 0.902 / 0.917 | 13.98 / 8.59 / 7.39 | 3.8 / 9.4 / 7.4 |
| | INTERACTION | 2 | 0.969 / 0.911 / 0.907 | 0.965 / 0.902 / 0.897 | 17.32 / 12.61 / 9.02 | 1.8 / 4.3 / 1.5 |
| | | 3 | 0.988 / 0.921 / 0.923 | 0.983 / 0.903 / 0.906 | 22.16 / 13.55 / 9.88 | 2.2 / 5.3 / 1.6 |
| | | 5 | 0.997 / 0.936 / 0.936 | 0.993 / 0.910 / 0.909 | 28.82 / 14.67 / 10.55 | 2.7 / 6.0 / 1.9 |
| | | 10 | 1.000 / 0.952 / 0.958 | 0.997 / 0.919 / 0.920 | 46.04 / 17.66 / 11.94 | 3.8 / 7.4 / 2.5 |
| | SINUSOID | 2 | 0.952 / 0.913 / 0.907 | 0.945 / 0.901 / 0.899 | 7.32 / 6.10 / 4.85 | 2.2 / 7.6 / 1.2 |
| | | 3 | 0.976 / 0.931 / 0.919 | 0.964 / 0.904 / 0.905 | 9.16 / 6.99 / 5.66 | 2.5 / 7.0 / 1.7 |
| | | 5 | 0.991 / 0.946 / 0.934 | 0.979 / 0.905 / 0.910 | 11.50 / 7.80 / 6.58 | 3.1 / 7.6 / 2.7 |
| | | 10 | 0.999 / 0.969 / 0.959 | 0.993 / 0.916 / 0.921 | 16.66 / 9.28 / 7.92 | 4.2 / 10.1 / 6.1 |
| | SOFTPLUS | 2 | 0.953 / 0.912 / 0.907 | 0.944 / 0.899 / 0.898 | 7.30 / 6.08 / 4.85 | 2.0 / 6.6 / 1.2 |
| | | 3 | 0.975 / 0.926 / 0.919 | 0.963 / 0.901 / 0.906 | 9.15 / 6.90 / 5.67 | 2.5 / 6.7 / 1.6 |
| | | 5 | 0.990 / 0.946 / 0.936 | 0.977 / 0.906 / 0.912 | 11.31 / 7.72 / 6.52 | 3.0 / 7.7 / 2.7 |
| | | 10 | 0.999 / 0.967 / 0.959 | 0.993 / 0.909 / 0.919 | 16.60 / 9.22 / 7.90 | 4.2 / 9.5 / 6.2 |

*Table 8.* Regression scaling results when varying the number of sources $K$. Details are otherwise the same as in Table 7.

| Suite | Setting | $N$ | Overall coverage | Worst-case coverage | Avg interval width | Total runtime |
|---|---|---|---|---|---|---|
| LINEAR | $\tau = 2.5$ | 1500 | 0.979 / 0.940 / 0.932 | 0.966 / 0.913 / 0.909 | 9.62 / 6.76 / 4.88 | 0.8 / 5.3 / 0.8 |
| | | 3000 | 0.976 / 0.933 / 0.926 | 0.962 / 0.906 / 0.909 | 8.49 / 6.36 / 4.98 | 1.3 / 4.5 / 1.3 |
| | | 4500 | 0.974 / 0.930 / 0.923 | 0.962 / 0.906 / 0.908 | 7.88 / 6.03 / 4.85 | 1.7 / 6.1 / 1.6 |
| | | 6000 | 0.973 / 0.927 / 0.920 | 0.961 / 0.903 / 0.905 | 7.50 / 5.87 / 4.79 | 2.2 / 7.0 / 1.8 |
| TEMPERATURE | $\tau = 0.5$ | 1500 | 0.992 / 0.932 / 0.933 | 0.986 / 0.910 / 0.909 | 7.60 / 4.02 / 2.93 | 0.8 / 5.8 / 0.7 |
| | | 3000 | 0.990 / 0.926 / 0.929 | 0.984 / 0.903 / 0.908 | 6.22 / 3.60 / 2.86 | 1.4 / 5.3 / 1.1 |
| | | 4500 | 0.989 / 0.927 / 0.924 | 0.983 / 0.905 / 0.906 | 5.77 / 3.50 / 2.81 | 1.8 / 6.1 / 1.3 |
| | | 6000 | 0.988 / 0.925 / 0.922 | 0.981 / 0.901 / 0.904 | 5.35 / 3.38 / 2.78 | 2.3 / 7.0 / 1.4 |
| | $\tau = 1.5$ | 1500 | 0.985 / 0.936 / 0.938 | 0.973 / 0.907 / 0.914 | 8.40 / 5.24 / 3.95 | 0.8 / 5.3 / 0.9 |
| | | 3000 | 0.980 / 0.933 / 0.927 | 0.968 / 0.905 / 0.906 | 7.09 / 4.90 / 3.86 | 1.4 / 5.3 / 1.4 |
| | | 4500 | 0.981 / 0.932 / 0.923 | 0.970 / 0.906 / 0.906 | 6.67 / 4.76 / 3.84 | 1.8 / 6.0 / 1.6 |
| | | 6000 | 0.979 / 0.931 / 0.922 | 0.964 / 0.901 / 0.905 | 6.14 / 4.56 / 3.75 | 2.3 / 7.7 / 1.8 |
| | $\tau = 2.5$ | 1500 | 0.980 / 0.940 / 0.937 | 0.965 / 0.909 / 0.913 | 9.75 / 6.85 / 5.02 | 0.8 / 5.1 / 1.0 |
| | | 3000 | 0.974 / 0.933 / 0.925 | 0.959 / 0.908 / 0.904 | 8.36 / 6.37 / 4.89 | 1.4 / 5.1 / 1.5 |
| | | 4500 | 0.971 / 0.930 / 0.924 | 0.956 / 0.901 / 0.907 | 7.90 / 6.19 / 4.92 | 1.8 / 6.7 / 1.7 |
| | | 6000 | 0.970 / 0.930 / 0.922 | 0.952 / 0.901 / 0.905 | 7.30 / 5.94 / 4.76 | 2.3 / 8.8 / 1.8 |
| | $\tau = 3.5$ | 1500 | 0.973 / 0.937 / 0.940 | 0.956 / 0.905 / 0.916 | 11.32 / 8.40 / 6.10 | 0.9 / 5.0 / 1.0 |
| | | 3000 | 0.969 / 0.935 / 0.927 | 0.950 / 0.906 / 0.904 | 9.94 / 7.93 / 5.94 | 1.4 / 5.5 / 1.5 |
| | | 4500 | 0.965 / 0.930 / 0.924 | 0.947 / 0.902 / 0.906 | 9.35 / 7.64 / 5.96 | 1.9 / 7.4 / 1.7 |
| | | 6000 | 0.960 / 0.931 / 0.922 | 0.940 / 0.904 / 0.902 | 8.57 / 7.41 / 5.73 | 2.4 / 8.2 / 1.8 |
| | $\tau = 4.5$ | 1500 | 0.972 / 0.937 / 0.940 | 0.953 / 0.906 / 0.917 | 13.89 / 10.32 / 7.39 | 0.9 / 5.2 / 1.0 |
| | | 3000 | 0.967 / 0.935 / 0.927 | 0.947 / 0.906 / 0.904 | 12.04 / 9.66 / 7.21 | 1.4 / 5.5 / 1.5 |
| | | 4500 | 0.964 / 0.931 / 0.924 | 0.945 / 0.903 / 0.903 | 11.33 / 9.54 / 7.24 | 1.9 / 6.9 / 1.6 |
| | | 6000 | 0.956 / 0.929 / 0.922 | 0.935 / 0.901 / 0.901 | 10.20 / 8.99 / 6.90 | 2.3 / 8.9 / 1.8 |
| NONLINEAR | LINEAR | 1500 | 0.981 / 0.939 / 0.931 | 0.968 / 0.913 / 0.911 | 9.55 / 6.79 / 4.90 | 0.8 / 5.0 / 0.8 |
| | | 3000 | 0.978 / 0.930 / 0.923 | 0.965 / 0.904 / 0.906 | 8.54 / 6.32 / 4.94 | 1.3 / 5.0 / 1.2 |
| | | 4500 | 0.976 / 0.929 / 0.923 | 0.964 / 0.904 / 0.908 | 7.95 / 6.02 / 4.89 | 1.7 / 5.7 / 1.5 |
| | | 6000 | 0.971 / 0.926 / 0.917 | 0.959 / 0.901 / 0.904 | 7.54 / 6.01 / 4.89 | 2.2 / 7.3 / 1.8 |
| | INTERACTION | 1500 | 0.991 / 0.934 / 0.932 | 0.985 / 0.917 / 0.910 | 34.35 / 18.35 / 10.02 | 0.9 / 1.9 / 0.8 |
| | | 3000 | 0.991 / 0.924 / 0.928 | 0.986 / 0.909 / 0.909 | 27.28 / 15.10 / 9.78 | 1.3 / 3.2 / 1.1 |
| | | 4500 | 0.989 / 0.924 / 0.923 | 0.984 / 0.906 / 0.906 | 24.03 / 14.18 / 9.79 | 1.7 / 3.8 / 1.3 |
| | | 6000 | 0.988 / 0.921 / 0.923 | 0.983 / 0.903 / 0.906 | 22.16 / 13.55 / 9.88 | 2.2 / 5.3 / 1.6 |
| | SINUSOID | 1500 | 0.983 / 0.941 / 0.932 | 0.972 / 0.915 / 0.909 | 11.89 / 8.03 / 5.84 | 0.9 / 4.8 / 0.8 |
| | | 3000 | 0.981 / 0.931 / 0.926 | 0.969 / 0.905 / 0.909 | 10.00 / 7.14 / 5.59 | 1.5 / 4.3 / 1.2 |
| | | 4500 | 0.980 / 0.930 / 0.922 | 0.968 / 0.905 / 0.906 | 9.59 / 7.03 / 5.64 | 2.0 / 5.7 / 1.4 |
| | | 6000 | 0.976 / 0.931 / 0.919 | 0.964 / 0.904 / 0.905 | 9.16 / 6.99 / 5.66 | 2.5 / 7.1 / 1.7 |
| | SOFTPLUS | 1500 | 0.982 / 0.937 / 0.931 | 0.970 / 0.910 / 0.909 | 12.25 / 7.97 / 5.75 | 1.0 / 2.7 / 0.8 |
| | | 3000 | 0.981 / 0.933 / 0.926 | 0.970 / 0.908 / 0.908 | 10.09 / 7.11 / 5.54 | 1.5 / 5.2 / 1.2 |
| | | 4500 | 0.979 / 0.931 / 0.921 | 0.967 / 0.906 / 0.906 | 9.53 / 7.01 / 5.60 | 2.0 / 6.5 / 1.3 |
| | | 6000 | 0.975 / 0.926 / 0.919 | 0.963 / 0.901 / 0.906 | 9.15 / 6.90 / 5.67 | 2.5 / 6.6 / 1.6 |

*Table 9.* Regression scaling results when varying the total sample size $N$. Details are otherwise the same as in Table 7.

| Suite | Setting | $\rho$ | Fit sources | Fit pooled | Learn $\lambda$ | Calibrate | Test eval | Total runtime |
|---|---|---|---|---|---|---|---|---|
| LINEAR | $\tau = 2.5$ | 1 | 1.9 / 1.8 / 0.0 | – / 0.8 / 0.0 | – / 6.0 / 1.4 | 0.0 / 0.0 / 0.0 | 0.4 / 0.5 / 0.4 | 2.3 / 9.2 / 1.8 |
| | | 2 | 1.8 / 1.7 / 0.0 | – / 0.8 / 0.0 | – / 4.2 / 1.5 | 0.0 / 0.0 / 0.0 | 0.3 / 0.5 / 0.4 | 2.1 / 7.3 / 1.9 |
| | | 4 | 1.5 / 1.5 / 0.0 | – / 0.8 / 0.0 | – / 3.8 / 1.8 | 0.0 / 0.0 / 0.0 | 0.3 / 0.5 / 0.4 | 1.9 / 6.6 / 2.2 |
| | | 8 | 1.3 / 1.3 / 0.0 | – / 0.8 / 0.0 | – / 4.9 / 2.3 | 0.0 / 0.0 / 0.0 | 0.3 / 0.5 / 0.4 | 1.7 / 7.5 / 2.7 |
| TEMPERATURE | $\tau = 0.5$ | 1 | 2.0 / 1.9 / 0.0 | – / 0.8 / 0.0 | – / 5.8 / 1.0 | 0.0 / 0.0 / 0.0 | 0.4 / 0.5 / 0.4 | 2.4 / 9.0 / 1.4 |
| | | 2 | 1.8 / 1.8 / 0.0 | – / 0.8 / 0.0 | – / 4.1 / 1.1 | 0.0 / 0.0 / 0.0 | 0.3 / 0.5 / 0.4 | 2.2 / 7.2 / 1.5 |
| | | 4 | 1.6 / 1.6 / 0.0 | – / 0.8 / 0.0 | – / 4.0 / 1.2 | 0.0 / 0.0 / 0.0 | 0.3 / 0.5 / 0.4 | 2.0 / 6.9 / 1.5 |
| | | 8 | 1.4 / 1.4 / 0.0 | – / 0.8 / 0.0 | – / 3.3 / 1.3 | 0.0 / 0.0 / 0.0 | 0.3 / 0.5 / 0.4 | 1.7 / 6.0 / 1.7 |
| | $\tau = 1.5$ | 1 | 2.1 / 1.9 / 0.0 | – / 0.8 / 0.0 | – / 6.5 / 1.4 | 0.0 / 0.1 / 0.0 | 0.4 / 0.5 / 0.4 | 2.5 / 9.9 / 1.8 |
| | | 2 | 1.8 / 1.8 / 0.0 | – / 0.8 / 0.0 | – / 4.1 / 1.6 | 0.0 / 0.0 / 0.0 | 0.3 / 0.5 / 0.4 | 2.2 / 7.3 / 1.9 |
| | | 4 | 1.6 / 1.6 / 0.0 | – / 0.8 / 0.0 | – / 4.4 / 1.8 | 0.0 / 0.0 / 0.0 | 0.3 / 0.5 / 0.4 | 1.9 / 7.3 / 2.2 |
| | | 8 | 1.4 / 1.4 / 0.0 | – / 0.9 / 0.0 | – / 4.7 / 2.3 | 0.0 / 0.0 / 0.0 | 0.3 / 0.5 / 0.4 | 1.7 / 7.4 / 2.7 |
| | $\tau = 2.5$ | 1 | 2.0 / 1.9 / 0.0 | – / 0.8 / 0.0 | – / 7.4 / 1.4 | 0.0 / 0.0 / 0.0 | 0.4 / 0.5 / 0.4 | 2.4 / 10.7 / 1.8 |
| | | 2 | 1.8 / 1.8 / 0.0 | – / 0.8 / 0.0 | – / 4.4 / 1.6 | 0.0 / 0.0 / 0.0 | 0.3 / 0.5 / 0.4 | 2.2 / 7.6 / 2.0 |
| | | 4 | 1.6 / 1.6 / 0.0 | – / 0.8 / 0.0 | – / 4.5 / 2.0 | 0.0 / 0.0 / 0.0 | 0.3 / 0.5 / 0.4 | 1.9 / 7.5 / 2.4 |
| | | 8 | 1.4 / 1.4 / 0.0 | – / 0.8 / 0.0 | – / 4.5 / 2.6 | 0.0 / 0.0 / 0.0 | 0.3 / 0.5 / 0.4 | 1.7 / 7.3 / 3.0 |
| | $\tau = 3.5$ | 1 | 2.0 / 2.0 / 0.0 | – / 0.9 / 0.0 | – / 6.9 / 1.4 | 0.0 / 0.0 / 0.0 | 0.4 / 0.5 / 0.4 | 2.4 / 10.3 / 1.8 |
| | | 2 | 1.9 / 1.9 / 0.0 | – / 0.9 / 0.0 | – / 4.5 / 1.6 | 0.0 / 0.0 / 0.0 | 0.3 / 0.5 / 0.4 | 2.2 / 7.8 / 2.0 |
| | | 4 | 1.7 / 1.6 / 0.0 | – / 0.9 / 0.0 | – / 4.5 / 2.1 | 0.0 / 0.0 / 0.0 | 0.3 / 0.5 / 0.4 | 2.0 / 7.5 / 2.5 |
| | | 8 | 1.5 / 1.4 / 0.0 | – / 0.9 / 0.0 | – / 3.8 / 2.6 | 0.0 / 0.0 / 0.0 | 0.3 / 0.5 / 0.4 | 1.8 / 6.6 / 3.0 |
| | $\tau = 4.5$ | 1 | 2.0 / 2.0 / 0.0 | – / 0.9 / 0.0 | – / 7.7 / 1.3 | 0.0 / 0.0 / 0.0 | 0.4 / 0.5 / 0.4 | 2.5 / 11.1 / 1.8 |
| | | 2 | 1.9 / 1.9 / 0.0 | – / 0.9 / 0.0 | – / 5.0 / 1.5 | 0.0 / 0.0 / 0.0 | 0.3 / 0.5 / 0.4 | 2.3 / 8.3 / 1.9 |
| | | 4 | 1.7 / 1.6 / 0.0 | – / 0.9 / 0.0 | – / 5.2 / 2.0 | 0.0 / 0.0 / 0.0 | 0.3 / 0.5 / 0.4 | 2.0 / 8.2 / 2.4 |
| | | 8 | 1.4 / 1.4 / 0.0 | – / 0.9 / 0.0 | – / 4.8 / 2.3 | 0.0 / 0.0 / 0.0 | 0.3 / 0.5 / 0.4 | 1.8 / 7.6 / 2.7 |
| NONLINEAR | LINEAR | 1 | 1.9 / 1.8 / 0.0 | – / 0.8 / 0.0 | – / 6.0 / 1.4 | 0.0 / 0.0 / 0.0 | 0.4 / 0.5 / 0.4 | 2.3 / 9.1 / 1.8 |
| | | 2 | 1.8 / 1.7 / 0.0 | – / 0.8 / 0.0 | – / 4.2 / 1.4 | 0.0 / 0.0 / 0.0 | 0.3 / 0.5 / 0.4 | 2.1 / 7.2 / 1.8 |
| | | 4 | 1.6 / 1.5 / 0.0 | – / 0.8 / 0.0 | – / 3.7 / 1.6 | 0.0 / 0.0 / 0.0 | 0.4 / 0.5 / 0.4 | 2.0 / 6.6 / 2.0 |
| | | 8 | 1.3 / 1.3 / 0.0 | – / 0.8 / 0.0 | – / 4.6 / 2.3 | 0.0 / 0.0 / 0.0 | 0.3 / 0.5 / 0.4 | 1.7 / 7.3 / 2.7 |
| | INTERACTION | 1 | 1.9 / 1.8 / 0.0 | – / 0.8 / 0.0 | – / 2.2 / 1.3 | 0.0 / 0.0 / 0.0 | 0.3 / 0.5 / 0.4 | 2.2 / 5.3 / 1.6 |
| | | 2 | 1.8 / 1.7 / 0.0 | – / 0.8 / 0.0 | – / 1.8 / 1.3 | 0.0 / 0.0 / 0.0 | 0.3 / 0.5 / 0.4 | 2.1 / 4.8 / 1.6 |
| | | 4 | 1.5 / 1.5 / 0.0 | – / 0.8 / 0.0 | – / 1.9 / 1.3 | 0.0 / 0.0 / 0.0 | 0.3 / 0.5 / 0.4 | 1.9 / 4.7 / 1.6 |
| | | 8 | 1.4 / 1.3 / 0.0 | – / 0.8 / 0.0 | – / 2.3 / 1.4 | 0.0 / 0.0 / 0.0 | 0.3 / 0.5 / 0.4 | 1.7 / 4.9 / 1.8 |
| | SINUSOID | 1 | 2.2 / 2.1 / 0.0 | – / 0.9 / 0.0 | – / 5.3 / 1.3 | 0.0 / 0.1 / 0.0 | 0.4 / 0.5 / 0.4 | 2.6 / 9.0 / 1.7 |
| | | 2 | 2.1 / 2.0 / 0.0 | – / 0.9 / 0.0 | – / 3.5 / 1.3 | 0.0 / 0.0 / 0.0 | 0.3 / 0.5 / 0.4 | 2.4 / 7.0 / 1.7 |
| | | 4 | 1.8 / 1.8 / 0.0 | – / 0.9 / 0.0 | – / 3.4 / 1.5 | 0.0 / 0.0 / 0.0 | 0.4 / 0.5 / 0.4 | 2.2 / 6.6 / 1.9 |
| | | 8 | 1.6 / 1.5 / 0.0 | – / 0.9 / 0.0 | – / 4.0 / 2.0 | 0.0 / 0.0 / 0.0 | 0.3 / 0.5 / 0.4 | 1.9 / 7.0 / 2.4 |
| | SOFTPLUS | 1 | 2.2 / 2.1 / 0.0 | – / 0.9 / 0.0 | – / 3.0 / 1.2 | 0.0 / 0.0 / 0.0 | 0.3 / 0.5 / 0.4 | 2.5 / 6.7 / 1.6 |
| | | 2 | 2.1 / 2.0 / 0.0 | – / 0.9 / 0.0 | – / 3.0 / 1.3 | 0.0 / 0.0 / 0.0 | 0.3 / 0.5 / 0.4 | 2.4 / 6.5 / 1.7 |
| | | 4 | 1.8 / 1.8 / 0.0 | – / 0.9 / 0.0 | – / 3.6 / 1.6 | 0.0 / 0.0 / 0.0 | 0.3 / 0.5 / 0.4 | 2.1 / 6.8 / 2.0 |
| | | 8 | 1.6 / 1.5 / 0.0 | – / 1.0 / 0.0 | – / 4.0 / 2.1 | 0.0 / 0.0 / 0.0 | 0.3 / 0.5 / 0.5 | 1.9 / 7.0 / 2.5 |

*Table 10.* Regression runtime breakdown when varying the balance ratio $\rho$. Each cell reports the mean runtime in seconds over 100 trials in the order BASELINE-AGG / MDCP / ORACLE; "–" indicates that the method does not use that stage. Details are otherwise the same as in Table 7.

| Suite | Setting | $K$ | Fit sources | Fit pooled | Learn $\lambda$ | Calibrate | Test eval | Total runtime |
|---|---|---|---|---|---|---|---|---|
| LINEAR | $\tau = 2.5$ | 2 | 1.6 / 1.5 / 0.0 | − / 0.8 / 0.0 | − / 5.2 / 0.8 | 0.0 / 0.0 / 0.0 | 0.3 / 0.5 / 0.3 | 1.9 / 8.0 / 1.1 |
| | | 3 | 1.8 / 1.8 / 0.0 | − / 0.8 / 0.0 | − / 3.9 / 1.4 | 0.0 / 0.0 / 0.0 | 0.3 / 0.5 / 0.4 | 2.2 / 7.1 / 1.8 |
| | | 5 | 2.1 / 2.1 / 0.0 | − / 0.8 / 0.0 | − / 4.0 / 2.5 | 0.0 / 0.1 / 0.0 | 0.6 / 0.7 / 0.6 | 2.7 / 7.7 / 3.1 |
| | | 10 | 2.6 / 2.6 / 0.0 | − / 0.8 / 0.0 | − / 4.6 / 7.0 | 0.0 / 0.1 / 0.0 | 1.3 / 1.3 / 1.0 | 3.9 / 9.5 / 8.0 |
| TEMPERATURE | $\tau = 0.5$ | 2 | 1.6 / 1.6 / 0.0 | − / 0.8 / 0.0 | − / 5.5 / 0.9 | 0.0 / 0.0 / 0.0 | 0.3 / 0.5 / 0.3 | 1.9 / 8.5 / 1.2 |
| | | 3 | 2.0 / 1.9 / 0.0 | − / 0.8 / 0.0 | − / 3.7 / 1.1 | 0.0 / 0.0 / 0.0 | 0.3 / 0.5 / 0.4 | 2.3 / 6.9 / 1.5 |
| | | 5 | 2.2 / 2.2 / 0.0 | − / 0.9 / 0.0 | − / 4.4 / 1.2 | 0.0 / 0.1 / 0.0 | 0.5 / 0.7 / 0.5 | 2.7 / 8.2 / 1.7 |
| | | 10 | 2.7 / 2.8 / 0.0 | − / 0.9 / 0.0 | − / 4.2 / 2.9 | 0.0 / 0.1 / 0.0 | 1.2 / 1.3 / 1.0 | 4.0 / 9.3 / 3.8 |
| | $\tau = 1.5$ | 2 | 1.6 / 1.6 / 0.0 | − / 0.8 / 0.0 | − / 5.7 / 1.0 | 0.0 / 0.0 / 0.0 | 0.3 / 0.5 / 0.3 | 1.9 / 8.6 / 1.3 |
| | | 3 | 1.9 / 1.9 / 0.0 | − / 0.8 / 0.0 | − / 4.4 / 1.4 | 0.0 / 0.0 / 0.0 | 0.3 / 0.5 / 0.4 | 2.2 / 7.6 / 1.8 |
| | | 5 | 2.2 / 2.1 / 0.0 | − / 0.8 / 0.0 | − / 4.3 / 2.3 | 0.0 / 0.1 / 0.0 | 0.5 / 0.7 / 0.6 | 2.7 / 8.1 / 2.9 |
| | | 10 | 2.7 / 2.7 / 0.0 | − / 0.8 / 0.0 | − / 5.2 / 7.1 | 0.0 / 0.2 / 0.0 | 1.3 / 1.3 / 1.0 | 4.0 / 10.2 / 8.1 |
| | $\tau = 2.5$ | 2 | 1.6 / 1.6 / 0.0 | − / 0.8 / 0.0 | − / 6.7 / 1.0 | 0.0 / 0.0 / 0.0 | 0.3 / 0.5 / 0.3 | 1.9 / 9.7 / 1.3 |
| | | 3 | 2.0 / 1.9 / 0.0 | − / 0.8 / 0.0 | − / 5.3 / 1.4 | 0.0 / 0.0 / 0.0 | 0.3 / 0.5 / 0.4 | 2.3 / 8.5 / 1.8 |
| | | 5 | 2.2 / 2.2 / 0.0 | − / 0.8 / 0.0 | − / 4.4 / 2.5 | 0.0 / 0.1 / 0.0 | 0.6 / 0.7 / 0.6 | 2.8 / 8.2 / 3.1 |
| | | 10 | 2.7 / 2.7 / 0.0 | − / 0.8 / 0.0 | − / 4.9 / 8.5 | 0.0 / 0.1 / 0.0 | 1.3 / 1.3 / 1.0 | 4.0 / 9.9 / 9.5 |
| | $\tau = 3.5$ | 2 | 1.7 / 1.6 / 0.0 | − / 0.9 / 0.0 | − / 5.9 / 1.1 | 0.0 / 0.0 / 0.0 | 0.3 / 0.5 / 0.3 | 2.0 / 8.9 / 1.4 |
| | | 3 | 2.0 / 2.0 / 0.0 | − / 0.9 / 0.0 | − / 4.8 / 1.4 | 0.0 / 0.0 / 0.0 | 0.3 / 0.5 / 0.4 | 2.4 / 8.2 / 1.8 |
| | | 5 | 2.3 / 2.2 / 0.0 | − / 0.9 / 0.0 | − / 5.0 / 2.5 | 0.0 / 0.1 / 0.0 | 0.6 / 0.8 / 0.6 | 2.8 / 8.9 / 3.1 |
| | | 10 | 2.8 / 2.8 / 0.0 | − / 0.9 / 0.0 | − / 5.3 / 8.7 | 0.0 / 0.2 / 0.0 | 1.3 / 1.3 / 1.0 | 4.2 / 10.4 / 9.7 |
| | $\tau = 4.5$ | 2 | 1.7 / 1.6 / 0.0 | − / 0.9 / 0.0 | − / 7.2 / 1.1 | 0.0 / 0.0 / 0.0 | 0.3 / 0.5 / 0.3 | 2.0 / 10.3 / 1.4 |
| | | 3 | 2.0 / 2.0 / 0.0 | − / 0.9 / 0.0 | − / 5.5 / 1.3 | 0.0 / 0.0 / 0.0 | 0.3 / 0.5 / 0.4 | 2.4 / 8.9 / 1.8 |
| | | 5 | 2.3 / 2.2 / 0.0 | − / 0.9 / 0.0 | − / 4.9 / 2.2 | 0.0 / 0.1 / 0.0 | 0.6 / 0.8 / 0.6 | 2.8 / 8.8 / 2.8 |
| | | 10 | 2.8 / 2.8 / 0.0 | − / 0.9 / 0.0 | − / 5.4 / 8.0 | 0.0 / 0.1 / 0.0 | 1.3 / 1.3 / 1.0 | 4.1 / 10.5 / 9.1 |
| NONLINEAR | LINEAR | 2 | 1.6 / 1.5 / 0.0 | − / 0.8 / 0.0 | − / 5.6 / 0.9 | 0.0 / 0.0 / 0.0 | 0.3 / 0.5 / 0.3 | 1.9 / 8.4 / 1.2 |
| | | 3 | 1.9 / 1.8 / 0.0 | − / 0.8 / 0.0 | − / 4.1 / 1.4 | 0.0 / 0.0 / 0.0 | 0.3 / 0.5 / 0.4 | 2.2 / 7.2 / 1.8 |
| | | 5 | 2.1 / 2.1 / 0.0 | − / 0.8 / 0.0 | − / 3.9 / 2.4 | 0.0 / 0.1 / 0.0 | 0.7 / 0.7 / 0.6 | 2.8 / 7.6 / 3.0 |
| | | 10 | 2.6 / 2.7 / 0.0 | − / 0.8 / 0.0 | − / 4.4 / 6.4 | 0.0 / 0.1 / 0.0 | 1.2 / 1.3 / 1.0 | 3.8 / 9.4 / 7.4 |
| | INTERACTION | 2 | 1.6 / 1.5 / 0.0 | − / 0.8 / 0.0 | − / 1.6 / 1.2 | 0.0 / 0.0 / 0.0 | 0.2 / 0.4 / 0.3 | 1.8 / 4.3 / 1.5 |
| | | 3 | 1.9 / 1.8 / 0.0 | − / 0.8 / 0.0 | − / 2.2 / 1.3 | 0.0 / 0.0 / 0.0 | 0.3 / 0.5 / 0.4 | 2.2 / 5.3 / 1.6 |
| | | 5 | 2.1 / 2.1 / 0.0 | − / 0.8 / 0.0 | − / 2.3 / 1.4 | 0.0 / 0.1 / 0.0 | 0.5 / 0.7 / 0.5 | 2.7 / 6.0 / 1.9 |
| | | 10 | 2.6 / 2.7 / 0.0 | − / 0.8 / 0.0 | − / 2.6 / 1.6 | 0.0 / 0.1 / 0.0 | 1.1 / 1.2 / 0.9 | 3.8 / 7.4 / 2.5 |
| | SINUSOID | 2 | 1.9 / 1.8 / 0.0 | − / 1.0 / 0.0 | − / 4.3 / 0.9 | 0.0 / 0.0 / 0.0 | 0.3 / 0.4 / 0.3 | 2.2 / 7.6 / 1.2 |
| | | 3 | 2.2 / 2.1 / 0.0 | − / 0.9 / 0.0 | − / 3.4 / 1.3 | 0.0 / 0.0 / 0.0 | 0.3 / 0.5 / 0.4 | 2.5 / 7.0 / 1.7 |
| | | 5 | 2.4 / 2.4 / 0.0 | − / 0.9 / 0.0 | − / 3.4 / 2.1 | 0.0 / 0.1 / 0.0 | 0.7 / 0.7 / 0.6 | 3.1 / 7.6 / 2.7 |
| | | 10 | 3.0 / 3.1 / 0.0 | − / 1.0 / 0.0 | − / 4.6 / 5.1 | 0.0 / 0.2 / 0.0 | 1.2 / 1.3 / 1.0 | 4.2 / 10.1 / 6.1 |
| | SOFTPLUS | 2 | 1.8 / 1.8 / 0.0 | − / 0.9 / 0.0 | − / 3.4 / 0.9 | 0.0 / 0.0 / 0.0 | 0.2 / 0.4 / 0.3 | 2.0 / 6.6 / 1.2 |
| | | 3 | 2.2 / 2.2 / 0.0 | − / 0.9 / 0.0 | − / 3.0 / 1.2 | 0.0 / 0.0 / 0.0 | 0.3 / 0.5 / 0.4 | 2.5 / 6.7 / 1.6 |
| | | 5 | 2.4 / 2.4 / 0.0 | − / 0.9 / 0.0 | − / 3.5 / 2.1 | 0.0 / 0.1 / 0.0 | 0.6 / 0.7 / 0.6 | 3.0 / 7.7 / 2.7 |
| | | 10 | 3.0 / 3.0 / 0.0 | − / 1.0 / 0.0 | − / 4.1 / 5.2 | 0.0 / 0.2 / 0.0 | 1.2 / 1.3 / 1.0 | 4.2 / 9.5 / 6.2 |

*Table 11.* Regression runtime breakdown when varying the number of sources $K$. Details are otherwise the same as in Table 10.

| Suite | Setting | $N$ | Fit sources | Fit pooled | Learn $\lambda$ | Calibrate | Test eval | Total runtime |
|---|---|---|---|---|---|---|---|---|
| LINEAR | $\tau = 2.5$ | 1500 | 0.7 / 0.7 / 0.0 | − / 0.6 / 0.0 | − / 3.9 / 0.7 | 0.0 / 0.0 / 0.0 | 0.1 / 0.1 / 0.1 | 0.8 / 5.3 / 0.8 |
| | | 3000 | 1.2 / 1.1 / 0.0 | − / 0.8 / 0.0 | − / 2.3 / 1.1 | 0.0 / 0.0 / 0.0 | 0.2 / 0.3 / 0.2 | 1.3 / 4.5 / 1.3 |
| | | 4500 | 1.5 / 1.4 / 0.0 | − / 0.8 / 0.0 | − / 3.5 / 1.3 | 0.0 / 0.0 / 0.0 | 0.2 / 0.4 / 0.3 | 1.7 / 6.1 / 1.6 |
| | | 6000 | 1.8 / 1.8 / 0.0 | − / 0.8 / 0.0 | − / 3.9 / 1.4 | 0.0 / 0.0 / 0.0 | 0.3 / 0.5 / 0.4 | 2.2 / 7.0 / 1.8 |
| TEMPERATURE | $\tau = 0.5$ | 1500 | 0.7 / 0.7 / 0.0 | − / 0.6 / 0.0 | − / 4.3 / 0.6 | 0.0 / 0.0 / 0.0 | 0.1 / 0.1 / 0.1 | 0.8 / 5.8 / 0.7 |
| | | 3000 | 1.2 / 1.1 / 0.0 | − / 1.3 / 0.0 | − / 2.6 / 1.0 | 0.0 / 0.0 / 0.0 | 0.2 / 0.2 / 0.2 | 1.4 / 5.3 / 1.1 |
| | | 4500 | 1.6 / 1.5 / 0.0 | − / 0.8 / 0.0 | − / 3.3 / 1.0 | 0.0 / 0.0 / 0.0 | 0.2 / 0.4 / 0.3 | 1.8 / 6.1 / 1.3 |
| | | 6000 | 1.9 / 1.9 / 0.0 | − / 0.9 / 0.0 | − / 3.7 / 1.0 | 0.0 / 0.0 / 0.0 | 0.3 / 0.5 / 0.4 | 2.3 / 7.0 / 1.4 |
| | $\tau = 1.5$ | 1500 | 0.8 / 0.7 / 0.0 | − / 0.6 / 0.0 | − / 3.8 / 0.8 | 0.0 / 0.0 / 0.0 | 0.1 / 0.1 / 0.1 | 0.8 / 5.3 / 0.9 |
| | | 3000 | 1.2 / 1.1 / 0.0 | − / 0.9 / 0.0 | − / 3.0 / 1.2 | 0.0 / 0.0 / 0.0 | 0.2 / 0.3 / 0.2 | 1.4 / 5.3 / 1.4 |
| | | 4500 | 1.5 / 1.5 / 0.0 | − / 0.8 / 0.0 | − / 3.2 / 1.3 | 0.0 / 0.0 / 0.0 | 0.2 / 0.4 / 0.3 | 1.8 / 6.0 / 1.6 |
| | | 6000 | 1.9 / 1.9 / 0.0 | − / 0.8 / 0.0 | − / 4.4 / 1.4 | 0.0 / 0.0 / 0.0 | 0.3 / 0.5 / 0.4 | 2.3 / 7.7 / 1.8 |
| | $\tau = 2.5$ | 1500 | 0.7 / 0.7 / 0.0 | − / 0.6 / 0.0 | − / 3.6 / 0.9 | 0.0 / 0.0 / 0.0 | 0.1 / 0.1 / 0.1 | 0.8 / 5.1 / 1.0 |
| | | 3000 | 1.2 / 1.1 / 0.0 | − / 0.9 / 0.0 | − / 2.9 / 1.3 | 0.0 / 0.0 / 0.0 | 0.2 / 0.3 / 0.2 | 1.4 / 5.1 / 1.5 |
| | | 4500 | 1.6 / 1.5 / 0.0 | − / 0.8 / 0.0 | − / 4.0 / 1.4 | 0.0 / 0.0 / 0.0 | 0.2 / 0.4 / 0.3 | 1.8 / 6.7 / 1.7 |
| | | 6000 | 1.9 / 2.1 / 0.0 | − / 0.8 / 0.0 | − / 5.3 / 1.4 | 0.0 / 0.0 / 0.0 | 0.3 / 0.5 / 0.4 | 2.3 / 8.8 / 1.8 |
| | $\tau = 3.5$ | 1500 | 0.8 / 0.8 / 0.0 | − / 0.6 / 0.0 | − / 3.5 / 0.9 | 0.0 / 0.0 / 0.0 | 0.1 / 0.1 / 0.1 | 0.9 / 5.0 / 1.0 |
| | | 3000 | 1.3 / 1.2 / 0.0 | − / 0.9 / 0.0 | − / 3.1 / 1.3 | 0.0 / 0.0 / 0.0 | 0.2 / 0.3 / 0.2 | 1.4 / 5.5 / 1.5 |
| | | 4500 | 1.6 / 1.6 / 0.0 | − / 0.9 / 0.0 | − / 4.5 / 1.4 | 0.0 / 0.0 / 0.0 | 0.3 / 0.4 / 0.3 | 1.9 / 7.4 / 1.7 |
| | | 6000 | 2.0 / 2.0 / 0.0 | − / 0.9 / 0.0 | − / 4.8 / 1.4 | 0.0 / 0.0 / 0.0 | 0.3 / 0.5 / 0.4 | 2.4 / 8.2 / 1.8 |
| | $\tau = 4.5$ | 1500 | 0.8 / 0.8 / 0.0 | − / 0.6 / 0.0 | − / 3.7 / 0.9 | 0.0 / 0.0 / 0.0 | 0.1 / 0.1 / 0.1 | 0.9 / 5.2 / 1.0 |
| | | 3000 | 1.3 / 1.2 / 0.0 | − / 0.9 / 0.0 | − / 3.2 / 1.3 | 0.0 / 0.0 / 0.0 | 0.2 / 0.3 / 0.2 | 1.4 / 5.5 / 1.5 |
| | | 4500 | 1.6 / 1.6 / 0.0 | − / 0.8 / 0.0 | − / 4.1 / 1.3 | 0.0 / 0.0 / 0.0 | 0.3 / 0.4 / 0.3 | 1.9 / 6.9 / 1.6 |
| | | 6000 | 2.0 / 2.0 / 0.0 | − / 0.9 / 0.0 | − / 5.5 / 1.3 | 0.0 / 0.0 / 0.0 | 0.3 / 0.5 / 0.4 | 2.3 / 8.9 / 1.8 |
| NONLINEAR | LINEAR | 1500 | 0.7 / 0.8 / 0.0 | − / 0.5 / 0.0 | − / 3.6 / 0.7 | 0.0 / 0.0 / 0.0 | 0.1 / 0.1 / 0.1 | 0.8 / 5.0 / 0.8 |
| | | 3000 | 1.1 / 1.1 / 0.0 | − / 0.8 / 0.0 | − / 2.8 / 1.0 | 0.0 / 0.0 / 0.0 | 0.2 / 0.3 / 0.2 | 1.3 / 5.0 / 1.2 |
| | | 4500 | 1.5 / 1.5 / 0.0 | − / 0.8 / 0.0 | − / 3.0 / 1.2 | 0.0 / 0.0 / 0.0 | 0.2 / 0.4 / 0.3 | 1.7 / 5.7 / 1.5 |
| | | 6000 | 1.8 / 1.8 / 0.0 | − / 0.8 / 0.0 | − / 4.1 / 1.4 | 0.0 / 0.0 / 0.0 | 0.3 / 0.6 / 0.4 | 2.2 / 7.3 / 1.8 |
| | INTERACTION | 1500 | 0.8 / 0.7 / 0.0 | − / 0.5 / 0.0 | − / 0.5 / 0.7 | 0.0 / 0.0 / 0.0 | 0.1 / 0.1 / 0.1 | 0.9 / 1.9 / 0.8 |
| | | 3000 | 1.1 / 1.1 / 0.0 | − / 0.8 / 0.0 | − / 1.1 / 0.9 | 0.0 / 0.0 / 0.0 | 0.2 / 0.2 / 0.2 | 1.3 / 3.2 / 1.1 |
| | | 4500 | 1.5 / 1.5 / 0.0 | − / 0.8 / 0.0 | − / 1.2 / 1.0 | 0.0 / 0.0 / 0.0 | 0.2 / 0.4 / 0.3 | 1.7 / 3.8 / 1.3 |
| | | 6000 | 1.9 / 1.8 / 0.0 | − / 0.8 / 0.0 | − / 2.2 / 1.3 | 0.0 / 0.0 / 0.0 | 0.3 / 0.5 / 0.4 | 2.2 / 5.3 / 1.6 |
| | SINUSOID | 1500 | 0.8 / 0.9 / 0.0 | − / 0.6 / 0.0 | − / 3.2 / 0.7 | 0.0 / 0.0 / 0.0 | 0.1 / 0.1 / 0.1 | 0.9 / 4.8 / 0.8 |
| | | 3000 | 1.3 / 1.3 / 0.0 | − / 0.9 / 0.0 | − / 1.9 / 1.0 | 0.0 / 0.0 / 0.0 | 0.2 / 0.3 / 0.2 | 1.5 / 4.3 / 1.2 |
| | | 4500 | 1.7 / 1.8 / 0.0 | − / 0.9 / 0.0 | − / 2.5 / 1.1 | 0.0 / 0.0 / 0.0 | 0.2 / 0.4 / 0.3 | 2.0 / 5.7 / 1.4 |
| | | 6000 | 2.2 / 2.1 / 0.0 | − / 1.0 / 0.0 | − / 3.4 / 1.3 | 0.0 / 0.0 / 0.0 | 0.3 / 0.6 / 0.4 | 2.5 / 7.1 / 1.7 |
| | SOFTPLUS | 1500 | 0.9 / 0.8 / 0.0 | − / 0.6 / 0.0 | − / 1.2 / 0.7 | 0.0 / 0.0 / 0.0 | 0.1 / 0.1 / 0.1 | 1.0 / 2.7 / 0.8 |
| | | 3000 | 1.3 / 1.3 / 0.0 | − / 0.9 / 0.0 | − / 2.7 / 1.0 | 0.0 / 0.0 / 0.0 | 0.2 / 0.3 / 0.2 | 1.5 / 5.2 / 1.2 |
| | | 4500 | 1.7 / 1.7 / 0.0 | − / 0.9 / 0.0 | − / 3.4 / 1.0 | 0.0 / 0.1 / 0.0 | 0.2 / 0.4 / 0.3 | 2.0 / 6.5 / 1.3 |
| | | 6000 | 2.2 / 2.1 / 0.0 | − / 0.9 / 0.0 | − / 3.0 / 1.2 | 0.0 / 0.0 / 0.0 | 0.3 / 0.5 / 0.4 | 2.5 / 6.6 / 1.6 |

*Table 12.* Regression runtime breakdown when varying the total sample size $N$. Details are otherwise the same as in Table 10.

