# OpenReview forum: "Multi-Distribution Robust Conformal Prediction"
_ICML.cc/2026/Conference — ICML 2026 regular_

### Official Review · Reviewer_4NE6 · 2026-03-02

**Soundness:** 3
**Presentation:** 2
**Significance:** 3
**Originality:** 3
**Overall Recommendation:** 5
**Confidence:** 4

**Summary:**

This work studies conformal prediction when the test point is drawn from a mixture of heterogeneous source distributions, with the specific challenge that the mixture component is unknown at test time. The objective is therefore to guarantee the desired coverage uniformly across all source distributions.

The authors first introduce max-p aggregation, a intuitive approach that takes the union of conformal sets constructed independently on each source dataset. While this approach ensures uniform coverage, it may lead to overly conservative prediction sets. To address this issue, the main contribution of the paper is a theoretically motivated conformity score designed to minimize the size of the prediction set under a uniform coverage constraint.

The authors derive a closed-form expression for this score function, which depends on theoretical quantities that must be estimated in practice. Under the assumption that used estimators are asymptotically consistent, they establish asymptotic optimality of the method in terms of difference with the optimal prediction set. The paper then describes how these quantities can be estimated in practice and provides empirical results on both synthetic and real-world datasets.

**Compliance With Llm Reviewing Policy:**

Affirmed.

**Final Justification:**

See Rebuttal Acknowledgement

**Key Questions For Authors:**

### Questions

1. **Consistency of conditional distribution estimates.**
   Do the proposed estimators of the conditional distribution of $Y \mid X = x$ satisfy asymptotic consistency under the considered setting? The size of the individual source datasets seems particularly important for this property, yet its impact is not thoroughly discussed. Additional experiments with heterogeneous and progressively increasing source sample sizes would help assess the sensitivity of the method to the quality of the training step and clarify how quickly the asymptotic regime is approached in practice.

2. **Clarification on the inclusion of $T(x)$.**
   Could the authors provide more details on the methodology used to incorporate the set $T(x)$ at the end of page 4? In particular, what would be the impact of setting $U_k = 0$ (respectively $U_k = 1$) for all $k$ in Theorem 3? A brief discussion of these extreme cases would help clarify the role of these quantities in the construction.

3. **Oracle baseline for synthetic experiments.**
   For the synthetic experiments, it would be helpful to include an additional oracle baseline with access to the true conditional distributions. This would allow readers to better assess how close the proposed method is to the optimal prediction set and quantify the gap induced by estimation error.

**Limitations:**

Yes

**Strengths And Weaknesses:**

## Strengths

This paper proposes an original solution to the problem of achieving uniform coverage over heterogeneous source datasets in conformal prediction. This setting is particularly relevant in applications related to fairness, where guarantees must hold uniformly across groups. The assumption that the test distribution component is unknown at inference time is both realistic and well motivated.

The derivation of the closed-form expression for the prediction set minimizing size under a uniform coverage constraint is elegant and ultimately quite intuitive. The theoretical development is substantial, and the empirical evaluation on both synthetic and real-world datasets supports the practical relevance of the approach.

## Weaknesses

- **Clarity and presentation.** The paper is dense and sometimes difficult to follow. Theorem statements are lengthy and could benefit from clearer structuring or intermediate explanations. For instance, despite the length of Theorem 2 and the accompanying text, I had to consult the proof to fully understand the result. Improving exposition would significantly enhance accessibility.

- **Related work on efficiency-based conformal prediction.** The discussion of related work is incomplete with respect to efficiency-oriented conformal prediction. In particular, *Distribution-free prediction bands for non-parametric regression* by Lei & Wasserman (2014) is closely related in spirit, as it studies size-optimal prediction sets (albeit in a single-source setting). The authors should also consider citing:
1. *Efficient and Differentiable Conformal Prediction with General Function Classes* (Bai et al., ICLR 2022),
2. *On Volume Minimization in Conformal Regression* (Le Bars & Humbert, ICML 2025).

These works are directly connected to the objective of minimizing prediction set size under coverage constraints.

- **Strong asymptotic assumptions.** The theoretical guarantees rely on asymptotic consistency of the estimated quantities entering the conformity score. This is a strong assumption, and it is unclear how realistic the corresponding regime is in practice. The paper provides estimators but does not sufficiently discuss the conditions under which asymptotic optimality can reasonably be expected.


Typos:
- line 076-077 (left column): $\mathbb{E}\leftarrow \mathbb{P}$
- Eq. (2): $K\leftarrow k$
- Proof of Theorem 1: $y \leftarrow Y_{n+1}$
- Eq. (10): In the integral $X \leftarrow x$

---

> ### Author Rebuttal · Authors · 2026-03-30
>
> We thank reviewer 4NE6 for appreciating the originality, practical relevance, theoretical development, and empirical demonstrations.
>
> 1. **Clarity of Thm. 2**
>
> We agree that the presentation of Thm. 2 could be improved. The optimal score is obtained by solving Eq. 4, and $\lambda^\*_k(x)$ are the Lagrange multipliers in the dual problem. The complementary slackness part is intended to describe the tightness of per-source coverage. We have revised Thm. 2 to (i) introduce the dual problem, (ii) state the optimal solution to Eq. 4 in terms of optimal $\lambda_k^\*(x)$ to the dual, and (iii) directly state the tightness result via $\lambda_k^*(x)$. We also streamlined Thm. 3, deferring the (technically more tedious) $\rho(T)>0$ to a remark.
>
> 2. **Literature on efficient CP**
>
> Thank you for highlighting this literature. We’ve added the papers mentioned and expanded the discussion in the main text and appendix. In the single-distribution setting, Lei and Wasserman (2014) study size-optimal CP bands via conditional density estimation with convergence to oracle under regularity conditions. Bai et al. (2022) formulate efficient CP as constrained ERM, while Le Bars and Humbert (2025) analyze excess volume loss in split CP. Our setting is different: MDCP minimizes set size subject to simultaneous worst-case coverage over multiple sources, which leads to different analysis and techniques.
>
> 3. **Asymptotic assumptions and consistency (weakness and Q1)**
>
> We agree that this deserves careful discussion.
> - First, these assumptions are used only for asymptotic efficiency, not for finite-sample validity: uniform coverage holds regardless of the estimation consistency.
> - The asymptotic regime is reasonable under standard regularity conditions for consistent estimation of $f_k$’s and the pooled density (e.g., for parametric [2] or nonparametric conditional density/class-probability estimators [3]). Given these ingredients, Appendix A.3 / Thm 5 provides a sieve-based route to consistency of $\hat\lambda$ (under standard smoothness conditions). Thus, smaller $n_k$’s can slow oracle approximation and make MDCP conservative, but do not hurt the finite-sample validity. We have added discussion in the main text (under Thm. 3 and 5 for consistency of $\hat{f}_k$ and $\hat\lambda$ and connection to efficiency gap).
> - We **conducted experiments** varying (i) balance (ii) K, and (iii) N. **In summary**, imbalance, large K, and small N (which lead to small $n_k$ for some groups) *moderately* reduces efficiency due to lower estimation quality. However, such degradation appears in baselines too, and MDCP remains more efficient (with larger gaps in harder settings) than naive max-p, and maintains tight worst-case coverage. **Please refer to our response to Reviewer fxPA for result tables.**
>
> 4. **Typos**: corrected.
>
> 5. **Clarification on $T(x)$ (Q2)**
>
> $T(x)$ characterizes the optimal solution with ties at the threshold (handled by $U_k \sim \mathrm{Unif}[0,1]$ on Page 4). Setting $U_k=0$ excludes the cutoff and induces undercoverage (due to mass at threshold). Setting $U_k=1$ includes the cutoff, which preserves coverage but enlarges the prediction sets. In practice, the difference is often negligible, or one may add small noise to the scores to break ties.
> To fully address the point, we implemented simulations with $U_k=0$ and $U_k=1$. Across all 8 synthetic settings, the three options changed **worst-case coverage by at most 0.003** and **set size by at most 0.09**, so the practical difference is negligible. **We are happy to provide raw tables in later discussion.**
>
> 6. **Oracle baseline (Q3)**
>
> We agree that it’s helpful to separate the estimation component. We add a baseline that estimates $\lambda_k^\*(x)$ based on true $f_k$’s and applies MDCP (even if $f_k$’s are known, $\lambda_k^\*(x)$ is not available in closed form, so it still needs to be solved numerically from the dual). Across all 8 simulation settings, MDCP achieves (i) similarly tight worst-case coverage as Oracle and (ii) efficiency inferior but close to Oracle, and (iii) the gap decreases with sample size. The results are added to all figures in Section 5, and summarized below.
>
> |Setup|Cov (single-source/max-p/MDCP/Oracle)|Size (single-source/max-p/MDCP/Oracle)|
> |---|---:|---:|
> |Class.linear|0.578/0.962/0.901/0.903|2.18/3.82/2.55/2.27|
> |Class. interaction|0.578/0.963/0.903/0.907|2.14/3.79/2.52/2.17|
> |Class.sinusoid|0.580/0.965/0.900/0.905|2.16/3.85/2.45/2.19|
> |Class.softplus|0.665/0.977/0.903/0.907|2.70/4.29/2.30/2.09|
> |Reg.linear|0.483/0.958/0.901/0.905|4.13/7.17/5.63/4.60|
> |Reg.interaction|0.799/0.983/0.905/0.905|14.11/21.40/13.12/9.42|
> |Reg.sinusoid|0.539/0.964/0.906/0.905|5.02/8.50/6.46/5.22|
> |Reg.softplus|0.563/0.964/0.902/0.904|5.26/8.73/6.52/5.30|
>
> [2] Van der Vaart, A. W. (2000). Asymptotic statistics (Vol. 3). Cambridge university press.
>
> [3] Györfi, László, et al. A distribution-free theory of nonparametric regression. Springer New York, 2002.

---

> > ### Author Rebuttal · Reviewer_4NE6 · 2026-04-02
> >
> > The authors have addressed most of my concerns. I have increased my score accordingly, assuming that the promised revisions will be incorporated in the final version of the paper.

---

### Official Review · Reviewer_fxPa · 2026-03-09

**Soundness:** 3
**Presentation:** 4
**Significance:** 3
**Originality:** 3
**Overall Recommendation:** 4
**Confidence:** 3

**Summary:**

The paper proposes a framework (MDCP) to construct conormal prediction sets that achieve uniform coverage guarantee over subpopulations, using the max-p strategy. They obtain theoretically optimal score function that achieves said guarantee using max-p aggregation strategy. The paper then outlines the protocol to learn the score functions and construct prediction sets in practice. Experiments with synthetic and real data demonstrates the effectiveness of the proposed approach and validates theoretical results.

The takeaway message for me was that suitably chosen score functions combined with max-p aggregation strategy leads to uniform coverage guarantees.

Overall, the paper is well-written and presented, with extensive experimental validation.

**Compliance With Llm Reviewing Policy:**

Affirmed.

**Final Justification:**

Authors satisfactorily addressed most of the concerns. Additional experiments are helpful. In my opinion, paper provides new insights into challenging problem of conformal prediction within multi-distribution settings and will be of interest to the readers of ICML. Hence, I maintain my positive assessment of the paper. However, some theoretical trade-offs such as group sizes and efficiency gains are not explored deeply. However, paper strengths dominates the weakness.

**Key Questions For Authors:**

1. How does the proposed MDCP framework perform on unbalanced groups in simulation studies? (weakness 2 above)

2. The trade-off between subgroup sizes and achievable efficient prediction sets needs to be investigated further. The paper assumes subgroup sizes are large enough to estimate weights and conditional densities, which can be unrealistic. Can you provide more discussion  in this direction?

3. Computational cost of estimating weights and overall procedure is not discussed. How does it relate to $K$?

**Limitations:**

No, the paper could discuss why the choice of the distribution shifts considered is made, how it is restrictive and distinct from other forms of distribution shifts considered in the literature. Also, such distribution shifts in real-world settings seems unnatural because of the assumed fixed $K$ known groups. Furthermore, computational burden of solving the optimization problem, in relation to $K$ should be addressed as well.

**Strengths And Weaknesses:**

**Strengths**

1. The paper focuses on the problem of constructing conformal prediction sets with a common threshold for scores to achieve uniform coverage guarantee (implying there is no need to know the source/group identity at the test time). The problem seems relevant in many contexts such as fairness, multi-source data, etc. as motivated in the paper.

2. Theoretical results regarding optimal score functions and worst-case coverage guarantees are interesting.


---


**Weaknesses**

1. The problem setup considered assumes fix number of subpopulations $K$. Conceptually, how is the proposed approach in terms of optimal scores differs from estimating mixture conditional probability distribution.

2. All the simulations are presented with balanced groups, but motivated examples such as fairness arises when we have unbalanced groups. Authors may add these experiments to validate their results empirically in such adverse settings.

---

> ### Author Rebuttal · Authors · 2026-03-30
>
> Thank you for appreciating the relevance of the problem, the technical soundness, the extensive experiments, and the presentation of the paper. **Due to space limits, throughout, we report a subset of results, but messages are similar across all settings, and we’d be happy to provide full tables in subsequent discussion.**
>
> 1. **Difference from estimating mixture cond. distribution**:
>
> Thanks for pointing out this connection. Our optimal score function turns out to take the form $h^\*(x,y)= \sum_{k=1}^K \lambda_k^\*(x) p_{k}(y | x)$; however, this is not mixture-density estimation in the usual sense. The coefficients $\lambda^\*(x)$ are unknown and arise as dual variables from the size-minimization problem; they are not source proportions or membership weights, and they need not sum to 1. More importantly, the objective is different: estimating a pooled mixture targets average prediction error, whereas MDCP optimizes prediction-set size subject to worst-case coverage.
>
> 2. **Imbalance, group size-efficiency tradeoff**:
>
> We agree this deserves discussion.
>
> - Small subgroup sample sizes affect efficiency, not finite-sample validity: max-p agg. remains uniformly valid for any score, while larger $n_k$ is only needed to accurately estimate $\hat f_k$​ and $\hat\lambda$ and approach the oracle-optimal set. In our asymptotic result, consistency of $\hat f_k$​​ and $\hat\lambda$ drives the size gap. When some groups are small, these estimates become noisier and the sets can be wider. We will explicitly discuss them below Thm. 3 and 5.
> - Our real-data studies already include substantially imbalanced groups (e.g., FMoW is highly imbalanced across regions), where MDCP still improves efficiency over naive agg. while maintaining tight worst-case coverage.
> - In response, we have varied the group-imbalance ratio $\rho=\max_k n_k/\min_k n_k$ using geometric allocation for $n_k$’s. We measure worst-case coverage, set size, and runtime for MDCP and naive max-p when (1) vary $\rho$ and fix N, (2) fix $\rho$ and vary N. Across settings, (i) imbalance and (ii) small N reduce efficiency due to degraded estimation quality, yet these affect all procedures, and MDCP remains much more efficient than max-p (large gap in harder problems). (iii) They do not affect validity: MDCP still has near-tight worst-case coverage. Results:
>   - a) **Varying $\rho$, fix total sample size N=6000**: (results are MDCP/max-p)
>
> |Setup|rho|Cov.|Size|Runtime (sec)|
> |---|---|---:|---:|---:|
> |Class.linear|1|0.903/0.962|2.42/3.64|54.8/7.9|
> |-|4|0.902/0.964|3.34/3.82|52.1/6.7|
> |-|8|0.904/0.969|3.67/4.24|51.3/5.7|
> |Reg. interaction|1|0.903/0.983|13.55/22.16|5.3/2.2|
> |-|4|0.904/0.985|14.20/24.60|4.7/1.9|
> |-|8|0.912/0.987|15.73/29.15|4.9/1.7|
>
>   - b) **Vary N**:
>
> |Setup|N|Cov.|Size|Runtime (sec)|
> |---|---|---:|---:|---:|
> |Class.linear|1500|0.914/0.975|2.87/4.57|19.8/2.2|
> |-|3000|0.907/0.968|2.63/4.01|30.2/3.9|
> |-|6000|0.902/0.960|2.45/3.59|52.8/7.7|
> |Reg. interact|1500|0.917/0.985|18.35/34.35|1.9/0.9|
> |-|3000|0.909/0.986|15.10/27.28|3.2/1.3|
> |-|6000|0.903/0.983|13.55/22.16|5.3/2.2|
>
> 3. **Computation cost**: Thanks for raising this point! **Takeaway**: Runtime increases with sample size $N$, while the effects of imbalance and $K$ are mild when $N$ is fixed.
>
> - See **last column in tables above** for runtime varying $N$ and imbalance.
> - Vary K, fix N=6000: (MDCP/max-p)
>
> |Setup|K=2 (sec)|K=5|K=10|
> |---|---|---:|---:|
> |class. linear|51.5/6.8|56.3/8.9|61.7/10.6|
> |reg. interaction|4.3/1.8|6.0/2.7|7.4/3.8|
>
> - **Runtime breakdown** (N=6000, K=3) shows the dominant component is $\lambda$-optimization, but it remains scalable (less than 1 min on 1-core CPU for 6000 samples).
>
> |Setup|Source-model training (sec)|Score opt. (sec)|Max-p agg. (sec)|Total (sec)|
> |---|---:|---:|---:|---:|
> |Class.linear|6.52|47.23|1.99|55.74|
> |Reg.interaction|2.04|3.00|0.51|5.55|
>
>
> 4. **Rationale of distribution shift**:
>
> We study a structured but practically important regime in which the calibration data come from a finite family of known sources, and the test distribution can be any mixture over these sources. This covers settings such as subpopulation shift and fairness; in these heavily-studied settings, MDCP can be viewed as the predictive inference counterpart of group DRO [1] (group-wise worst-case coverage v.s. worst-case performance).
> Our distribution shift is more structured than arbitrary shift, which makes principled calibration possible. It’s distinct from covariate shift and label shift where the test distribution is identifiable by reweighting; here both $P^{(k)}(X)$ and $P^{(k)}(Y\mid X)$ can vary across sources. We will make explicit the restrictions here: we assume a fixed, known source family with data for each source, and do not directly cover unseen sources or misspecified sources. We have added this discussion to Appendix A.1.
>
> [1] Hashimoto, T., Srivastava, M., Namkoong, H., and Liang, P. Fairness without demographics in repeated loss minimization. ICML 2018.

---

> > ### Author Rebuttal · Reviewer_fxPa · 2026-04-02
> >
> > I thank the authors for their detailed response. Additional experiments are helpful. I will maintain my already positive assessment.

---

### Official Review · Reviewer_uC1S · 2026-03-17

**Soundness:** 3
**Presentation:** 3
**Significance:** 3
**Originality:** 3
**Overall Recommendation:** 4
**Confidence:** 4

**Summary:**

The paper studies a setting where labeled data may come from multiple source distributions while the test data can come from an arbitrary member or a mixture of them. In this setting, the paper proposes a framework, Multi-Distribution Conformal Prediction (MDCP), to construct prediction sets that balance the efficiency-validity tradeoff and guarantee uniform coverage over the sources. To achieve this, the authors introduce a max-p aggregation scheme and show the optimality and tightness of this scheme. The paper also presents separate end-to-end algorithms for classification and regression. Authors demonstrate the performance of their approach on simulations and real data.

**Compliance With Llm Reviewing Policy:**

Affirmed.

**Final Justification:**

I maintain my recommendation of a weak accept. The authors mentioned they will include the relevant related work and clarify definitions in the updated version. Further, a discussion of the limitations should be included in the updated version. Given these discussions are included, the paper's contributions and merits will be beneficial for the community.

**Key Questions For Authors:**

I do not have specific questions for authors currently. I would request the authors to address the comments above and I can follow up with questions during the discussion period.

**Limitations:**

The authors have currently not discussed the limitations and potential negative societal impact of their work. While the conclusion briefly mentions some open questions and the paper discusses potential positive societal impact in the impact statement, an extended discussion on the limitations and potential negative societal impact is required in my view.

**Strengths And Weaknesses:**

**Soundness:** The submission is technically sound theoretically and empirically. The theoretical results are complete and appear correct. The algorithms are well-motivated and appropriately explained. The assumptions are stated clearly and the results are sufficiently explained in the appendix. Empirical evaluation over both simulated and real data is appreciated, where the proposed algorithms outperform the baselines.

**Presentation:** The paper is well-structured overall. The paper is largely well-written; however the terms conditional and marginal validity/coverage should be formally defined to avoid confusion with their standard $X-$conditional notions.

A major comment here is regarding missing important related work. The paper discusses the group-conditional coverage guarantees in Gibbs et al., 2025, but the method in Gibbs et al. also guarantees conditional coverage over a class of covariate shifts. This should be acknowledged and compared with in the paper. Similarly, Conformal Prediction Under Covariate Shift by Tibshirani et., 2019 should be cited and discussed. I would encourage authors to do a more thorough review of prior and concurrent literature.

**Significance:** The problem studied in the paper is an important one and has diverse and significant implications. The flexible and principled methods proposed in the paper aim to address this problem and the effectiveness of the approach is demonstrated through multiple experiments.

**Originality:** The paper’s originality lies in proposing methods for a setting where the test distribution is unknown or a mixture of the sources, which calls for new techniques. This is different from past work where subgroup information is generally required during the test time. I would encourage the authors to include missing related work and extend the discussion.

---

> ### Author Rebuttal · Authors · 2026-03-30
>
> We thank reviewer uC1S for the appreciation of the technical and empirical soundness as well as the writing quality of our manuscript. We respond to the questions and comments below.
>
> 1. **Formally define conditional coverage**: we have added the formal definition of multi-distribution conditional coverage in the first paragraph of Section 3.1: $\min_{k\in[K]}\mathbb{P}_{P^{(k)}}(Y\in C(X)\mid X=x)\geq 1-\alpha$, and clearly distinguish it from distribution-free conditional coverage in the standard conformal sense. We have also clarified that ``marginal coverage’’ refers to equation (1).
>
> 2. **Related literature**: Thank you for highlighting this connection. This is currently only briefly discussed in Appendix A.1, and we agree it should be made explicit in the paper. We expand the discussion in the manuscript as follows:
>
> > In particular, Gibbs et al. (2025) view the group membership as part of the covariate, and cast the problem as protecting against a family of covariate shifts (e.g., due to changes in the group membership), which further relates to conformal prediction under covariate shift (Tibshirani et al., 2019). Our setting differs in two important ways: (i) the source/group identity is not observed at test time, so one must output a single prediction set that works for all the sources; and (ii) MDCP allows both $P^{(k)}(X)$ and $P^{(k)}(Y|X)$ to vary across sources, rather than assuming only a shift in $P(X)$. This is because once the group membership is unknown, the distribution shift between labeled sources and test data cannot be captured by a covariate shift assumption. As such, the techniques we develop are in sharp contrast to those methods.
>
> We would be more than happy to discuss further and expand on any thread of literature the reviewer finds necessary.
>
> 3. **Limitations and potential negative societal impact**: We agree that an explicit discussion is needed. We add the following statement in the manuscript:
>
> > While our framework aims to mitigate the distribution shift issue across distributions, some limitations and risks are worth discussing. First, the worst-case marginal coverage guarantee operates at the population level and does not preclude undercoverage for specific individuals. Second, in fairness-motivated applications, we still require source/group labels during calibration, which might not be available in highly sensitive scenarios. Finally, if calibration data for a source is scarce, MDCP may produce unnecessarily wide prediction sets for that group, potentially lowering the utility in the underrepresented groups even though the uncertainty quantification is valid.

---

> > ### Author Rebuttal · Reviewer_uC1S · 2026-04-03
> >
> > Thanks for the responses! I will look forward to the extended discussion in the updated paper. I continue to support acceptance of the paper.

---

### Decision · Program_Chairs · 2026-04-30

**Decision:**

Accept (regular)

**Comment:**

This paper proposes a novel method for learning conformity scores for coverage validity for multiple distributions. The reviewers agree on the novelty and technical soundness and share concerns on the coverage of related work. One reviewer also recommend having more discussion to explore the tradeoff between group sizes and efficiency gain. I also agree with the comment that a discussion about conformal methods under different distribution shift (covariate shift and label shift) will be helpful. I recommend authors to incorporate suggestions from the reviewers to improve the paper.